

# (SPT-)LSM theorems from projective non-invertible symmetries

**Salvatore D. Pace, Ho Tat Lam and Ömer M. Aksoy**

Department of Physics, Massachusetts Institute of Technology, Cambridge, MA 02139, USA

## Abstract

Projective symmetries are ubiquitous in quantum lattice models and can be leveraged to constrain their phase diagram and entanglement structure. In this paper, we investigate the consequences of projective algebras formed by non-invertible symmetries and lattice translations in a generalized $1 + 1$D quantum XY model based on group-valued qudits. This model is specified by a finite group $G$ and enjoys a projective $\text{Rep}(G) \times Z(G)$ and translation symmetry, where symmetry operators obey a projective algebra in the presence of symmetry defects. For invertible symmetries, such projective algebras imply Lieb-Schultz-Mattis (LSM) anomalies. However, this is not generally true for non-invertible symmetries, and we derive a condition on $G$ for the existence of an LSM anomaly. When this condition is not met, we prove an SPT-LSM theorem: any unique and gapped ground state is necessarily a non-invertible weak symmetry protected topological (SPT) state with non-trivial entanglement, for which we construct an example fixed-point Hamiltonian. The projectivity also affects the dual symmetries after gauging $\text{Rep}(G) \times Z(G)$ sub-symmetries, giving rise to non-Abelian and non-invertible dipole symmetries, as well as non-invertible translations. We complement our analysis with the SymTFT, where the projectivity causes it to be a topological order non-trivially enriched by translations. Throughout the paper, we develop techniques for gauging $\text{Rep}(G)$ symmetry and inserting its symmetry defects on the lattice, which are applicable to other non-invertible symmetries.



# 1 Introduction

One of the most fruitful aspects of symmetries in quantum many-body systems and quantum field theory (QFT) is their ability to constrain physical phenomena non-perturbatively. A straightforward example is how the symmetry's associated conservation laws constrain the system's time evolution and provide selection rules on correlation functions. Beyond these, symmetries also offer much more powerful constraints, particularly on the allowed quantum phases (i.e., on renormalization group flows) and the entanglement structure of ground state wavefunction.

Symmetries with 't Hooft anomalies are such examples. They are defined as symmetries that are incompatible with symmetry protected topological (SPT) phases—gapped phases characterized by a non-degenerate, symmetric ground state on all closed spatial manifolds [1–10]. As a result, 't Hooft anomalies enforce *all* symmetric systems to have gapless excitations or long-range/topological order and have ground states with long-range entanglement [11].

't Hooft anomalies are also obstructions to gauging the global symmetries, which can be used as a method to detect them. While initially discovered for continuous symmetries in QFTs with fermions [12–15], 't Hooft anomalies can arise generically for continuous or discrete symmetries [16, 17], with or without fermions, and in the continuum or on the lattice [6, 18–22]. For ordinary internal symmetries, 't Hooft anomalies are in one-to-one correspondence with SPTs in one higher dimension through anomaly inflow, endowing the spatial boundaries of SPT phases with zero modes and gapless excitations [23, 24].

In lattice models, there can exist mixed 't Hooft anomalies between internal and crystalline symmetries. Such anomalies are called Lieb-Schultz-Mattis (LSM) anomalies, and their obstructions to SPT phases lead to the celebrated LSM Theorem [25–31]. While its original incarnation applies to half-integer spin chains with $SO(3)$ spin-rotation and translation symmetries, LSM anomalies have been extended to systems in higher dimensions [28–31], with only discrete internal symmetries [4, 32–35], with crystalline symmetries beyond translations [36–44], with long-ranged interactions [45, 46] and that are fermionic [47–50]. When there is an LSM anomaly, it is usually the case that the entire internal symmetry group cannot be gauged while preserving the crystalline symmetries [22]. It is similar to mixed 't Hooft anomalies of internal symmetries that prevent a sub-symmetry from being gauged without explicitly breaking the rest of the symmetry. Furthermore, gauging internal sub-symmetries leads to dual internal and crystalline symmetries with a non-trivial interplay [22, 51], similar to the effect of 't Hooft anomalies of internal symmetries on their dual symmetries [52–54]. In fact, 't Hooft anomalies of internal symmetries in continuum QFTs can match LSM anomalies in quantum lattice models [21, 55–58]. Similar to 't Hooft anomalies, there is an anomaly inflow mechanism for LSM anomalies where the anomaly inflow theory is a crystalline SPT phase [49, 59–62].

't Hooft and LSM anomalies formally arise from non-trivial projective phases produced when fusing and deforming (networks of) symmetry defects, as well as when multiplying symmetry operators in the presence of their defects. Anomalies, however, are not the only constraints on quantum phases and entanglement patterns from these projective phases. For example, consider two on-site, unitary symmetry operators $\mathcal{U} = \prod_j U_j$ and $\mathcal{V} = \prod_j V_j$ that *locally* form the projective representation $U_j V_j = e^{i\theta_j} V_j U_j$. While this local projective algebra implies an LSM anomaly involving lattice translations when $\mathcal{U}$ and $\mathcal{V}$ are translation invariant, it does not do so generally.[1] When an LSM anomaly is absent, these local projective representations can nevertheless still enforce the allowed SPT phases to have non-trivial short-range entanglement (i.e., prevent a symmetric product state from existing). Such constraints on the allowed entanglement structure of SPTs are called SPT-LSM constraints [62–66] and can occur from magnetic translations and similar interplays between crystalline and internal symmetries.

Alongside these developments on anomalies of ordinary symmetries, there has been a recent flurry of interest in generalizing the notion of global symmetries (see Refs. 67–72 for reviews). With more symmetries come more opportunities for their non-perturbative applications, and like ordinary symmetries, these generalized symmetries have corresponding conservation laws, can characterize phases of matter, and can have 't Hooft anomalies. A particular generalization of symmetries is non-invertible symmetries whose symmetry transformations do not have an inverse and therefore are not described by group theory but (higher)-categorical

---

[1]A simple example is a translation-invariant $1 + 1$D system of $\mathbb{Z}_4 \times \widetilde{\mathbb{Z}}_4$ qudits on sites $j$ with on-site symmetries $\mathcal{U} = \prod_j X_j \widetilde{X}_j$ and $\mathcal{V} = \prod_j (Z_j \widetilde{Z}_j)^{2j+1}$ ($X_j$, $\widetilde{X}_j$, $Z_j$, and $\widetilde{Z}_j$ are the qudits' shift and clock operators). While these symmetries are projectively represented locally, satisfying $U_j V_j = -V_j U_j$ with $U_j = X_j \widetilde{X}_j$ and $V_j = (Z_j \widetilde{Z}_j)^{2j+1}$, there is no LSM anomaly since there exists SPT states [63]. An LSM anomaly is absent because $\mathcal{V}$ is a modulated symmetry operator that does not commute with lattice translations $T$. Indeed, from a symmetry defect perspective, the local projective algebra $U_j V_j = -V_j U_j$ does not imply projectivity in the $\mathcal{U}$ symmetry defect Hilbert space, where the twisted translation operator $T_{\text{tw}} = U_1 T$ satisfies $T_{\text{tw}} \mathcal{V} = \mathcal{W} \mathcal{V} T_{\text{tw}}$ where $\mathcal{W} = -\prod_j (Z_j \widetilde{Z}_j)^2$ itself is a symmetry operator.

Table 1: The projective algebras arising from inserting $\text{Rep}(G) \times Z(G) \times \mathbb{Z}_L$ symmetry defects in the $G$-based XY model, where the phase $e^{i\phi_\Gamma(z)} = \chi_\Gamma(z)/d_\Gamma$ with $d_\Gamma$ the dimension of the irrep $\Gamma$ of $G$ and $\chi_\Gamma$ the character of $\Gamma$. $T_{\text{tw}}^{(\bullet)}$ denotes the twisted translation operator in the presence of a $\bullet$ symmetry defect. These projective algebras follow from a local projective algebra of matrix product operators on each site of the spatial lattice (see Eq. (44)).

| Translation defects | $z \in Z(G)$ defect | $\Gamma \in \text{Rep}(G)$ defect |
|---|---|---|
| $R_\Gamma U_z = \left( e^{i\phi_\Gamma(z)} \right)^L U_z R_\Gamma$ | $R_\Gamma T_{\text{tw}}^{(z)} = e^{i\phi_\Gamma(z)} T_{\text{tw}}^{(z)} R_\Gamma$ | $T_{\text{tw}}^{(\Gamma)} U_z = e^{i\phi_\Gamma(z)} U_z T_{\text{tw}}^{(\Gamma)}$ |

theory [52, 53, 73–75]. Like ordinary symmetries, non-invertible symmetries can be anomalous and constrain phases and RG flows, which have been explored extensively in relativistic QFTs [73, 76–83].

Many aspects of non-invertible symmetries in QFTs also appear in lattice models. For example, anyon chains [84–88] and fusion surface models [89–91] are constructed to have fusion (higher)-categorical symmetry, which appears in relativistic QFTs. Understanding anomalies and similar constraints from non-invertible symmetries in quantum lattice models offers new non-perturbative tools to study these systems. On the lattice, anomalous non-invertible symmetries appear commonly either in non-tensor product Hilbert spaces or do not act fully internally by mixing with crystalline symmetries [92–96]. A natural question then is whether we can derive constraints on possible quantum phases and their low-energy entanglement structure from non-invertible symmetries that act fully internally on a tensor product Hilbert space.

## 1.1 Summary

In this paper, we explore the physical consequences and signatures of a projective algebra involving non-invertible internal symmetries and translation symmetries in quantum spin chain models defined on a tensor product Hilbert space. We focus on a $\text{Rep}(G) \times Z(G)$ internal symmetry, where $G$ is a finite group, $\text{Rep}(G)$ is the representation category of $G$, and $Z(G) = \{z \in G \mid zg = gz, \ \forall g \in G\}$ is the center of $G$. This is a non-invertible symmetry whenever $G$ is non-Abelian. In a periodic length $L$ chain, we consider the scenario where this internal symmetry gives rise to non-trivial projective algebras involving $\mathbb{Z}_L$ lattice translations. Denoting by $R_\Gamma$, $U_z$, and $T$ the $\text{Rep}(G)$, $Z(G)$, and $\mathbb{Z}_L$ symmetry operators, respectively, these projective algebras arising from inserting symmetry defects and are summarized in Table 1.

This $\text{Rep}(G) \times Z(G) \times \mathbb{Z}_L$ symmetry arises in a generalized $1 + 1$D XY model, defined on a tensor product Hilbert space of $G$ qudits and constructed from $G$-based Pauli operators. When $G = \mathbb{Z}_2$, this model reduces to the ordinary quantum XY model. After reviewing $G$ qudits and their group-based Pauli operators [97] in Section 2, we introduce this $G$-based XY model in Section 3. We contextualize much of our discussion of the $\text{Rep}(G) \times Z(G) \times \mathbb{Z}_L$ symmetry in the $G$-based XY model, but our results and discussion apply to all symmetric Hamiltonians. For example, to derive Table 1, we insert $\text{Rep}(G)$ symmetry defects in Section 3.2 into the $G$-based XY model. However, Table 1 is an intrinsic property of the symmetry regardless of the Hamiltonian. Furthermore, the method used for inserting the $\text{Rep}(G)$ symmetry defects applies generally to all symmetries in $1 + 1$D quantum lattice models represented by matrix product operators.

We introduce, and discuss the physical consequences of, this projective algebra in Section 4. In particular, depending on the properties of $G$, the projective algebras can give rise to an LSM anomaly for the $\text{Rep}(G) \times Z(G) \times \mathbb{Z}_L$ symmetry or a new type of SPT-LSM constraint for non-

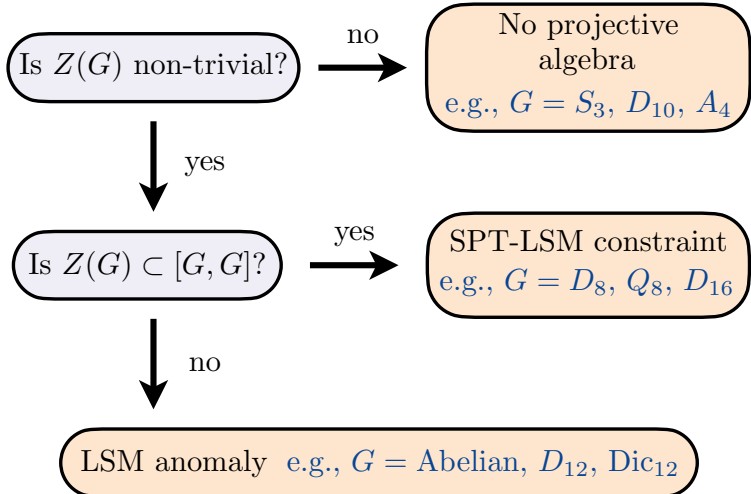

Figure 1: The physical consequences of the projective $\text{Rep}(G) \times Z(G) \times \mathbb{Z}_L$ symmetry depend on properties of $G$ as summarized in this flow chat. Namely, if the center $Z(G) = \{z \in G \mid zg = gz \; \forall g \in G\}$ of $G$ is trivial, then the projective algebras are trivial and there are no physical consequences. If $Z(G)$ is non-trivial, then the projective algebra can give rise to an LSM anomaly or SPT-LSM constraint depending on if $Z(G)$ is contained in the commutator subgroup $[G, G] = \{g^{-1}h^{-1}gh \mid \forall g, h \in G\}$ of $G$. For the examples for $G$: $S_n$ is the symmetric group of degree $n$, $D_{2n}$ is the dihedral group of order $2n$, $A_n = [S_n, S_n]$ is the alternating group of degree $n$, $Q_8$ is the quaternion group, and $\text{Dic}_n$ is the dicyclic group of order $n$.

invertible weak SPTs. As summarized by Fig. 1, whether the projective algebra disallows an SPT or allows one with non-trivial entanglement depends on whether $Z(G)$ is contained in the commutator subgroup $[G, G]$ of $G$. When it is not, we show that the $\text{Rep}(G) \times Z(G) \times \mathbb{Z}_L$ symmetry has an LSM anomaly that disallows an SPT and enforces the ground state to have long-range entanglement. This generalizes the LSM anomaly in the $G = \mathbb{Z}_2$, ordinary XY model [4, 32, 34]. When $Z(G)$ is in $[G, G]$, there is an SPT-LSM constraint that enforces all allowed SPTs to have non-trivial entanglement and be a non-invertible weak SPTs. In particular, it is an SPT where translation defects are dressed by $\text{Rep}(G)$ symmetry charges, which causes the ground state to be annihilated by the $\text{Rep}(G)$ symmetry operators for particular system sizes. We argue that this SPT-LSM constraint is intrinsic to the projective algebra, and arises regardless of the microscopic degrees of freedom. Furthermore, in Section 4.4, we construct and analyze an exactly solvable model for $G = D_8$ realizing such a non-invertible weak SPT. These weak SPTs for $G = D_8$ are in one-to-one correspondence with SPTs protected by an internal $\text{Rep}(D_8) \times \mathbb{Z}_2 \times \mathbb{Z}_2$ symmetry, as indicated by the crystalline equivalence principle [59]. In Appendix C, we construct and classified the topological quantum field theories (TQFTs) for these $\text{Rep}(D_8) \times \mathbb{Z}_2 \times \mathbb{Z}_2$ SPTs.

The projective algebras arising from the $\text{Rep}(G) \times Z(G) \times \mathbb{Z}_L$ symmetry has signatures beyond its non-perturbative constraints. Namely, they affect the gauging web of the $\text{Rep}(G) \times Z(G) \times \mathbb{Z}_L$ symmetry. We explore this gauging web in Section 5, which is summarized in Fig. 2. We construct it by directly gauging the $Z(G)$ and the non-invertible $\text{Rep}(G)$ symmetries on the lattice and construct quantum lattice models with new symmetries: non-Abelian and non-invertible dipole symmetries and non-invertible translation symmetries. We discuss various details of these gauging procedures in Appendix D, and find that it is straightforward to gauge $\text{Rep}(G)$ with the help of the group-based Pauli operators. In fact, our gauging procedure can be generalized to gauge any $\text{Rep}(H)$ symmetry, where $H$ is a Hopf algebra, us-

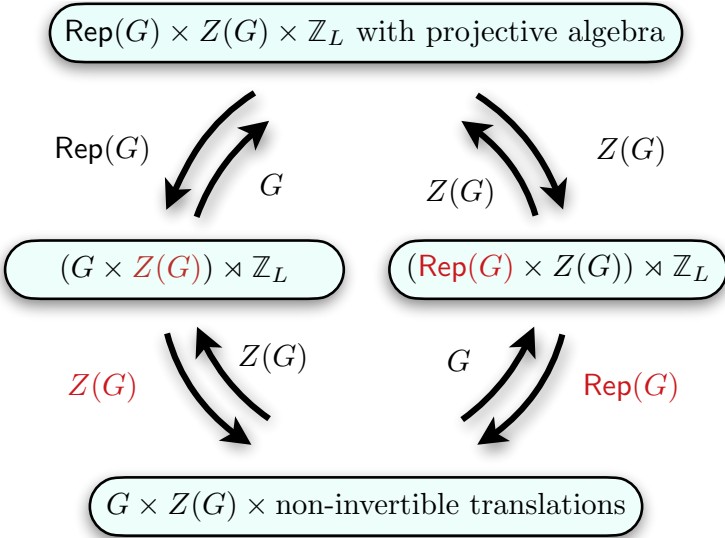

Figure 2: The projective algebra arising from the $\mathrm{Rep}(G) \times Z(G)$ internal and $\mathbb{Z}_L$ translation symmetry leads to the gauging web shown, where arrows between dual symmetries are labeled by the internal sub-symmetry gauged. Because of the projective algebra, the translation symmetry plays a non-trivial role in all gauged models, either giving rise to dual modulated symmetries (colored in red) or becoming non-invertible.

ing Hopf algebra based Pauli operators [98]. We complement this gauging web using the symmetry topological field theory (SymTFT) in Section 6. The corresponding SymTFT is a symmetry enriched topological order (SET) with translation symmetry acting non-trivially on the anyons.

## 2 $G$-qudits and group-based Pauli operators

In this section, we review group-based qudits and their generalized Pauli operators [97], which we extensively use throughout the paper. Our discussion will specialize to systems whose space is a finite periodic chain of length $L$ such that on each site $j$ resides a degree of freedom labeled by group elements $g \in G$ — a $G$-qudit — where $G$ is a finite group. We denote by $\bar{g} \in G$ the inverse of element $g$. The system's Hilbert space then has the tensor product decomposition:

$$\mathcal{H} = \bigotimes_j \mathcal{H}_j, \qquad \mathcal{H}_j \cong \mathbb{C}[G] := \mathbb{C}^{|G|}, \tag{1}$$

where $|G|$ denotes the order of $G$. Any state $|\psi\rangle$ in the $|G|^L$-dimensional Hilbert space $\mathcal{H}$ can be expanded in a basis in which each basis state is labeled by a string of elements in $G$, i.e., $|\psi\rangle \equiv |g_1, g_2, \cdots, g_L\rangle$ with $g_j \in G$. We use the collective label $|\boldsymbol{g}\rangle$ to denote the state labeled by such a string of group elements, i.e., $|\boldsymbol{g}\rangle := |g_1, g_2, \cdots, g_L\rangle$, and the short-hand notation $|g_j \boldsymbol{h}\rangle$ to denote the state $|h_1, h_2, \cdots, g h_j, \cdots, h_L\rangle$.

Because states in the local Hilbert space $\mathbb{C}[G]$ are labeled by the elements in $G$, they transform under a regular representation of the group $G$. It is then convenient to introduce operators implementing the group action on $\mathbb{C}[G]$. These are the so-called group-based Pauli operators [97] (see Ref. 98 for a further generalization using Hopf algebras) that generalize the Pauli $\sigma^x$ and $\sigma^z$ matrices acting on qubits, i.e., $G = \mathbb{Z}_2$-qudits.

The group-based $X$ operators acting on the $G$-qudit at site $j$ are defined as

$$\overrightarrow{X}_j^{(g)} = \sum_{\{\boldsymbol{h}\}} \left| g_j \, \boldsymbol{h} \middle\rangle\!\middle\langle \boldsymbol{h} \right|, \tag{2a}$$

$$\overleftarrow{X}_j^{(g)} = \sum_{\{\boldsymbol{h}\}} \left| \boldsymbol{h} \, \bar{g}_j \middle\rangle\!\middle\langle \boldsymbol{h} \right|, \tag{2b}$$

which implement left and right multiplication by $g$ and $\bar{g}$ at site $j$, respectively. Furthermore, it is convenient to introduce the notation $\overleftrightarrow{X}_j^{(g)} = \overrightarrow{X}_j^{(g)} \overleftarrow{X}_j^{(g)}$. From their definitions, the group-based $X$ operators satisfy

$$\begin{aligned}
\overrightarrow{X}_j^{(g)} \overrightarrow{X}_j^{(h)} &= \overrightarrow{X}_j^{(gh)}, & \overleftarrow{X}_j^{(g)} \overleftarrow{X}_j^{(h)} &= \overleftarrow{X}_j^{(gh)}, \\
\left(\overrightarrow{X}_j^{(g)}\right)^\dagger &= \overrightarrow{X}_j^{(\bar{g})}, & \left(\overleftarrow{X}_j^{(g)}\right)^\dagger &= \overleftarrow{X}_j^{(\bar{g})},
\end{aligned} \tag{3a}$$

and obey the commutation relations:

$$\left[\overrightarrow{X}_i^{(g)}, \overleftarrow{X}_j^{(h)}\right] = 0, \tag{3b}$$

$$\left[\overrightarrow{X}_i^{(g)}, \overrightarrow{X}_j^{(h)}\right] = \delta_{ij}\left(\overrightarrow{X}_i^{(gh)} - \overrightarrow{X}_i^{(hg)}\right), \tag{3c}$$

$$\left[\overleftarrow{X}_i^{(g)}, \overleftarrow{X}_j^{(h)}\right] = \delta_{ij}\left(\overleftarrow{X}_i^{(gh)} - \overleftarrow{X}_i^{(hg)}\right). \tag{3d}$$

Therefore, when $G$ is non-Abelian, the group-based $X$ operators do not all commute with one another. We note that for $z \in Z(G)$, the center of the group $G$, the left and right multiplication operators are identified as $\overrightarrow{X}_j^{(z)} = \overleftarrow{X}_j^{(\bar{z})}$. These operators commute with all other group-based $X$ operators. When $G$ is Abelian, all the left and right multiplication operators are identified, and they all commute with each other.

The group-based $Z$ operator acting on site $j$ is the matrix product operator (MPO) tensor

$$[Z_j^{(\Gamma)}]_{\alpha\beta} = \sum_{\{\boldsymbol{h}\}} [\Gamma(h_j)]_{\alpha\beta} \, |\boldsymbol{h}\rangle\langle\boldsymbol{h}| \equiv \alpha \, \blacksquare\!\!-\!\!\boxed{Z_j^{(\Gamma)}}\!\!-\!\!\blacksquare \, \beta \tag{4}$$

where $\Gamma\colon G \to \mathrm{GL}(d_\Gamma, \mathbb{C})$ is a $d_\Gamma$-dimensional irrep of $G$ acting on a $d_\Gamma$-dimensional virtual Hilbert space. The indices $\alpha, \beta \in \{1, 2, \cdots, d_\Gamma\}$ label states of this virtual Hilbert space such that $[\Gamma(g_j)]_{\alpha\beta}$ is the matrix element of representation $\Gamma(g_j)$ between virtual states $|\alpha\rangle$ and $|\beta\rangle$. In what follows, we assume summation over repeated virtual indices. From the properties of irreps, the group-based $Z$ operators satisfy

$$[Z_j^{(\Gamma)}]_{\alpha\gamma}[Z_k^{(\Gamma)}]_{\gamma\beta} = \sum_{\{\boldsymbol{h}\}} [\Gamma(h_j h_k)]_{\alpha\beta} \, |\boldsymbol{h}\rangle\langle\boldsymbol{h}|. \tag{5}$$

Furthermore, contracting over the virtual space yields the MPO

$$\mathrm{Tr}[Z_j^{(\Gamma)}] = \sum_{\{\boldsymbol{h}\}} \chi_\Gamma(h_j) \, |\boldsymbol{h}\rangle\langle\boldsymbol{h}| \equiv \boxed{Z_j^{(\Gamma)}} \tag{6}$$

where $\chi_\Gamma(g) = \mathrm{Tr}[\Gamma(g)]$ is the character of the conjugacy class $[g]$ in irrep $\Gamma$. Because of the group-theoretic identity

$$\sum_\Gamma d_\Gamma \chi_\Gamma(g) = |G| \, \delta_{g,1}, \tag{7a}$$

the group-based $Z$ operators satisfy

$$\frac{1}{|G|}\sum_\Gamma d_\Gamma \operatorname{Tr}[Z_j^{(\bar\Gamma)} Z_k^{(\Gamma)}] = \sum_{\{\boldsymbol g\}} \delta_{g_j, g_k} |\boldsymbol g\rangle\langle\boldsymbol g|\,, \tag{7b}$$

where $\bar\Gamma(g) \equiv \Gamma(\bar g)$, and generalizations thereof.

While all group-based $Z$ operators commute with one another (assuming the virtual state labels $\alpha$ and $\beta$ are held fixed), the group-based $X$ and $Z$ operators when acting on the same qudit have the non-trivial commutation relations

$$\overrightarrow{X}^{(h)} [Z^{(\Gamma)}]_{\alpha\beta} = [\Gamma(\bar h)\, Z^{(\Gamma)}]_{\alpha\beta}\, \overrightarrow{X}^{(h)}\,, \tag{8a}$$

$$\overleftarrow{X}^{(h)} [Z^{(\Gamma)}]_{\alpha\beta} = [Z^{(\Gamma)}\, \Gamma(h)]_{\alpha\beta}\, \overleftarrow{X}^{(h)}\,. \tag{8b}$$

The group-based $Z$ operators are diagonal in the $|\boldsymbol g\rangle$ basis and satisfy

$$[Z_j^{(\Gamma)}]_{\alpha\beta} |\boldsymbol g\rangle = [\Gamma(g_j)]_{\alpha\beta} |\boldsymbol g\rangle\,. \tag{9}$$

Another useful basis to consider is one whose single-particle states are labeled by the matrix elements of the irreps $\Gamma$: $\left|\Gamma_{\alpha\beta}\right\rangle$. These are related to the group element states $|g\rangle$ using a non-Abelian Fourier transform

$$\left|\Gamma_{\alpha\beta}\right\rangle = \sqrt{\frac{d_\Gamma}{|G|}} \sum_g [\Gamma(g)]_{\alpha\beta} |g\rangle\,, \tag{10a}$$

$$|g\rangle = \sum_\Gamma \sqrt{\frac{d_\Gamma}{|G|}} [\Gamma(\bar g)]_{\beta\alpha} \left|\Gamma_{\alpha\beta}\right\rangle\,, \tag{10b}$$

and span the same $|G|$-dimensional Hilbert space $\mathbb{C}[G]$ thanks to the great orthogonality identity

$$\frac{d_\Gamma}{|G|}\sum_g [\Gamma(\bar g)]_{\beta\alpha} [\Gamma'(g)]_{\alpha'\beta'} = \delta_{\Gamma,\Gamma'}\, \delta_{\alpha,\alpha'}\, \delta_{\beta,\beta'}\,. \tag{10c}$$

The group-based $X$ operators act on these local basis states as

$$\begin{aligned}\overrightarrow{X}^{(g)} \left|\Gamma_{\alpha\beta}\right\rangle &= [\Gamma(\bar g)]_{\alpha\gamma} \left|\Gamma_{\gamma\beta}\right\rangle\,, \\ \overleftarrow{X}^{(g)} \left|\Gamma_{\alpha\beta}\right\rangle &= [\Gamma(g)]_{\gamma\beta} \left|\Gamma_{\alpha\gamma}\right\rangle\,.\end{aligned} \tag{10d}$$

Therefore, a many-body $G$-qudit state can be specified by a choice of irrep $\Gamma_j$ and a matrix element $(\alpha_j, \beta_j)$ at every site. We denote such a collection of irreps and matrix elements as $\boldsymbol{\Gamma_{\alpha\beta}}$ and define

$$|\boldsymbol{\Gamma_{\alpha\beta}}\rangle = \bigotimes_j |\Gamma_{j;\alpha_j\beta_j}\rangle\,. \tag{11}$$

Since a $G$ qudit is simply a $|G|$-level quantum mechanical system, there is a correspondence between $G$ and $\widetilde{G}$-qudits whenever $|G| = |\widetilde{G}|$. This implies that there is always a mapping between such $G$-based Pauli operators and $|\widetilde{G}|$-based Pauli operators. For example, $G$-based Pauli operators can always be expressed as $\mathbb{Z}_{|G|}$-based clock and shift operators. However, such a relationship is generally complicated. In Appendix B, we demonstrate the relation between $D_{2n}$-qudits and $\mathbb{Z}_n \times \mathbb{Z}_2$ qudits by constructing the $D_{2n}$-based Pauli operators in terms of $\mathbb{Z}_n$ and $\mathbb{Z}_2$ clock and shift operators.

## 3 Group-based quantum XY model

It is fruitful to construct and explore Hamiltonian lattice models of $G$-qudits using their group-based Pauli operators [97–100]. Consider a $1 + 1$D quantum lattice model of $G$-qudits, defined on a periodic lattice of length $L$, with Hilbert space (1) and Hamiltonian

$$H_{XY} = \sum_j \left( \sum_\Gamma J_\Gamma \mathrm{Tr}\left( Z_j^{(\overline{\Gamma})} Z_{j+1}^{(\Gamma)} \right) + \sum_g K_g \overleftarrow{X}_j^{(g)} \overrightarrow{X}_{j+1}^{(g)} \right) + \mathrm{H.c.}, \tag{12}$$

where $J_\Gamma$ and $K_g$ are real coupling constants. This Hamiltonian is constructed from only two-body terms and depends on the real coupling constants $J_\Gamma$ and $K_g$. The Hamiltonian (12) generalizes the quantum XY model [25]. Indeed, when $G = \mathbb{Z}_2$, there are two one-dimensional irreps $\Gamma = \mathbf{1}, \mathbf{1}'$ in which the non-trivial element $g \in \mathbb{Z}_2$ is represented by 1 or $-1$, respectively. In this case, the group-based Pauli operators $\overrightarrow{X}_j^{(g)} = \overleftarrow{X}_j^{(g)}$ and $Z_j^{(\mathbf{1}')}$ reduce to the usual Pauli $\sigma^x$ and $\sigma^z$ operators. The non-trivial terms in Hamiltonian (12) then becomes

$$H_j^{(\mathbf{1}')} = \sigma_j^z \sigma_{j+1}^z, \qquad H_j^{(g)} = \sigma_j^x \sigma_{j+1}^x, \tag{13}$$

which is the XY model after the unitary transformation $(\sigma_j^x, \sigma_j^z) \to (\sigma_j^x, \sigma_j^y)$. Therefore, we will refer to the Hamiltonian (12) as $G$-based XY model.

For a general group $G$, the phase diagram of the $G$-based XY model (12) as a function of coupling constants $(J_\Gamma, K_g)$ is interesting and rich. We can analytically obtain its ground states at two fixed points. When $K_g = 0$ and $J_\Gamma = -d_\Gamma$, using Eq. (7), the Hamiltonian can be written as

$$H_{XY} = -2|G| \sum_{j,\mathbf{g}} \delta_{g_j, g_{j+1}} |\mathbf{g}\rangle \langle \mathbf{g}|, \tag{14}$$

which is a commuting projector Hamiltonian that has $|G|$ degenerate ground states $|\mathrm{GS}; g\rangle = \bigotimes_j |g\rangle$ labeled by the group elements $g$. Turning on $K_g \ll 1$ lifts the ground state degeneracy from $|G|$ to $|Z(G)|$.

In the opposite limit, when $J_\Gamma = 0$ and $K_g = -K$, the Hamiltonian is diagonalized in the $\left| \Gamma_{\alpha\beta} \right\rangle$ basis and becomes

$$H_{XY} = -K \sum_j \sum_{\Gamma,\alpha,\beta} \frac{|G|}{d_\Gamma} P_{j,j+1;\alpha,\beta}^{(\Gamma)}, \tag{15}$$

where we used Eq. (10c) to simplify the Hamiltonian. Here, $P_{j,j+1;\alpha,\beta}^{(\Gamma)}$ is a two-site projector that projects the Hilbert space $\mathcal{H}_j \otimes \mathcal{H}_{j+1}$ at site $j$ and $j+1$ onto the subspace spanned by the states $|\psi_{\alpha\beta}\rangle = \sum_\gamma \left| \Gamma_{\alpha,\gamma} \right\rangle \otimes \left| \Gamma_{\gamma,\beta} \right\rangle$. For this commuting projector Hamiltonian, the number of ground states equals the number of irreps, and they are given by $|\mathrm{GS}; \Gamma\rangle = \bigotimes_j \left| \Gamma_{\alpha_j, \alpha_{j+1}} \right\rangle$, where summation over repeated virtual indices is assumed.

### 3.1 $\mathrm{Rep}(G) \times Z(G) \times \mathbb{Z}_L$ symmetry operators

Analytically determining the phase diagram of Hamiltonian (12) is challenging away from the two fixed points. Nevertheless, we can still deduce aspects of the phase diagram by investigating the kinematic properties of the lattice model, namely its symmetries.

The Hamiltonian (12) has a $\mathbb{Z}_L$ lattice translation symmetry and various internal symmetries. For any operator $\mathcal{O}_j$ acting on $G$-qudits in a neighborhood of the site $j$, the translation symmetry can be generated by the operator

$$T: \mathcal{O}_j \to \mathcal{O}_{j+1}. \tag{16}$$

Among the many internal symmetries, in this paper, we will explore the symmetry generated by the operators[2]

$$U_z = \prod_{j=1}^{L} \overrightarrow{X}_j^{(z)},$$ (17a)

where $z \in Z(G)$, and

$$R_\Gamma = \mathrm{Tr}\left(\prod_{j=1}^{L} Z_j^{(\Gamma)}\right) \equiv \boxed{Z_1^{(\Gamma)}} - \boxed{Z_2^{(\Gamma)}} - \cdots - \boxed{Z_L^{(\Gamma)}}$$ (17b)

It is straightforward to confirm that these operators commute with the $G$-based XY model Hamiltonian. $U_z$ is a unitary operator that obeys the $Z(G)$ operator algebra

$$U_{z_1} \times U_{z_2} = U_{z_1 z_2}.$$ (18)

On the other hand, using properties of the characters $\chi_\Gamma$, the $R_\Gamma$ operators obey

$$R_{\Gamma_a} \times R_{\Gamma_b} = \sum_{\Gamma_c} N_{ab}^c R_{\Gamma_c},$$ (19)

where $N_{ab}^c \in \mathbb{Z}_{\geq 0}$ are the multiplicities in the tensor product of irreps $\Gamma_a \otimes \Gamma_b \simeq \bigoplus_c N_{ab}^c \Gamma_c$. Therefore, the symmetry operators $U_z$ and $R_\Gamma$ are $Z(G)$ and $\mathrm{Rep}(G)$ symmetry operators,[3] respectively.

When $G$ is Abelian, all irreps of $G$ are one-dimensional, and their fusion defines a group that is isomorphic to $G$ itself. In this case, the symmetry operators $R_\Gamma$ are unitary operators acting on the Hilbert space $\mathcal{H}$, forming a unitary representation of $G$. When $G$ is non-Abelian, however, the operator $R_\Gamma$ can have a non-trivial kernel spanned by states $|\boldsymbol{g}\rangle$ with $\chi_\Gamma(\prod_j g_j) = 0$, and is therefore not an invertible operator. Because for every $\Gamma$ with $d_\Gamma > 1$ there is at least one $g \in G$ with $\chi_\Gamma(g) = 0$ [101, Theorem 3.15], the $\mathrm{Rep}(G)$ symmetry is non-invertible when $G$ is non-Abelian.

In fact, the two previous limits of the $G$-based XY model, namely Eqs. (14) and (15), are characterized by these symmetries. Indeed, when $K_g \ll 1$ and $J_\Gamma = -d_\Gamma$, the $Z(G)$ symmetry is completely spontaneously broken, i.e., $Z(G) \xrightarrow{\mathrm{ssb}} 1$. And in the opposite limit of $J_\Gamma \ll 1$ and $K_g = -K$, the $\mathrm{Rep}(G)$ symmetry is spontaneously broken.

## 3.2 $\mathrm{Rep}(G) \times Z(G) \times \mathbb{Z}_L$ symmetry defects

Not all operators that commute with a lattice Hamiltonian are symmetry operators. For a conserved operator to be a symmetry operator, it must also have a corresponding symmetry defect. The existence of a symmetry defect restricts symmetry operators to be operators described by particular quantum operations [102].

While a symmetry operator implements the symmetry across time, a symmetry defect is a non-dynamical, localized modification to a Hamiltonian that implements the symmetry across space. Consequently, inserting a symmetry defect imposes a twisted periodic boundary condition. In one-dimensional space, the translation operator in the presence of a symmetry defect s satisfies

$$T^L = \prod_{j=1}^{L} U_j^{(\mathsf{s})},$$ (20)

---

[2]There is also an $\mathrm{Inn}(G) \simeq G/Z(G)$ symmetry represented by $A_g = \prod_j \overleftrightarrow{X}_j^{(g)}$, where $g \in G$, which does not play a role in our discussion.

[3]The symmetry operator algebra does not uniquely determine the fusion category describing the symmetry. Here, we call (17b) a $\mathrm{Rep}(G)$ symmetry operator since it obeys the correct operator algebra and, as shown in Appendix D.2, gauging the entire symmetry yields a dual $G$ symmetry.

where $U_j^{(\mathsf{s})}$ are unitary operators that move the symmetry defect. The product $\prod_{j=1}^{L} U_j^{(\mathsf{s})}$ is the $\mathsf{s}$ symmetry operator acting on the $\mathsf{s}$ defect Hilbert space. Because each $U_j^{(\mathsf{s})}$ is unitary, the location of the defect does not affect the spectrum of the Hamiltonian. Symmetry defects are, therefore, topological defects.

Having identified $\mathrm{Rep}(G) \times Z(G) \times \mathbb{Z}_L$ operators that commute with the $G$-based XY model, we now discuss their corresponding symmetry defects. In particular, we will independently discuss the symmetry defects of the $\mathrm{Rep}(G)$, $Z(G)$, and $\mathbb{Z}_L$ sub-symmetries one at a time. When discussing symmetry defects of internal symmetries, we will always restrict ourselves to discussing the symmetry defects described by simple objects in the corresponding fusion category. While we will specialize our discussion to the $G$-based quantum XY model, what we work out can be straightforwardly applied to other models with these sub-symmetries.

### 3.2.1  $Z(G)$ symmetry defects

We first review aspects of invertible symmetry defects by discussing the $Z(G)$ symmetry defects in the $G$-based XY model. Because it is an invertible symmetry and $z \times \bar{z} = 1$, we can create a $z$ symmetry defect and its orientation reversal—a $\bar{z}$ symmetry defect—using a local operator on the $G$-based XY model's Hilbert space (1). $z$-symmetry defects can be created in pairs this way because they are invertible and have quantum dimension 1.

We can create a $z$ defect at link $\langle I-1, I \rangle$ and $\bar{z}$ defect at $\langle J, J+1 \rangle$ using the truncated symmetry operator

$$U_z^{(I,J)} = \prod_{I \leq j \leq J} \overrightarrow{X}_j^{(z)} . \tag{21}$$

Acting $U_z^{(I,J)}$ onto a state creates a pair of $z$ and $\bar{z}$ defects. Similarly, conjugating the Hamiltonian $H_{XY}$ by $U_z^{(I,J)}$ yields the Hamiltonian with $z$ and $\bar{z}$ symmetry defects inserted:

$$H_{XY}^{(z_I, \bar{z}_J)} = H_{XY} + \sum_\Gamma J_\Gamma \Big[ \big( e^{-i\phi_\Gamma(z)} - 1 \big) \mathrm{Tr}\big( Z_{I-1}^{(\overline{\Gamma})} Z_I^{(\Gamma)} \big) + \big( e^{i\phi_\Gamma(z)} - 1 \big) \mathrm{Tr}\big( Z_J^{(\overline{\Gamma})} Z_{J+1}^{(\Gamma)} \big) + \mathrm{H.c.} \Big] . \tag{22}$$

The modified Hamiltonian $H_{XY}^{(z_I, \bar{z}_J)}$ is locally the same as the original $H_{XY}$ away from the links $\langle I-1, I \rangle$ and $\langle J, J+1 \rangle$ where the symmetry defects reside. This is a side effect of the locality properties of defects. Using this locality, we can deduce the Hamiltonian with a single $z$ defect inserted at $\langle I-1, I \rangle$:

$$H_{XY;z}^{\langle I-1, I \rangle} = H_{XY} + \sum_\Gamma J_\Gamma \Big[ \big( e^{-i\phi_\Gamma(z)} - 1 \big) \mathrm{Tr}\big( Z_{I-1}^{(\overline{\Gamma})} Z_I^{(\Gamma)} \big) + \mathrm{H.c.} \Big] . \tag{23}$$

After inserting the $z$ symmetry defect, the internal $\mathrm{Rep}(G) \times Z(G)$ symmetry operators still commute with the Hamiltonian. However, while the $\mathbb{Z}_L$ translation operator $T$ no longer does, the twisted translation operator

$$T_{\mathrm{tw}}^{(z,I)} = \overrightarrow{X}_I^{(z)} T , \tag{24}$$

does commute with (23). Therefore, in the presence of a $z$ symmetry defect, the system satisfies the twisted periodic boundary conditions

$$(T_{\mathrm{tw}}^{(z,I)})^L = U_z . \tag{25}$$

In other words, in the $z$ symmetry defect Hilbert space, the $Z(G)$ is non-trivially extended by the lattice translation group $\mathbb{Z}_L$.

Physically, translating by $T$ moves the defect from $\langle I-1, I \rangle$ to $\langle I, I+1 \rangle$ and then $\overrightarrow{X}_I^{(z)}$ moves it back from $\langle I, I+1 \rangle$ to $\langle I-1, I \rangle$. Indeed, the defect Hamiltonian (23) satisfies

$$H_{XY;z}^{\langle I, I+1 \rangle} = \overrightarrow{X}_I^{(\bar{z})} H_{XY;z}^{\langle I-1, I \rangle} \overrightarrow{X}_I^{(z)} . \tag{26}$$

Therefore, the symmetry defect is a topological defect in the sense that its location does not affect the spectrum of the defect Hamiltonian.

### 3.2.2 Rep($G$) symmetry defects

We next discuss inserting Rep($G$) symmetry defects into the system, which are labeled by the irreps $\Gamma$ of $G$. Because the quantum dimension of a $\Gamma$ symmetry defect is $d_\Gamma$, the Hilbert space after inserting a single $\Gamma$ defect must be $d_\Gamma$ times larger than the original defect-free Hilbert space

$$\mathcal{H} = \bigotimes_{j=1}^{L} \mathbb{C}^{|G|}. \tag{27}$$

Consequently, the procedure for inserting a Rep($G$) defect must be generalized from the method used for the invertible $Z(G)$ symmetry defect. Here we specialize to Rep($G$) in tensor-product $G$ qudit models, but our methodology can be applied to any symmetry in $1+1$D represented by an MPO whose MPO tensors are unitary. Furthermore, we refer the reader to Refs. 88 and 103 for discussion of symmetry defects (invertible and non-invertible) and their insertions using the categorical formalism for general anyon chain models.

Consider the truncated Rep($G$) symmetry operators

$$[R_\Gamma^{(I,J)}]_{\alpha\beta} = \Big[ \prod_{I \le j \le J} Z_j^{(\Gamma)} \Big]_{\alpha\beta} \equiv \alpha \,\blacksquare\!-\!\boxed{Z_I^{(\Gamma)}}\!-\!\boxed{Z_{I+1}^{(\Gamma)}}\!-\cdots-\!\boxed{Z_{J-1}^{(\Gamma)}}\!-\!\boxed{Z_J^{(\Gamma)}}\!-\!\blacksquare\, \beta \ . \tag{28}$$

This operator maps states from $\mathcal{H}$ to states in $\mathcal{H}$ and does not create Rep($G$) symmetry defects. To create a pair of Rep($G$) symmetry defects, we modify $[R_\Gamma^{(I,J)}]_{\alpha\beta}$ by bending the horizontal, virtual legs upwards such that their corresponding virtual Hilbert spaces are promoted to physical, defect Hilbert spaces. Doing so yields

$$R_\Gamma^{(I,J)} \equiv \boxed{Z_I^{(\Gamma)}}\!-\!\boxed{Z_{I+1}^{(\Gamma)}}\!-\cdots-\!\boxed{Z_{J-1}^{(\Gamma)}}\!-\!\boxed{Z_J^{(\Gamma)}} = \sum_{\alpha,\beta=1}^{d_\Gamma} |\alpha\rangle_L \otimes |\beta\rangle_R \otimes [R_\Gamma^{(I,J)}]_{\alpha\beta}, \tag{29}$$

which maps states from $\mathcal{H}$ to the defect Hilbert space[4]

$$\mathcal{H}_{\Gamma\otimes\Gamma^*} \cong \mathbb{C}_L^{d_\Gamma} \otimes \mathbb{C}_R^{d_\Gamma} \otimes \mathcal{H}, \tag{30}$$

with $\Gamma$ and $\Gamma^*$ symmetry defects inserted at links $\langle I-1, I\rangle$ and $\langle J, J+1\rangle$, respectively. The $\Gamma^*$ symmetry defect is the orientation reversal of the $\Gamma$ symmetry defect, and as an irrep of $G$, $\Gamma^*(g)$ is the complex conjugation of $\Gamma(g)$ (i.e., the dual irrep of $\Gamma$) since $R_\Gamma^\dagger = R_{\Gamma^*}$. Physically, each $\mathbb{C}^{d_\Gamma}$ describes the internal degrees of freedom of a Rep($G$) defect. Furthermore, while $R_\Gamma^{(I,J)}$ creates this symmetry defect pair, $[R_\Gamma^{(I,J)}]^\dagger$ annihilates it, and these two operators obey

$$[R_\Gamma^{(I,J)}]^\dagger R_\Gamma^{(I,J)} = d_\Gamma \mathbf{1}_\mathcal{H}, \tag{31}$$

---

[4]More precisely, this defect Hilbert space is

$$\mathcal{H}_{\Gamma\otimes\Gamma^*} = \bigotimes_{j=1}^{I-1} \mathbb{C}^{|G|} \otimes \mathbb{C}_L^{d_\Gamma} \otimes \bigotimes_{j=I}^{J} \mathbb{C}^{|G|} \otimes \mathbb{C}_R^{d_\Gamma} \otimes \bigotimes_{j=J+1}^{L} \mathbb{C}^{|G|},$$

which is isomorphic to (30). Since $\mathcal{H}$ is a tensor product Hilbert space, the location of $\mathbb{C}^{d_\Gamma}$ in the tensor product does not matter. This is not true, however, for non-tensor product Hilbert spaces (see Ref. 88).

where $\mathbf{1}_{\mathcal{H}}$ is the identity operator for the defect-free Hilbert space (27). The factor of $d_\Gamma$ on the right-hand side is consistent with the relativistic QFT result where the correlation function of a contractible topological defect line equals its quantum dimension.

The defect Hamiltonian $H_{XY}^{(\Gamma_I, \Gamma_J^*)}$ for the group-based XY model $H_{XY}$ with $\Gamma$ and $\Gamma^*$ defects inserted at links $\langle I-1, I\rangle$ and $\langle J, J+1\rangle$ is defined as

$$R_\Gamma^{(I,J)} H_{XY} = H_{XY}^{(\Gamma_I, \Gamma_J^*)} R_\Gamma^{(I,J)}. \tag{32}$$

Using this and denoting by

$$\widehat{\Gamma}(g) = \sum_{\alpha,\beta=1}^{d_\Gamma} [\Gamma(g)]_{\alpha\beta} \, |\alpha\rangle\langle\beta| \,, \tag{33}$$

we find

$$\begin{aligned}
H_{XY}^{(\Gamma_I, \Gamma_J^*)} = \mathbf{1}_{d_\Gamma^2} \otimes H_{XY} &+ \sum_g K_g \left[ \left( \widehat{\Gamma}(g) \otimes \mathbf{1}_{d_\Gamma} - \mathbf{1}_{d_\Gamma^2} \right) \otimes \overleftarrow{X}_{I-1}^{(g)} \overrightarrow{X}_I^{(g)} + \text{H.c.} \right] \\
&+ \sum_g K_g \left[ \left( \mathbf{1}_{d_\Gamma} \otimes \widehat{\Gamma}^*(g) - \mathbf{1}_{d_\Gamma^2} \right) \otimes \overleftarrow{X}_J^{(g)} \overrightarrow{X}_{J+1}^{(g)} + \text{H.c.} \right],
\end{aligned} \tag{34}$$

where $\mathbf{1}_n$ is the $n \times n$ identity matrix. Like for the $Z(G)$ defect, this defect Hamiltonian is modified only near the symmetry defects, and using this locality, we can deduce the defect Hamiltonian for a single $\Gamma$ symmetry defect. In particular, the defect Hamiltonian for a $\Gamma$ symmetry defect at link $\langle I-1, I\rangle$ is

$$H_{XY;\Gamma}^{\langle I-1,I\rangle} = \mathbf{1}_{d_\Gamma} \otimes H_{XY} + \sum_g K_g \left[ \left( \widehat{\Gamma}(g) - \mathbf{1}_{d_\Gamma} \right) \otimes \overleftarrow{X}_{I-1}^{(g)} \overrightarrow{X}_I^{(g)} + \text{H.c.} \right], \tag{35}$$

and the corresponding defect Hilbert space is isomorphic to

$$\mathcal{H}_\Gamma \cong \mathbb{C}^{d_\Gamma} \otimes \mathcal{H}. \tag{36}$$

After inserting the $\Gamma$ symmetry defect, the internal $\text{Rep}(G) \times Z(G)$ symmetry operators $\mathbf{1}_{d_\Gamma} \otimes R_\Gamma$ and $\mathbf{1}_{d_\Gamma} \otimes U_z$ still commute with the Hamiltonian. However, while the $\mathbb{Z}_L$ translation operator $\mathbf{1}_{d_\Gamma} \otimes T$ no longer does, the twisted translation operator

$$T_{\text{tw}}^{(\Gamma, I)} = \widehat{Z}_I^{(\Gamma)} (\mathbf{1}_{d_\Gamma} \otimes T), \tag{37}$$

where

$$\widehat{Z}_j^{(\Gamma)} = \sum_{\alpha,\beta=1}^{d_\Gamma} [Z_j^{(\Gamma)}]_{\alpha\beta} \otimes |\alpha\rangle\langle\beta| \,. \tag{38}$$

does commute with (35). Therefore, in the presence of a $\Gamma$ symmetry defect, the system satisfies the twisted periodic boundary conditions[5]

$$(T_{\text{tw}}^{(z,I)})^L = \prod_{j=I}^{L+I-1} \widehat{Z}_j^{(\Gamma)}. \tag{39}$$

Physically, translating by $T$ moves the defect from $\langle I-1, I\rangle$ to $\langle I, I+1\rangle$ and then $\widehat{Z}_I^{(\Gamma)}$ moves it back from $\langle I, I+1\rangle$ to $\langle I-1, I\rangle$. Indeed, the defect Hamiltonian (35) satisfies

$$H_{XY;\Gamma}^{\langle I,I+1\rangle} = \widehat{Z}_I^{(\Gamma)\dagger} H_{XY;\Gamma}^{\langle I-1,I\rangle} \widehat{Z}_I^{(\Gamma)}. \tag{40}$$

Therefore, the symmetry defect is a topological defect in the sense that its location does not affect the spectrum of the defect Hamiltonian.

---

[5]After inserting a $\Gamma$ symmetry defect at $\langle I-1, I\rangle$, the potentially non-invertible $R_\Gamma$ symmetry operator becomes the unitary operator $\prod_{j=I}^{L+I-1} \widehat{Z}_j^{(\Gamma)}$, which acts on the $\Gamma$ defect Hilbert space $\mathcal{H}_\Gamma$. When the defect degrees of freedom are traced out, this unitary operator becomes $\sum_\alpha \langle\alpha| \prod_{j=I}^{L+I-1} \widehat{Z}_j^{(\Gamma)} |\alpha\rangle = R_\Gamma$.

### 3.2.3 Translation symmetry defects

Having discussed the symmetry defects for the internal $Z(G)$ and Rep$(G)$ symmetries, we now review aspects of symmetry defects for the $\mathbb{Z}_L$ lattice translation symmetry. Because translations are a crystalline symmetry, their symmetry defects are qualitatively different than internal symmetry defects [21].

A translation defect is inserted by removing/adding a lattice site [22, 56, 93]. Following Ref. 21, one way to motivate this definition of a translation symmetry defect is to note that after inserting an invertible symmetry defect whose symmetry operator is $U$, the system satisfies twisted boundary conditions $T^L = U$. Therefore, inserting a translation defect is equivalent to enforcing the twisted boundary condition $T^L = T$, or equivalently

$$T^{L-1} = 1 \,. \tag{41}$$

This is just ordinary periodic boundary conditions for a length $L-1$ chain; hence, inserting a $T$ symmetry defect is implemented by removing a lattice site.

## 4 Projective algebra from symmetry defects and its consequences

In the previous section, we introduced the $G$-based XY model (12) and discussed its internal Rep$(G) \times Z(G)$ symmetry, its translation symmetry, and their corresponding symmetry defects. In this section, we discuss the physical consequences of these symmetries. In particular, we show that there is a projective algebra arising from the Rep$(G) \times Z(G) \times \mathbb{Z}_L$ symmetry that constrains the allowed phases in symmetric models. We will show that when $Z(G)$ is represented non-trivially in at least one one-dimensional irrep of $G$ (i.e., when $Z(G) \not\subset [G, G]$),[6] the projective algebra leads to an LSM anomaly precluding the emergence of SPT phases and enforcing long-ranged entanglement in the ground state. In contrast, when this is not the case (i.e., when $Z(G) \subset [G, G]$), the projective algebra does not give rise to an LSM anomaly but instead an SPT-LSM constraint that enforces non-trivial entanglement in the ground state, either short-ranged or long-ranged. The allowed SPT phases are necessarily non-invertible weak SPTs, for which translation defects are dressed by Rep$(G)$ symmetry charges.

### 4.1 Projective Rep$(G) \times Z(G) \times \mathbb{Z}_L$ algebra

To begin our discussion, let us return to the $G$-based XY model without any Rep$(G)$ or $Z(G)$ symmetry defects inserted. For general system size $L$, the Rep$(G)$ and $Z(G)$ symmetry operators $R_\Gamma$ and $U_z$, respectively, do not commute but instead satisfy the projective algebra

$$R_\Gamma U_z = \mathrm{e}^{\mathrm{i}\phi_\Gamma(z)L} U_z R_\Gamma \,, \tag{42}$$

where $\mathrm{e}^{\mathrm{i}\phi_\Gamma(z)} = \chi_\Gamma(z)/d_\Gamma$ is a phase because $z \in Z(G)$. Therefore, when the number of lattice sites satisfies

$$L = L^* \in \mathrm{lcm}(\{|z|, z \in Z(G)\}) \mathbb{Z}_{\geq 0} \,, \tag{43}$$

---

[6]All one-dimensional irreps of a finite group $G$ are trivial exactly in its commutator subgroup $[G, G] = \{g^{-1}h^{-1}gh \mid \forall g, h \in G\}$. Indeed, since each element of $[G, G]$ is a commutator, it is represented trivially in all one-dimensional irreps $\Gamma_1$: $\Gamma_1(g^{-1}h^{-1}gh) = \mathrm{e}^{-\mathrm{i}\theta_1(g)}\mathrm{e}^{-\mathrm{i}\theta_1(h)}\mathrm{e}^{\mathrm{i}\theta_1(g)}\mathrm{e}^{\mathrm{i}\theta_1(h)} = 1$. Furthermore, since the quotient group $G/[G, G]$ is Abelian, it must always has a faithful irrep [101], so any group element $g \notin [G, G]$ must be represented non-trivially in at least one one-dimensional irrep of $G$. Therefore, all elements of $Z(G)$ are trivially represented in one-dimensional irreps if and only if $Z(G) \subset [G, G]$.

with lcm denoting the least common multiple and $|z|$ the order of $z$, these two symmetry operators commute. Otherwise, they fail to commute by a phase. The projective phase in (42) arises from the local projective algebra of $\overrightarrow{X}_j^{(z)}$ and the MPO $Z_j^{(\Gamma)}$:

$$Z_j^{(\Gamma)} \overrightarrow{X}_j^{(z)} = \mathrm{e}^{\mathrm{i}\phi_\Gamma(z)} \overrightarrow{X}_j^{(z)} Z_j^{(\Gamma)}. \tag{44}$$

The projective algebra (42) has an interesting interpretation in terms of translation defects (see Section 3.2.3). Let $L = L^*$ denote a system size in which there are no translation defects inserted. Inserting a translation defect then causes $L$ to change from $L^*$ to $L^* + 1$, after which the $\mathrm{Rep}(G)$ and $Z(G)$ symmetry operators form a non-trivial projective algebra. Therefore, Eq. (42) can be interpreted as the $\mathrm{Rep}(G) \times Z(G)$ symmetry being projectively realized in the translation-defect Hilbert spaces.

What happens after inserting the $Z(G)$ or $\mathrm{Rep}(G)$ symmetry defects discussed in Sections 3.2.1 and 3.2.2, respectively? After inserting a $z \in Z(G)$ symmetry defect at link $\langle L, 1 \rangle$, the translation operator is modified to $T_{\mathrm{tw}}^{(z)} = \overrightarrow{X}_1^{(z)} T$, which commutes with $U_z$ but does not generally commute with $R_\Gamma$:

$$R_\Gamma T_{\mathrm{tw}}^{(z)} = \mathrm{e}^{\mathrm{i}\phi_\Gamma(z)} T_{\mathrm{tw}}^{(z)} R_\Gamma. \tag{45}$$

On the other hand, inserting a $\Gamma \in \mathrm{Rep}(G)$ symmetry defect causes the translation operator to become $T_{\mathrm{tw}}^{(\Gamma)} = \widehat{Z}_1^{(\Gamma)} (\mathbf{1}_{d_\Gamma} \otimes T)$, which satisfies

$$T_{\mathrm{tw}}^{(\Gamma)} U_z = \mathrm{e}^{\mathrm{i}\phi_\Gamma(z)} U_z T_{\mathrm{tw}}^{(\Gamma)}. \tag{46}$$

Therefore, inserting a $Z(G)$ ($\mathrm{Rep}(G)$) symmetry defect causes the $\mathrm{Rep}(G)$ ($Z(G)$) symmetry and translations to form a non-trivial projective algebra. We summarize the various projective algebras arising from symmetry defects in Table 1.

The projective phases in Eqs. (42), (45), and (46) cannot be redefined away by any redefinition of symmetry operators by phase factors. This is because, in all these equations, the same operators appear on both sides of the equations, which causes any phase redefinitions to be canceled. Therefore, this projective phase is an invariant property of the $\mathrm{Rep}(G) \times Z(G) \times \mathbb{Z}_L$ symmetry and has physical consequences. In the remainder of this section, we will explore the different consequences of the projective phases in Eqs. (42), (45), and (46).

Before moving on to the physical consequences, we finish this subsection by contextualizing the projective algebras in an example we will return to several times. Namely, for the case where $G$ is the dihedral group of order 8,

$$D_8 \simeq \mathbb{Z}_2 \ltimes \mathbb{Z}_4 = \left\langle r, s \mid r^2 = s^4 = 1, \ r s r = s^3 \right\rangle, \tag{47}$$

which, along with the quaternion group $Q_8$, is the smallest non-Abelian finite group with a non-trivial center. The properties of $D_8$ and its irreps are reviewed in Appendix A. It has a non-trivial $Z(D_8) = \mathbb{Z}_2$ center generated by the element $s^2$. The group $D_8$ has four one-dimensional irreps, three of which are non-trivial and denoted by $\mathsf{P}_i$ ($i = 1, 2, 3$), and one two-dimensional irrep that we denote by $\mathsf{E}$. The corresponding $\mathrm{Rep}(D_8)$ symmetry operators satisfy the operator algebra

$$\begin{aligned} R_{\mathsf{P}_i} R_{\mathsf{P}_i} &= 1, & R_{\mathsf{P}_1} R_{\mathsf{P}_2} &= R_{\mathsf{P}_3}, \\ R_{\mathsf{P}_i} R_{\mathsf{E}} &= R_{\mathsf{E}} R_{\mathsf{P}_i} = R_{\mathsf{E}}, & R_{\mathsf{E}} R_{\mathsf{E}} &= 1 + \sum_i R_{\mathsf{P}_i}. \end{aligned} \tag{48}$$

The $\mathbb{Z}_2$ center is trivially represented in all one-dimensional irreps. Therefore, the only non-trivial projective phase is due to the two-dimensional irrep $\mathsf{E}$ with $\phi_{\mathsf{E}}(s^2) = \pi$, and the non-trivial instances of Eqs. (42), (45), and (46) are

$$R_{\mathsf{E}} U_{s^2} = -U_{s^2} R_{\mathsf{E}}, \qquad R_{\mathsf{E}} T_{\mathrm{tw}}^{(s^2)} = -T_{\mathrm{tw}}^{(s^2)} R_{\mathsf{E}}, \qquad T_{\mathrm{tw}}^{(\mathsf{E})} U_{s^2} = -U_{s^2} T_{\mathrm{tw}}^{(\mathsf{E})}, \tag{49}$$

respectively.

## 4.2 LSM anomalies

As we have shown, inserting defects of one sub-symmetry leads to a projective algebra for the other two. This is reminiscent of how mixed anomalies often manifest in $1 + 1$D and suggests that there could be an LSM anomaly between $\text{Rep}(G)$, $Z(G)$, and translation symmetries. Indeed, when $G$ is an Abelian group $A$, then $\text{Rep}(A) \cong Z(A) \cong A$, and the projective algebra, in fact, indicates an LSM anomaly (see Refs. [4, 21, 32] for the case of $G = \mathbb{Z}_2$). In what follows, we show that an LSM anomaly can also occur even when $G$ is non-Abelian. To do so, we assume that $Z(G)$ is non-trivial and consider the local Hamiltonian

$$H = \sum_{\alpha=1}^{L} T^\alpha O_1 T^{-\alpha} , \tag{50}$$

where $O_1$ is a $\text{Rep}(G) \times Z(G)$-symmetric operator localized around site $j = 1$. By construction, this Hamiltonian is invariant under the $\text{Rep}(G) \times Z(G) \times \mathbb{Z}_L$ symmetry. We assume that $L$ is chosen such that Eq. (43) holds, and further assume that the Hamiltonian $H$ has a non-degenerate and gapped ground state $|\psi\rangle$. To prove the LSM anomaly, we will show that the latter assumption leads to contradiction and, therefore, cannot be true.

Following Section 3.2.1, we now insert a $Z(G)$ symmetry defect corresponding to the group element $z \in Z(G)$, which yields the $\text{Rep}(G) \times Z(G)$ symmetric Hamiltonian

$$H_{\text{tw}} = \sum_{\alpha=1}^{L} (T_{\text{tw}}^{(z)})^\alpha O_1 (T_{\text{tw}}^{(z)})^{-\alpha} , \tag{51}$$

where $T_{\text{tw}}^{(z)} := \overrightarrow{X}_1^{(z)} T$ is the $z$-twisted translation operator that satisfies $[T_{\text{tw}}^{(z)}]^L = U_z$ and the projective algebra (45). Since $O_1$ is a local operator with compact support, Hamiltonians $H_{\text{tw}}$ and $H$ differ only by a finite number of local terms with support on sites around the link $\langle L, 1 \rangle$. Following the robustness of non-degenerate and gapped ground states to invertible symmetry-twisted boundary conditions [34, 104], $H_{\text{tw}}$ is expected to have a non-degenerate and gapped ground state $|\psi_{\text{tw}}\rangle$ since, by assumption, $H$ has such a ground state $|\psi\rangle$. However, when $z$ is represented non-trivially on a one-dimensional irrep $\Gamma_1$, i.e., the phase $\phi_{\Gamma_1}(z)$ in Eq. (45) is non-vanishing, $H_{\text{tw}}$ cannot have a non-degenerate and gapped ground state. This is because, on the one hand, the corresponding $R_{\Gamma_1}$ is a unitary operator, so the non-degenerate ground state must satisfy

$$R_{\Gamma_1} |\psi_{\text{tw}}\rangle = e^{i\theta_{\Gamma_1}} |\psi_{\text{tw}}\rangle , \tag{52}$$

while on the other hand, the projective algebra (45) requires the expectation value

$$\langle \psi_{\text{tw}} | R_{\Gamma_1} | \psi_{\text{tw}} \rangle = \langle \psi_{\text{tw}} | U_z^\dagger R_{\Gamma_1} U_z | \psi_{\text{tw}} \rangle = e^{i\phi_{\Gamma_1}(z)} \langle \psi_{\text{tw}} | R_{\Gamma_1} | \psi_{\text{tw}} \rangle , \tag{53}$$

to then be vanishing. Hence, all eigenstates of $H_{\text{tw}}$, including the ground state, must be degenerate for such a $z$, which contradicts our initial assumption that the untwisted Hamiltonian $H$ has a non-degenerate and gapped ground state $|\psi\rangle$. Note that the phase $\phi_{\Gamma_1(z)}$ is non-vanishing whenever $Z(G) \not\subset [G, G]$ since there is then always a $z \notin [G, G]$ represented non-trivially in a one-dimensional irrep $\Gamma_1$ (see Footnote 6). Therefore, when $Z(G) \not\subset [G, G]$, there is an LSM anomaly preventing an SPT phase due to the center $Z(G)$ and the invertible part of the $\text{Rep}(G)$ symmetry. This is a generalization of the LSM theorems proved in Refs. [34, 44, 50] using twisted boundary conditions.

The above proof applies to groups $G$ for which $Z(G) \not\subset [G, G]$. Let us now consider the case for which $Z(G) \subset [G, G]$ such that $Z(G)$ is represented trivially in all one-dimensional irreps. For example, the group $D_8$ satisfies this. In this case, the only non-trivial phases in Eq. (45)

are ones that originate from a higher-dimensional irrep. These phases, however, do not imply degeneracy since for a two or higher-dimensional irrep, the operator $R_\Gamma$ is non-invertible. In particular, the projective algebra Eq. (45) can be satisfied by a non-degenerate eigenstate that is annihilated by the non-invertible symmetry $R_\Gamma$.

Alternatively, when $Z(G) \subset [G, G]$, one may try inserting a $\mathrm{Rep}(G)$ symmetry defect, choosing an irrep $\Gamma$ with dimension greater than one, to see if there is an LSM anomaly. In this case, the projective algebra (46) involves only unitary operators $U_z$ and $T_{\mathrm{tw}}^{(\Gamma)}$. While this projective algebra implies degeneracy, we can not immediately use it to rule out a non-degenerate gapped ground state for the untwisted Hamiltonian. This is because inserting a *non-invertible* $\Gamma$ symmetry defect enlarges the global Hilbert space $\mathcal{H}$ to $\mathbb{C}^{d_\Gamma} \otimes \mathcal{H}$ (see Section 3.2.2). Hence, the degeneracy due to the algebra (46) might be accounted for by the $d_\Gamma$-dimensional defect Hilbert space (i.e., the degeneracy due to the projective algebra (46) matches degeneracy from the $d_\Gamma$-dimensional Hilbert space) and would not have implications to the defect-free theory. This is, in fact, always the case when $Z(G) \subset [G, G]$. To see this, let $k$ be the smallest positive integer such that

$$e^{ik\,\phi_\Gamma(z)} = 1 . \tag{54}$$

Since $\phi_\Gamma(z)$ is the phase appearing in the projective algebra, the above naively implies a $k$-fold degeneracy. However, recall that for any $d_\Gamma > 1$ irrep $\Gamma$, the determinant $\det \Gamma$ of the irrep defines a one-dimensional representation. Therefore, because we assume $Z(G)$ is represented trivially in all one-dimensional irreps,[7]

$$\det(\Gamma(z)) = e^{id_\Gamma\,\phi_\Gamma(z)} = 1 . \tag{55}$$

Hence, the integer $k$ always divides $d_\Gamma$ and the degeneracy due to the projective algebra (46) can always be accounted for by the dimension $d_\Gamma$ of the defect Hilbert space.

Therefore, while the projective algebra implies an LSM anomaly when $Z(G) \not\subset [G, G]$, it does not when $Z(G) \subset [G, G]$.

## 4.3 Implications on entanglement

We have shown that when the center $Z(G)$ is represented trivially on all one-dimensional irreps, the projective algebras (45) and (46) do not imply an LSM anomaly. However, as we next show, the projective algebras still always constrain the entanglement structure of the ground state. More precisely, a non-degenerate ground state cannot be a product state and, thus, must carry non-zero entanglement. To show this, we next identify contradictions that arise when we assume that the non-degenerate ground state of an arbitrary local symmetric Hamiltonian is the product state

$$|\mathrm{GS}\rangle = \bigotimes_j |\psi_j\rangle , \qquad |\psi_j\rangle = \sum_{g_j \in G} c_{j,g_j} |g_j\rangle . \tag{56}$$

We note that if such a ground state exists, then by the assumption that the Hamiltonian is local, the ground state for any system size $L$ is the product state of the same local state $|\psi_j\rangle$.

First, let us collect some useful identities based on our assumptions. Because $|\mathrm{GS}\rangle$ is a product state, it must be translational invariant and satisfy [35]

$$T |\mathrm{GS}\rangle = |\mathrm{GS}\rangle , \tag{57}$$

which constraints the coefficients $c_{j,g}$ to be independent of $j$ (i.e., $c_{j,g} \equiv c_g$). Similarly, since we assume that it is a non-degenerate ground state, it must be $Z(G)$-symmetric and satisfy

$$U_z |\mathrm{GS}\rangle = e^{i\varphi_z L} |\mathrm{GS}\rangle , \tag{58}$$

---

[7]We thank *mathoverflow* user *SashaP* for pointing out this fact [105].

which implies $c_{\bar{z}g} = e^{i\varphi_z} c_g$. Since the operator $U_z$ is a product of local operators and the ground state $|\mathrm{GS}\rangle$ is a product state, a local version of Eq. (58) also holds:

$$\overrightarrow{X}_j^{(z)} |\mathrm{GS}\rangle = e^{i\varphi_z} |\mathrm{GS}\rangle \, . \tag{59}$$

Finally, we note that under the $\mathrm{Rep}(G)$ symmetry, the state $|\mathrm{GS}\rangle$ transforms as

$$\begin{aligned}
R_\Gamma |\mathrm{GS}\rangle &= \sum_{\{g\}} \chi_\Gamma \Big(\prod_{j=1}^{L} g_j\Big) \Big(\prod_{j=1}^{L} c_{g_j}\Big) |g_1, g_2, \cdots, g_L\rangle \\
&\equiv \lambda_\Gamma^{(L)} |\mathrm{GS}\rangle \, ,
\end{aligned} \tag{60}$$

where $\lambda_\Gamma^{(L)}$ is the eigenvalue of the ground state under $R_\Gamma$ and the last line follows from the fact that by assumption the ground state is non-degenerate.

Because of these identities, for any system size $L$, we have on the one hand

$$R_\Gamma \overrightarrow{X}_j^{(z)} |\mathrm{GS}\rangle = e^{i\varphi_z} \lambda_\Gamma^{(L)} |\mathrm{GS}\rangle \, , \tag{61a}$$

while on the other hand using projective algebra (42) and Eq. (60) gives

$$R_\Gamma \overrightarrow{X}_j^{(z)} |\mathrm{GS}\rangle = e^{i\phi_\Gamma(z)} \overrightarrow{X}_j^{(z)} R_\Gamma |\mathrm{GS}\rangle = e^{i\phi_\Gamma(z)} \lambda_\Gamma^{(L)} e^{i\varphi_z} |\mathrm{GS}\rangle \, . \tag{61b}$$

Since by Schur's lemma, there always exists an irrep $\Gamma$ of $G$ and $z \in Z(G)$ such that $e^{i\phi_\Gamma(z)} \neq 1$, these two equations are compatible only if $\lambda_\Gamma^{(L)} = 0$ for any system size $L$, i.e., if the product state (56) is annihilated by $R_\Gamma$ for any $L$.

However, as we shall show this cannot be correct. To see this, without a loss of generality, let us choose a particular group element $g \in G$ for which the coefficient $c_g$ is non-vanishing. This means that if we expand the ground state $|\mathrm{GS}\rangle$ in the $|g\rangle$ basis, the state $\otimes_{j=1}^{L} |g\rangle$ appears with non-vanishing coefficient $(c_g)^L$. This basis state transforms under the $\mathrm{Rep}(G)$ symmetry operator $R_\Gamma$ by the character $\chi_\Gamma(g^L)$. Now, since the ground state for any system size $L$ is the product state of the same local state $|\psi_j\rangle$, we can choose a size $L^*$ such that $g^{L^*} = 1$. At this system size, one has

$$\lambda_\Gamma^{(L^*)} = \chi_\Gamma\big(g^{L^*}\big) = d_\Gamma \, , \tag{62}$$

which contradicts with the statement that $R_\Gamma$ must annihilate the ground state $|\mathrm{GS}\rangle$ at any system size $L$ since $d_\Gamma \neq 0$. Therefore, the ground state cannot be a product state as we assumed in the beginning.

While we used the $G$-qudit representation of $\mathrm{Rep}(G)$ symmetry to reach Eq. (62), it is expected that this equation holds for some system size $L^*$ independent of the microscopic details. This is because we expect any local lattice Hamiltonian with a non-degenerate gapped ground state to have an IR description in terms of a topological quantum field theory (TQFT) when an appropriate thermodynamic limit is taken. In TQFTs with $\mathrm{Rep}(G)$-symmetric vacua, the vacuum expectation value of $\mathrm{Rep}(G)$ symmetry operators must match their quantum dimensions [73]. However, if there is no system size for which Eq. (62) applies, then there is no thermodynamic limit for which IR theory is described by a TQFT which leads to a contradiction.

We conclude that there is no local Hamiltonian whose non-degenerate and gapped ground state is a simple product state of the form (56). This is to say that even when the center $Z(G)$ is represented trivially on all one-dimensional irreps and, hence, there is no LSM anomaly, the projective algebras (42), (45), and (46) still has non-trivial implications on low-energy properties. In particular, the ground state must carry entanglement and cannot be a trivial product state with no entanglement. This is reminiscent of the SPT-LSM theorems for invertible

symmetries [63,64], which, while not ruling out a short-range entangled ground state, requires any symmetric SPT state to be non-trivial and support degenerate boundary states. From that point of view, projective algebra involving $\text{Rep}(G) \times Z(G) \times \mathbb{Z}_L$ either leads to usual LSM anomalies, which necessarily require long-range entanglement, or SPT-LSM constraints that require any SPT state to be distinct from a trivial product state.

## 4.4 Entangled non-invertible weak SPTs

The projective algebras (42) and (45) do not preclude a symmetric SPT ground state if the center is trivially represented in all one-dimensional irreps. This leads to the new possibility of a $\text{Rep}(G) \times Z(G) \times \mathbb{Z}_L$ invariant SPT state. What are the properties of such an SPT state?

As discussed in Section 4.3, a $\text{Rep}(G) \times Z(G) \times \mathbb{Z}_L$-symmetric SPT Hamiltonian cannot have a trivial product state as its non-degenerate ground state. Furthermore, the projective algebra (42) requires that depending on the system size $L$, such an SPT ground state must be annihilated by all non-invertible symmetries $R_\Gamma$ for which the projective phase $e^{i\phi_\Gamma(z)L}$ is non-trivial.[8] From a decorated domain wall perspective of SPTs [106–108], we interpret this as translation defects carrying non-trivial $\text{Rep}(G)$ charges since at $L = L^*$ for $L^*$ in (43), we expect all $R_\Gamma$ acting on the ground state as $d_\Gamma$, but for $L = L^* + 1$ some $R_\Gamma$ annihilate the ground state. By breaking the translation symmetry explicitly, it is possible to choose an enlarged unit cell such that the ground state is a trivial product state that is not annihilated for any system size. The non-degenerate ground state of $\text{Rep}(G) \times Z(G) \times \mathbb{Z}_L$-symmetric Hamiltonian then can be thought of as a weak SPT state protected by both internal $\text{Rep}(G) \times Z(G)$ and $\mathbb{Z}_L$ translation symmetries. However, as opposed to certain weak SPT states with only invertible symmetries,[9] such a non-invertible weak SPT state cannot be a product state. For this reason, we will call the non-degenerate and gapped ground states of $\text{Rep}(G) \times Z(G) \times \mathbb{Z}_L$-symmetric Hamiltonians *entangled non-invertible weak SPTs*.[10]

### 4.4.1 Example $G = D_8$

Let us give a concrete example of such an entangled weak SPT Hamiltonian and study its ground state properties. We consider the group $G = D_8$ (see Appendix A for a review) for which an SPT Hamiltonian is given by[11]

$$H_{\text{SPT}} = -\sum_j \overleftarrow{X}_j^{(r)} \overrightarrow{X}_{j+1}^{(r)} + \sum_j [Z_j^{(\text{E})}]_{12} [Z_j^{(\text{E})}]_{21}. \tag{63}$$

This Hamiltonian consists of pairwise commuting terms that are invariant under $\text{Rep}(D_8) \times \mathbb{Z}_2$ symmetry and has a non-degenerate and gapped ground state – see Appendix B for a treatment of this Hamiltonian when written in terms of $\mathbb{Z}_4 \times \mathbb{Z}_2$ clock and shift operators. The entangled

---

[8]One way for a ground state $|\psi\rangle$ to necessarily be annihilated by a non-invertible symmetry operator $R$ is for $|\psi\rangle$ to carry a non-trivial one-dimensional charge of an invertible symmetry $U$ that satisfies $RU = R$. Indeed, since $U|\psi\rangle = e^{i\theta}|\psi\rangle$ with $\theta \notin 2\pi\mathbb{Z}$, it then follows that $|\psi\rangle$ satisfies $R|\psi\rangle = RU|\psi\rangle = e^{i\theta}R|\psi\rangle$ and, therefore, $R|\psi\rangle = 0$. A similar conclusion arises if the ground state satisfies $R_a|\psi\rangle = d_a e^{i\theta}|\psi\rangle$ with a non-invertible symmetry operator $R_a$ that itself satisfies $RR_a = d_a R$. It is unclear, however, whether all such annihilated ground states obey such relations.

[9]The simplest example is the two weak SPT states protected by $\mathbb{Z}_2$ internal and $\mathbb{Z}_L$ translation symmetries. The two distinct weak SPTs are the ground states of $\mathbb{Z}_2$-paramagnets $H = \pm \sum_j \sigma_j^x$ and are distinguished by the $\mathbb{Z}_2$ charge attached to each translation unit cell.

[10]SPTs protected by internal non-invertible symmetries have been studied in Refs. [74,86,109–112].

[11]There are additional entangled weak SPT phases that are compatible with the $\text{Rep}(D_8) \times \mathbb{Z}_2 \times \mathbb{Z}_L$ projective algebra (49). See Appendix C for related discussions based on TQFTs.

weak SPT ground state is given by

$$|\psi_{\text{GS}}\rangle = \sum_{\{\alpha_j,\varphi_j=0,1\}} (-1)^{\sum_j \alpha_j(\varphi_j+\varphi_{j-1})} \bigotimes_j \left|\psi_{\alpha_j,\varphi_j}\right\rangle, \tag{64a}$$

where

$$\left|\psi_{\alpha_j,\varphi_j}\right\rangle = \frac{1}{\sqrt{2}} \left(\left|s^{2\alpha_j+1}\right\rangle + (-1)^{\varphi_j}\left|s^{2\alpha_j+1}\,r\right\rangle\right). \tag{64b}$$

Because of the phase factor $(-1)^{\sum_j \alpha_j(\varphi_j+\varphi_{j-1})}$, this is state is not a trivial product state. When open boundary conditions are imposed, Hamiltonian (63) supports degenerate edge states. Under the global symmetries, the ground state (64) transforms as

$$\begin{aligned}
U_{s^2}\left|\psi_{\text{GS}}\right\rangle &= \left|\psi_{\text{GS}}\right\rangle, & T\left|\psi_{\text{GS}}\right\rangle &= \left|\psi_{\text{GS}}\right\rangle, \\
R_{\text{P}_1}\left|\psi_{\text{GS}}\right\rangle &= \left|\psi_{\text{GS}}\right\rangle, & R_{\text{P}_2}\left|\psi_{\text{GS}}\right\rangle &= (-1)^L\left|\psi_{\text{GS}}\right\rangle, & R_{\text{E}}\left|\psi_{\text{GS}}\right\rangle &= (1+(-1)^L)\left|\psi_{\text{GS}}\right\rangle.
\end{aligned} \tag{65}$$

While $|\psi_{\text{GS}}\rangle$ is translationally invariant, it carries non-trivial $R_{\text{P}_2}$ charge and is annihilated by the non-invertible $R_{\text{E}}$ operator when $L$ is odd.

Let us study the SPT Hamiltonian (63) when an E-twist is introduced. Following Section 3.2.2, we find the defect Hamiltonian to be

$$H_{\text{SPT}}^{\text{E}} = -\sum_{j=1}^{L-1} \overleftarrow{X}_j^{(r)}\overrightarrow{X}_{j+1}^{(r)} - \widehat{\text{E}}(r)\overleftarrow{X}_L^{(r)}\overrightarrow{X}_1^{(r)} + \sum_j [Z_j^{(\text{E})}]_{12}\,[Z_j^{(\text{E})}]_{21}, \tag{66}$$

where $\widehat{\text{E}}(r)$ is the operator acting on the internal two-dimensional Hilbert space of the E defect. There are two degenerate ground states of twisted Hamiltonian (66) depending on the defect state. If we denote by $|\pm\rangle$ the eigenstate of $\widehat{\text{E}}(r)$ with eigenvalue $\pm 1$, the two ground states are found to be

$$\begin{aligned}
\left|\psi_{\text{GS}}^+\right\rangle &= |+\rangle \otimes \left(\sum_{\{\alpha_j,\varphi_j=0,1\}} (-1)^{\sum_j \alpha_j(\varphi_j+\varphi_{j-1})} \bigotimes_j \left|\psi_{\alpha_j,\varphi_j}\right\rangle\right), \\
\left|\psi_{\text{GS}}^-\right\rangle &= |-\rangle \otimes \left(\sum_{\{\alpha_j,\varphi_j=0,1\}} (-1)^{\alpha_1}(-1)^{\sum_j \alpha_j(\varphi_j+\varphi_{j-1})} \bigotimes_j \left|\psi_{\alpha_j,\varphi_j}\right\rangle\right).
\end{aligned} \tag{67}$$

These two states transform under the $\text{Rep}(G)$ symmetry in the same manner as the untwisted SPT ground state (64) while they differ under the action of $\mathbb{Z}_2$ center and twisted translation symmetries as

$$\begin{aligned}
U_{s^2}\left|\psi_{\text{GS}}^+\right\rangle &= +\left|\psi_{\text{GS}}^+\right\rangle, & U_{s^2}\left|\psi_{\text{GS}}^-\right\rangle &= -\left|\psi_{\text{GS}}^-\right\rangle, \tag{68a} \\
T_{\text{E}}\left|\psi_{\text{GS}}^+\right\rangle &= \left|\psi_{\text{GS}}^-\right\rangle, & T_{\text{E}}\left|\psi_{\text{GS}}^-\right\rangle &= \left|\psi_{\text{GS}}^+\right\rangle. \tag{68b}
\end{aligned}$$

When a E defect is inserted, the symmetry operators $U_{s^2}$ and $T_{\text{E}}$ act like Pauli $\sigma^z$ and $\sigma^x$ matrices on the ground state manifold. Therefore, they anti-commute reflecting the projective algebra (46).

## 5 Gauging web

Thus far, we have explored the projective algebras from the $\text{Rep}(G) \times Z(G) \times \mathbb{Z}_L$ symmetry (see Table 1) through its constraints on the allowed phases and ground state entanglement

patterns of symmetric Hamiltonians. However, these projective algebras also influence the gauging web[12] of $\text{Rep}(G) \times Z(G) \times \mathbb{Z}_L$ symmetric models. Indeed, it is well known that similar projective algebras that give rise to LSM anomalies [22, 51] and 't Hooft anomalies [52, 53] of finite invertible symmetries can be probed by gauging anomaly-free sub-symmetries, where after gauging, the dual symmetry has a non-trivial interplay with the symmetries that were not gauged.

In this section, we explore the effect of the projective algebra between the internal $\text{Rep}(G) \times Z(G)$ symmetry and $\mathbb{Z}_L$ lattice translations on the gauging web of the $G$-based XY model (12). As summarized in Fig. 2, the dual symmetries include non-invertible and non-Abelian modulated symmetries as well as non-invertible lattice translation symmetries. We will gauge various $Z(G)$ and $\text{Rep}(G)$ sub-symmetries on the lattice in the context of the $G$-based XY model. We refer the reader to Appendix D where we discuss in detail gauging $G$ and $\text{Rep}(G)$ symmetries in $G$-qudit lattice models.

In what follows, we assume that the number of lattice sites $L$ satisfies Eq. (43). For such system sizes, the $Z(G)$ symmetry operators $U_z$ and $\text{Rep}(G)$ symmetry operators commute. When $L$ does not satisfy (43), $U_z$ and $R_\Gamma$ no longer commute and obey the non-trivial projective algebra (42). Then, there are irreps $\Gamma$ for which $R_\Gamma$ is a charged operator under the $Z(G)$ symmetry and $z \in Z(G)$ for which $U_z$ is charged under the $\text{Rep}(G)$ symmetry. Therefore, for such system sizes, gauging one symmetry will explicitly break the other symmetry down to a sub-symmetry.

## 5.1 Gauging $Z(G)$: Non-invertible dipole symmetry

We start by gauging the $Z(G)$ sub-symmetry of the $G$-based XY model (12) represented by $U_z$ (17a). To do so, we introduce $Z(G)$-qudits onto the links $\langle j, j+1 \rangle$ of the lattice, acted on by the generalized Pauli operators $\overrightarrow{\mathcal{X}}^{(z)}_{j,j+1} = \overleftarrow{\mathcal{X}}^{(\bar{z})}_{j,j+1}$ and $\mathcal{Z}^{(\rho)}$ where $z \in Z(G)$ and $\rho : Z(G) \to U(1)$ is an irrep of $Z(G)$. The gauging is then implemented by enforcing the Gauss laws

$$G^{(z)}_j = \overleftarrow{\mathcal{X}}^{(z)}_{j-1,j} \, \overrightarrow{X}^{(z)}_j \, \overrightarrow{\mathcal{X}}^{(z)}_{j,j+1} \overset{!}{=} 1, \tag{69}$$

where the notation $\overset{!}{=}$ means the equality holds on the constrained gauge-invariant Hilbert space. These Gauss laws trivialize the $Z(G)$ symmetry since its symmetry operators can be written as $U_z = \prod_j G^{(z)}_j$ which is set to 1 when (69) is enforced.

The gauged Hamiltonian model is found by minimally coupling the new $Z(G)$-qudits to the $G$-based XY model to ensure the Hamiltonian commutes with the Gauss operators $G^{(z)}_j$. Doing so, we find

$$H_{XY/Z(G)} = \sum_j \left( \sum_\Gamma J_\Gamma \, \mathcal{Z}^{(\rho_\Gamma)}_{j,j+1} \text{Tr}\left( Z^{(\bar{\Gamma})}_j Z^{(\Gamma)}_{j+1} \right) + \sum_g K_g \, \overleftarrow{X}^{(g)}_j \overrightarrow{X}^{(g)}_{j+1} \right) + \text{H.c.}, \tag{70}$$

where the irrep $\rho_\Gamma$ of $Z(G)$ is defined as[13]

$$\rho_\Gamma(z) = e^{i\phi_\Gamma(z)} = \frac{\chi_\Gamma(z)}{d_\Gamma}. \tag{71}$$

---

[12]Gauging webs are sometimes called duality webs or orbifold groupoids.

[13]It follows from Schur's lemma that $\rho_\Gamma$, defined by Eq. (71), is an irrep of $Z(G)$. Indeed, a corollary of Schur's lemma is that for any irrep $\Gamma$ of $G$, $\Gamma(z) = \frac{\chi_\Gamma(z)}{d_\Gamma} \mathbf{1}_{d_\Gamma}$ for all $z \in Z(G)$, where $\mathbf{1}_{d_\Gamma}$ is the $d_\Gamma \times d_\Gamma$ identity matrix. Therefore, $\rho_\Gamma(z_1 z_2) = \frac{\chi_\Gamma(z_1 z_2)}{d_\Gamma} = \frac{1}{d_\Gamma} \text{Tr}(\Gamma(z_1 z_2)) = \frac{1}{d_\Gamma} \text{Tr}(\Gamma(z_1)\Gamma(z_2)) = \frac{1}{d_\Gamma} \frac{\chi_\Gamma(z_1)}{d_\Gamma} \frac{\chi_\Gamma(z_2)}{d_\Gamma} \text{Tr}(\mathbf{1}_{d_\Gamma}) = \frac{\chi_\Gamma(z_1)}{d_\Gamma} \frac{\chi_\Gamma(z_2)}{d_\Gamma} = \rho_\Gamma(z_1)\rho_\Gamma(z_2)$, and $\rho_\Gamma$ is indeed an irrep of $Z(G)$.

Following Appendix D.3, this Hamiltonian can be rotated to a basis with a tensor product Hilbert space where there are $Z(G)$ qudits on links and $\mathrm{Inn}(G) \simeq G/Z(G)$ qudits on sites. For simplicity, we will discuss the gauged model's symmetries in the basis used here.

The dual symmetries are those that commute with the gauged Hamiltonian $H_{XY/Z(G)}$ and the Gauss operators $G_j^{(z)}$. Because we gauged a $Z(G)$ symmetry, we expect there to be a dual $Z(G)$ symmetry. And, indeed, there is one that is represented by

$$U_\rho^\vee = \prod_{j=1}^{L} \mathcal{Z}_{j,j+1}^{(\rho)}. \tag{72}$$

This is a $Z(G)$ symmetry because the characters of irreps $\rho$ multiply according to the Pontryagin dual $\mathrm{Hom}(Z(G), U(1)) \simeq Z(G)$ of $Z(G)$. The gauged model is clearly translation invariant, but what about the $\mathrm{Rep}(G)$ symmetry (17b) of the $G$-based XY model? The $\mathrm{Rep}(G)$ operator $\mathrm{Tr}(\prod_{j=1}^{L} Z_j^{(\Gamma)})$ does not commute with the Gauss operators, but since we assume $L$ satisfies (43), it can be made gauge-invariant by minimal coupling.[14] Doing so, we find

$$R_\Gamma = \mathrm{Tr}\left( \prod_{j=1}^{L} Z_j^{(\Gamma)} (\mathcal{Z}_{j,j+1}^{(\rho_\Gamma)})^{-j} \right) \equiv \sum_{\{\boldsymbol{g},\boldsymbol{z}\}} \chi_\Gamma\left( \prod_{j=1}^{L} g_j z_{j,j+1}^{-j} \right) |\boldsymbol{g} ; \boldsymbol{z}\rangle\langle \boldsymbol{g} ; \boldsymbol{z}| , \tag{73}$$

which is a $\mathrm{Rep}(G)$ operator that commutes with $H_{XY/Z(G)}$ and $G_j^{(z)}$. Thus, the $\mathrm{Rep}(G)$ symmetry survives the gauging. However, after gauging, the $\mathrm{Rep}(G)$ operator is now a modulated symmetry operator that satisfies

$$T R_\Gamma T^\dagger = U_{\rho_\Gamma}^\vee R_\Gamma. \tag{74}$$

Therefore, after gauging $Z(G)$, the dual symmetry is a non-trivial extension of the $\mathbb{Z}_L$ lattice translations by $\mathrm{Rep}(G) \times Z(G)$ due to the projective algebras in Table 1. Interestingly, when $G$ is non-Abelian, this is a type of non-invertible dipole symmetry.

## 5.2 Gauging $\mathrm{Rep}(G)$: Non-Abelian dipole symmetry

Let us return to the $G$-based XY model, and instead of gauging the $Z(G)$ sub-symmetry, we now gauge the $\mathrm{Rep}(G)$ sub-symmetry.[15] Because $G$-qudits naturally have operators associated with both group elements and irreps of $G$—the $X$ and $Z$ type operators, respectively—we can gauge $\mathrm{Rep}(G)$ using $G$ qudits. Therefore, we now introduce new $G$-qudits onto the links $\langle j, j+1 \rangle$ of the lattice and denote their generalized Pauli operators by $\overrightarrow{\mathcal{X}}_{j,j+1}^{(g)}$, $\overleftarrow{\mathcal{X}}_{j,j+1}^{(g)}$, and $\mathcal{Z}_{j,j+1}^{(\Gamma)}$. To gauge $\mathrm{Rep}(G)$, we then enforce the Gauss laws

$$[G_j^{(\Gamma)}]_{\alpha\beta} = [\mathcal{Z}_{j-1,j}^{(\Gamma)} Z_j^{(\Gamma)} \mathcal{Z}_{j,j+1}^{(\overline{\Gamma})}]_{\alpha\beta} \overset{!}{=} \delta_{\alpha\beta}. \tag{75}$$

These are MPO identities, and because they apply for all $\Gamma$, they implement the local constraints $h_j = \bar{h}_{j-1,j} h_{j,j+1}$ for all the site and link qudit states $\boldsymbol{h}$ and h. The Gauss laws trivialize the $\mathrm{Rep}(G)$ symmetry operators $R_\Gamma$ since they can be written as $R_\Gamma = \mathrm{Tr}(\prod_{j=1}^{L} G_j^{(\Gamma)})$. Furthermore, it is known from QFT and category theory arguments that gauging a finite $G$ symmetry yields a dual $\mathrm{Rep}(G)$ symmetry and that gauging the dual $\mathrm{Rep}(G)$ symmetry returns back the $G$-symmetric theory [52]. In Appendix D, we confirm that the Gauss laws (75), in fact, satisfy this on the lattice.

---

[14]For general $L$, because $U_z R_\Gamma U_z^\dagger = (\chi_\Gamma(z)/d_\Gamma)^{-L} R_\Gamma$, the $\mathrm{Rep}(G)$ symmetry after gauging is explicitly broken down to a sub-symmetry formed by $R_\Gamma$ whose irreps $\Gamma$ satisfy $(\chi_\Gamma(z)/d_\Gamma)^L = 1$ for all $z$.

[15]We refer the reader to Appendix D.2 for a more detailed discussion of gauging $\mathrm{Rep}(G)$ at the level of bond algebras.

The gauged Hamiltonian model is found by minimally coupling the new $G$-qudits to the $G$-based XY model to ensure the Hamiltonian commutes with the Gauss operators $G_j^{(\Gamma)}$. Doing so, we find

$$H_{XY/\mathrm{Rep}(G)} = \sum_j \left( \sum_\Gamma J_\Gamma \mathrm{Tr}\left( Z_j^{(\overline{\Gamma})} Z_{j+1}^{(\Gamma)} \right) + \sum_g K_g \overleftarrow{\mathcal{X}}_{j,j+1}^{(g)} \overleftarrow{X}_j^{(g)} \overrightarrow{X}_{j+1}^{(g)} \right) + \mathrm{H.c.} \tag{76}$$

To discuss the dual symmetries, and for exploring the gauging web further in the next subsection, it is convenient to make a change of basis. In particular, consider the unitary operator

$$W = \prod_j W_j, \qquad W_j = \sum_{\boldsymbol{h}, \mathsf{h}} \left| \cdots, \mathsf{h}_{j-1,j} h_j \bar{\mathsf{h}}_{j,j+1}, \cdots ; \mathsf{h} \right\rangle \left\langle \boldsymbol{h} ; \mathsf{h} \right|, \tag{77}$$

where $\boldsymbol{h}$ and $\mathsf{h}$ denote the configurations of $G$-qudits on sites and links, respectively. The unitary operator $W_j$ is a group-based control gate that shifts the value of $G$-qudits on site $j$ according to the $G$-qudit states on links $\langle j-1, j \rangle$ and $\langle j, j+1 \rangle$. Using $W$, we perform the unitary transformation

$$\begin{aligned} \overleftarrow{\mathcal{X}}_{j,j+1}^{(g)} \overleftarrow{X}_j^{(g)} \overrightarrow{X}_{j+1}^{(g)} &\to \overleftarrow{\mathcal{X}}_{j,j+1}^{(g)}, \\ [\mathcal{Z}_{j-1,j}^{(\Gamma)} Z_j^{(\Gamma)} \mathcal{Z}_{j,j+1}^{(\overline{\Gamma})}]_{\alpha\beta} &\to [Z_j^{(\Gamma)}]_{\alpha\beta}, \end{aligned} \tag{78}$$

after which, the Gauss laws become $[Z_j^{(\Gamma)}]_{\alpha\beta} \stackrel{!}{=} \delta_{\alpha\beta}$ and, in the physical Hilbert space, the Hamiltonian is

$$H_{XY/\mathrm{Rep}(G)} = \sum_j \left( \sum_\Gamma J_\Gamma \mathrm{Tr}\left( \mathcal{Z}_{j,j+1}^{(\overline{\Gamma})} \mathcal{Z}_{j-1,j}^{(\Gamma)} \mathcal{Z}_{j,j+1}^{(\overline{\Gamma})} \mathcal{Z}_{j+1,j+2}^{(\Gamma)} \right) + \sum_g K_g \overleftarrow{\mathcal{X}}_{j,j+1}^{(g)} \right) + \mathrm{H.c.} \tag{79}$$

Having gauged $\mathrm{Rep}(G)$ in the $G$-based XY model and found the resulting Hamiltonian, we can now identify its dual symmetries. Since we gauged the $\mathrm{Rep}(G)$ symmetry, we expect the model $H_{XY/\mathrm{Rep}(G)}$ to have a dual $G$ symmetry. And, indeed, it does have a $G$ symmetry, which is represented by

$$R_g^\vee = \prod_j \overrightarrow{\mathcal{X}}_{j,j+1}^{(g)}. \tag{80}$$

Since the $Z(G)$ symmetry represented by $\prod_j \overrightarrow{X}_j^{(z)}$ was not gauged, it will have a non-trivial image under the $\mathrm{Rep}(G)$ gauging map. To find it, we first note that because we assume $L$ satisfies (43), we can write $\prod_j \overrightarrow{X}_j^{(z)}$ before gauging in terms of $\mathrm{Rep}(G)$-symmetric operators as $\prod_{j=1}^{L-1} (\overleftarrow{X}_j^{(\bar{z})} \overrightarrow{X}_{j+1}^{(\bar{z})})^j$. Then, after minimally coupling $\overleftarrow{\mathcal{X}}_{j,j+1}^{(\bar{z})}$ and then conjugating by $W$, we find that the image of $U_z$ after gauging is

$$U_z = \prod_j \overrightarrow{\mathcal{X}}_{j,j+1}^{(z^j)}. \tag{81}$$

The operator $U_z$ is now a modulated symmetry operator after gauging $\mathrm{Rep}(G)$ and transforms under lattice translations as

$$T U_z T^\dagger = (R_z^\vee)^\dagger U_z. \tag{82}$$

This means that the dual symmetry group after gauging is $(G \times Z(G)) \rtimes \mathbb{Z}_L$, which is a type of non-Abelian dipole symmetry.[16] Therefore, gauging the $\mathrm{Rep}(G)$ sub-symmetry leads to a dual symmetry for which the projective algebra (42) is replaced by a non-trivial extension of $\mathbb{Z}_L$ translations by the internal $G \times Z(G)$.

---

[16] See appendix E for additional discussion on general aspects of finite non-Abelian modulated symmetries.

### 5.3 Gauging Rep$(G) \times Z(G)$: Non-invertible translations

Having found the gauged models upon gauging Rep$(G)$ and $Z(G)$ individually in the $G$-based XY model, let us now consider the model constructed by gauging the entire Rep$(G) \times Z(G)$ internal symmetry. While it is possible to do so by starting with the $G$-based XY model and gauging the entire Rep$(G) \times Z(G)$ symmetry at once, it is convenient to instead start from the gauged Rep$(G)$ model (79) and then gauge its modulated $Z(G)$ sub-symmetry represented by (81).

There are many ways to gauge finite Abelian modulated symmetries [95]. To gauge the sub-symmetry represented by $U_z$, but not the $G$ sub-symmetry (80), we introduce $Z(G)$-qudits onto the sites, which are acted on by the group-based Pauli operators $\overleftarrow{\mathsf{X}}_j^{(z)}$, $\overrightarrow{\mathsf{X}}_j^{(z)}$, and $\mathsf{Z}_j^{(\rho)}$. The Gauss laws are

$$G_j^{(z)} = \overleftarrow{\mathsf{X}}_j^{(z)} \overrightarrow{\mathcal{X}}_{j,j+1}^{(z^j)} \overrightarrow{\mathsf{X}}_{j+1}^{(z)} \overset{!}{=} 1, \tag{83}$$

which are ones that were not considered in Ref. 95. Upon minimal coupling of the Hamiltonian (79), this gives rise to the gauged model

$$H_{XY/\text{Rep}(G) \times Z(G)} = \sum_j \left( \sum_\Gamma J_\Gamma \, [\mathsf{Z}_j^{(\rho_\Gamma)}]^{-j+} [\mathsf{Z}_{j+1}^{(\rho_\Gamma)}]^{j+1} \, \text{Tr} \left( \mathcal{Z}_{j,j+1}^{(\overline{\Gamma})} \mathcal{Z}_{j-1,j}^{(\Gamma)} \mathcal{Z}_{j,j+1}^{(\overline{\Gamma})} \mathcal{Z}_{j+1,j+2}^{(\Gamma)} \right) + \sum_g K_g \, \overleftarrow{\mathcal{X}}_{j,j+1}^{(g)} \right) + \text{H.c.} \tag{84}$$

After gauging the modulated $Z(G)$ symmetry, the $G$ symmetry (80) is unaffected and remains represented by the unitary operators

$$R_g^\vee = \prod_j \overrightarrow{\mathcal{X}}_{j,j+1}^{(g)}. \tag{85}$$

Since $Z(G)$ was gauged, as expected, there is a dual $Z(G)$ symmetry represented by

$$U_z^\vee = \prod_j \mathsf{Z}_j^{(\rho)}. \tag{86}$$

Interestingly, despite the original $Z(G)$ symmetry being modulated, the dual $Z(G)$ symmetry is uniform. However, because only the modulated $Z(G)$ symmetry was gauged, the Gauss laws (83) explicitly break the lattice translation symmetry down to a subgroup. Indeed, the Gauss operators $G_j^{(z)}$ and the Hamiltonian (84) are not symmetric with respect to $T$ but $T^l$ with $l = \text{lcm}(\{|z|, z \in Z(G)\})$, the least common multiple of the order of all elements in $Z(G)$.

Although ordinary lattice translations $T$ is explicitly broken, its avatar still persists after gauging: there exists a non-invertible translation symmetry $\mathsf{D}_T := \mathsf{D}\,T$, where $\mathsf{D}$ implements the map

$$\begin{aligned}
\mathcal{Z}_{j-1,j}^{(\overline{\Gamma})} \mathcal{Z}_{j,j+1}^{(\Gamma)} &\to \mathcal{Z}_{j-1,j}^{(\overline{\Gamma})} \mathsf{Z}_j^{(\rho_\Gamma)} \mathcal{Z}_{j,j+1}^{(\Gamma)}, \\
\overleftarrow{\mathsf{X}}_j^{(z)} \overrightarrow{\mathsf{X}}_{j+1}^{(z)} &\to \overleftarrow{\mathsf{X}}_j^{(z)} \overrightarrow{\mathcal{X}}_{j,j+1}^{(z)} \overrightarrow{\mathsf{X}}_{j+1}^{(z)}.
\end{aligned} \tag{87}$$

$\mathsf{D}$ is a non-invertible operator because it has a non-trivial kernel:

$$\left( \prod_j \overrightarrow{\mathcal{X}}_{j,j+1}^{(z)} \right) \mathsf{D} = \left( \prod_j \mathsf{Z}_j^{(\rho_\Gamma)} \right) \mathsf{D} = \mathsf{D}. \tag{88}$$

Furthermore, on an infinite chain, we can deduce the general transformation law of $\mathsf{D}$ from (87), which maps

$$\overrightarrow{\mathsf{X}}_j^{(z)} \to \overrightarrow{\mathsf{X}}_j^{(z)} \prod_{k \leq j} \overrightarrow{\mathcal{X}}_{k-1,k}^{(z)}, \tag{89}$$

$$\mathcal{Z}_{j,j+1}^{(\overline{\Gamma})} \to \mathcal{Z}_{j,j+1}^{(\overline{\Gamma})} \prod_{k \geq j} \mathsf{Z}_{k+1}^{(\rho_\Gamma)}. \tag{90}$$

This transformation preserves the generalized Pauli operators' algebra, and using it, it is clear that $D_T$ commutes with (84). Therefore, gauging the $\text{Rep}(G) \times Z(G)$ internal symmetry in this way leads to a dual internal $G \times Z(G)$ symmetry but, due to the projective algebra (42), with a non-invertible lattice translation symmetry. This generalizes the $G = \mathbb{Z}_2$ case discussed in Ref. 22 for the ordinary quantum XY model.

# 6  The Symmetry TFT

In Section 5, we studied the gauging web of the $\text{Rep}(G) \times Z(G)$ symmetry directly in the 1+1D $G$-based XY model. An alternative and fruitful perspective of gauging web is provided by the Symmetry TFT (SymTFT). The SymTFT is a powerful tool for separating the symmetries of a theory from its dynamics using a topological theory in one higher dimension, and all symmetries related by discrete gauging share the same SymTFT [113–119]. In this section, we construct the SymTFT for the gauging web explored in the main text.

## 6.1  SymTFT as an SET

A systematic way to find the SymTFT for an invertible finite symmetry in $d + 1$D is to start with its $d + 2$D SPT characterizing its possible 't Hooft anomaly and then gauge the symmetry. Since all symmetries related by discrete gauging have the same SymTFT, we can construct the SymTFT for the gauging web starting from the anomaly-free $G \times Z(G)$ dipole symmetry in $1 + 1$D from Section 5.2. Because this symmetry is 't Hooft anomaly-free, its $2 + 1$D SPT in one higher dimension is the trivial SPT (i.e., the product state SPT). We will construct its SymTFT on the lattice, and the resulting SymTFT will be formulated as a quantum code.

Consider a two-dimensional square lattice $\Lambda$ with $G$-qudits on sites $r \equiv (x, y) \in \mathbb{Z}_{L_x} \times \mathbb{Z}_{L_y}$ with periodic boundary conditions, and a Hilbert space with a tensor product decomposition of the local Hilbert spaces $\mathbb{C}[G]$ for each site. While constructing the SymTFT, we assume the periodic boundary conditions $(x, y) \sim (x + L_x, y) \sim (x, y + L_y)$. The $G \times Z(G)$ dipole symmetry extended to this two-dimensional system is generated by

$$U_g = \prod_{r \in \Lambda} \overrightarrow{X}_r^{(g)}, \qquad D_z = \prod_{r \in \Lambda} \left( \overrightarrow{X}_r^{(z)} \right)^x . \tag{91}$$

Since this symmetry is 't Hooft anomaly free (i.e., it is on-site), its related SPT is the trivial paramagnet model

$$H = -\sum_{r \in \Lambda} \left( \frac{1}{|G|} \sum_{g \in G} \overrightarrow{X}_r^{(g)} \right), \tag{92}$$

where the overall factor $1/|G|$ is included to make $H$ a sum of commuting projectors.

Following Ref. 95, we gauge the $G \times Z(G)$ symmetry (91) of $H$ sequentially by first gauging the $G$ sub-symmetry represented by $U_g$ and then gauging the $Z(G)$ symmetry represented by $D_z$. To gauge the $G$ sub-symmetry, we introduce $G$-qudits onto the links of the lattice, making the local Hilbert space $\mathbb{C}[G]$ on each edge and $\mathbb{C}[G]$ on each site. We label the links $\langle r, r + e_x \rangle$ and $\langle r, r + e_y \rangle$ using the notation $(r, x)$ and $(r, y)$, respectively. The Gauss laws introducing the gauge redundancy are

$$G_r^{(g)} := \overrightarrow{X}_r^{(g)} A_r^{(g)} \stackrel{!}{=} 1 , \tag{93}$$

where

$$A_r^{(g)} := \overleftarrow{X}_{r-e_x,x}^{(g)} \overrightarrow{X}_{r,x}^{(g)} \overleftarrow{X}_{r-e_y,y}^{(g)} \overrightarrow{X}_{r,y}^{(g)} \equiv \quad (94)$$

Furthermore, since $G$ is a discrete group, the new $G$-qudits residing on links obey the flatness condition

$$\sum_{\{g\}} \delta_{g_{r,x}\,g_{r+e_x,y}\,g_{r+e_y,x}^{-1}\,g_{r,y}^{-1},\,1} |g\rangle\langle g| \overset{!}{=} 1, \quad (95)$$

which, in terms of the generalized Pauli operators, is the local constraint

$$\sum_{\Gamma} \frac{d_\Gamma}{|G|} \mathcal{F}_r^{(\Gamma)} \overset{!}{=} 1, \quad (96)$$

where

$$\mathcal{F}_r^{(\Gamma)} := \mathrm{Tr}\left(Z_{r,x}^{(\Gamma)} Z_{r+e_x,y}^{(\Gamma)} Z_{r+e_y,x}^{(\overline{\Gamma})} Z_{r,y}^{(\overline{\Gamma})}\right) \equiv \quad . \quad (97)$$

It is easily verified that

$$A_r^{(g)} \mathcal{F}_{r'}^{(\Gamma)} = \mathcal{F}_{r'}^{(\Gamma)} A_r^{(g)}, \quad (98)$$

for all $r$, $r'$, $g$, and $\Gamma$. Using the Gauss laws (93) and enforcing the flatness condition (96) energetically, the gauged model's Hamiltonian is the quantum double model Hamiltonian

$$H = -\frac{1}{|G|} \sum_{r \in \Lambda} \left( \sum_g A_r^{(g)} + \sum_\Gamma d_\Gamma \mathcal{F}_r^{(\Gamma)} \right). \quad (99)$$

The original $G$-qudits on sites can now be rotated away using a unitary transformation, and the entire Hilbert space is a tensor product of $\mathbb{C}[G]$ belonging to each edge.

After gauging the $G$ symmetry, the operator $D_z$ that generates the $Z(G)$ sub-symmetry of (91) becomes

$$D_z = \prod_{r \in \Lambda} \overrightarrow{X}_{r,x}^{(z)}, \quad (100)$$

acting on only the $G$-qudits that reside on horizontal links. To gauge this sub-symmetry, we introduce $Z(G)$-qudits onto the links, acted on by the generalized Pauli operators $\overrightarrow{\mathcal{X}}^{(z)} = \overleftarrow{\mathcal{X}}^{(\bar{z})} \equiv \mathcal{X}^{(z)}$ and $\mathcal{Z}^{(\rho)}$ where $\rho$ are the irreps of $(Z(G))$. This makes the local Hilbert space on each edge $\mathbb{C}[G \times Z(G)]$ subject to the Gauss laws $\mathcal{G}_r^{(z)} \overset{!}{=} 1$ where

$$\mathcal{G}_r^{(z)} := \overrightarrow{X}_{r,x}^{(z)} \mathcal{X}_{r-e_x,x}^{(\bar{z})} \mathcal{X}_{r,x}^{(z)} \mathcal{X}_{r-e_y,y}^{(\bar{z})} \mathcal{X}_{r,y}^{(z)} \equiv \quad (101)$$

Furthermore, the new qudits obey a flatness condition. In terms of their Pauli operators, this is the local constraint $B_{\Box_r}^{(\rho)} \stackrel{!}{=} 1$ where

$$B_{\Box_r}^{(\rho)} := \mathcal{Z}_{r,x}^{(\rho)} \, \mathcal{Z}_{r+e_x,y}^{(\rho)} \, \mathcal{Z}_{r+e_y,x}^{(\overline{\rho})} \, \mathcal{Z}_{r,y}^{(\overline{\rho})} \equiv \begin{array}{c} \mathcal{Z}^{(\overline{\rho})} \\ \mathcal{Z}^{(\overline{\rho})} \Box \mathcal{Z}^{(\rho)} \\ \mathcal{Z}^{(\rho)} \end{array}. \tag{102}$$

Minimally coupling $\mathcal{Z}$ and then energetically enforcing the Gauss laws (101) and flatness condition (102), we arrive at the commuting projector Hamiltonian

$$H_{\text{SymTFT}} = -\sum_{r \in \Lambda} \left( \mathbb{A}_r + \mathbb{G}_r + \mathbb{B}_{\Box_r} + \mathbb{F}_{\Box_r} \right),$$

$$\mathbb{A}_r = \frac{1}{|G|} \sum_g A_r^{(g)}, \qquad\qquad \mathbb{G}_r = \frac{1}{|Z(G)|} \sum_z \mathcal{G}_r^{(z)}, \tag{103}$$

$$\mathbb{B}_{\Box_r} = \frac{1}{|Z(G)|} \sum_\rho B_{\Box_r}^{(\rho)}, \qquad\qquad \mathbb{F}_{\Box_r} = \frac{1}{|G|} \sum_\Gamma d_\Gamma F_{\Box_r}^{(\Gamma)},$$

where $A_r^{(g)}$ is in (94), $\mathcal{G}_r^{(z)}$ is in (101), $B_{\Box_r}^{(\rho)}$ is in (102) and

$$F_{\Box_r}^{(\Gamma)} := \mathcal{Z}_{r,y}^{(\rho_\Gamma)\dagger} \, \text{Tr}\left( Z_{r,x}^{(\Gamma)} Z_{r+e_x,y}^{(\Gamma)} Z_{r+e_y,x}^{(\overline{\Gamma})} Z_{r,y}^{(\overline{\Gamma})} \right) \equiv \begin{array}{c} Z^{(\overline{\Gamma})} \\ Z^{(\overline{\Gamma})} \Box Z^{(\Gamma)} \\ \mathcal{Z}^{(\rho_\Gamma)\dagger} \\ Z^{(\Gamma)} \end{array}. \tag{104}$$

In the minimally coupled version of $F_{\Box_r}^{(\Gamma)}$, $\rho_\Gamma$ is an irrep of $Z(G)$ defined by $\rho_\Gamma(z) = \chi_\Gamma(z)/d_\Gamma$, which is the overall phase factor of $\Gamma(z)$ (see footnote 13). When $G = \mathbb{Z}_N$, the anyons of this stabilizer code was studied in Ref. 120.

The ground state subspace of the Hamiltonian 103 is

$$\mathcal{V} = \text{Span}_\mathbb{C}\left\{ |\psi\rangle \in \mathcal{H} \,\middle|\, \mathbb{A}_r |\psi\rangle = \mathbb{G}_r |\psi\rangle = \mathbb{B}_\Box |\psi\rangle = \mathbb{F}_\Box |\psi\rangle = |\psi\rangle \right\}, \tag{105}$$

where the total Hilbert space has the tensor product decomposition

$$\mathcal{H} = \bigotimes_e \mathbb{C}[G \times Z(G)], \tag{106}$$

over the edges $e$ of $\Lambda$. $\mathcal{V}$ is the code space that defines the SymTFT for the $1+1$D $G \times Z(G)$ dipole symmetry and all symmetries related to the $G \times Z(G)$ dipole symmetry by discrete gauging.

The allowed operators acting within the ground states—the logical operators—are those that commute with the stabilizers (i.e., the symmetry operators of 103). For general $G$, these operators are complicated modulated ribbon operators, reflecting how the translation symmetry of the underlying lattice $\Lambda$ enriches the $G \times Z(G)$ gauge theory in a non-trivial way, yielding the SymTFT for the $G \times Z(G)$ dipole symmetry [121]. In this case, the SymTFT is a topological order enriched by lattice translations where translations act as an anyon automorphism. Such

translation symmetry enriched topological orders (SETs) have recently gathered much attention, and are known to exhibit UV/IR mixing and have anyons with restricted mobility and position-dependent braiding [120, 122–134]. Recall that in a $2 + 1D$ $G$ gauge theory, anyons $a_{([g],\alpha)}$ are labeled by conjugacy classes $[g]$ of $G$ and irreps $\alpha$ of the centralizer $C(g)$ of $g$. From the translation action of the $G \times Z(G)$ and $\text{Rep}(G) \times Z(G)$ dipole symmetries, Eqs. (74) and (82) respectively, lattice translations in the $x$ direction act on the purely electric and magnetic anyons as

$$a_{([1],\Gamma)} \rightarrow b_{([1],\rho_\Gamma)} \, a_{([1],\Gamma)} \,,$$
$$b_{([z],\mathbf{1})} \rightarrow b_{([z],\mathbf{1})} \, \bar{a}_{([z],\mathbf{1})} \,, \tag{107}$$

where $a$ label anyons of $G$ gauge theory and $b$ of $Z(G)$ gauge theory. We emphasize that Eq. (107) shows only how the purely electric and magnetic anyons transform and does not include the generally non-trivial action on dyonic anyons.

## 6.2 Gauging web and gapped boundaries

To study the gauging web explored in Section 5 using this SymTFT, we now enforce periodic boundary conditions only in the $x$-direction and designate the boundary at $y = L_y$ as the symmetry boundary and $y = 1$ boundary as the physical boundary. The Hamiltonian is then

$$H = H_{\text{Sym}} + H_{\text{SymTFT}} + H_{\text{Phys}} \,, \tag{108}$$

where $H_{\text{Sym}} + H_{\text{SymTFT}}$ is a commuting projector model with $H_{\text{SymTFT}}$ given by 103 and $H_{\text{Sym}}$ a sum of commuting projectors whose stabilizers act on the degrees of freedom on the symmetry boundary. We call such stabilizers boundary stabilizers. The classification of gapped boundaries of this quantum code is the same as those for a $G \times Z(G)$ quantum double model. Different gapped boundaries are labeled by the pairs $(\text{Cl}(K), [\omega])$ where $\text{Cl}(K)$ is an equivalence class of subgroups $K \subset G \times Z(G)$ up to conjugation and the cohomology class $[\omega] \in H^2(BK, U(1))$ for a representative subgroup $K \in \text{Cl}(K)$ [135–137]. The degrees of freedom residing on boundary edges are $K$-qudits.

The symmetry boundary for the $G \times Z(G)$ non-Abelian dipole symmetry is the Dirichlet boundary condition. This boundary comes from placing $\mathbb{Z}_1$ qudits on the boundary edges and choosing the boundary stabilizers to be

$$
\mathcal{Z}^{(\bar{\rho})} \;\boxed{\phantom{xx}}^{\mathbf{1}}\; \mathcal{Z}^{(\rho)} \,, \qquad
\begin{matrix} Z^{(\bar{\Gamma})} \\ \mathcal{Z}^{(\rho_\Gamma)\dagger} \end{matrix} \;\boxed{\phantom{xx}}^{\mathbf{1}}\; Z^{(\Gamma)} \,, \tag{109}
$$
$$\mathcal{Z}^{(\rho)} \qquad\qquad Z^{(\Gamma)}$$

where edges colored red are boundary edges. In the quantum information language, this boundary is the rough boundary for both the $G$ and $Z(G)$-qudits. The boundary operators commuting with these stabilizers as well as $H_{\text{SymTFT}}$ are

$$R_g^\vee = \prod_{j=1}^{L_x} \overleftarrow{X}_{(j,L_y-1),y}^{(g)} \,, \tag{110}$$

$$U_z = \prod_{j=1}^{L_x} \mathcal{X}_{(j,L_y-1),y}^{(\bar{z})} \, \overleftarrow{X}_{(j,L_y-1),y}^{(z^j)} \,. \tag{111}$$

As expected, these are $G \times Z(G)$ dipole symmetry operators that obey

$$T \, U_z \, T^\dagger = (R_{\bar{z}}^\vee)^\dagger \, U_z \,, \tag{112}$$

which is the same algebra as (82) appearing in the gauging web discussed in Section 5.2.

To explore the gauging web using the SymTFT, we now change the symmetry boundary by choosing different boundary stabilizers. Because the projective $\mathrm{Rep}(G) \times \mathbb{Z}(G)$ symmetry arises by gauging the $G$ symmetry, it can be realized by changing the $G$-qudit's rough boundary to a smooth boundary. Placing $G$-qudits on the boundary edges, the boundary stabilizers are now

$$
\begin{array}{ccc}
\underset{\mathcal{Z}^{(\rho)}}{\overset{\mathbf{1}}{\underset{\mathcal{Z}^{(\bar{\rho})} \quad \square \quad \mathcal{Z}^{(\rho)}}{}}} , &
\underset{Z^{(\Gamma)}}{\overset{Z^{(\bar{\Gamma})}}{\underset{\mathcal{Z}^{(\rho_\Gamma)\dagger}}{}}} Z^{(\Gamma)} , &
\overset{\overleftarrow{X}^{(g)} \quad \overrightarrow{X}^{(g)}}{\underset{\overleftarrow{X}^{(g)}}{\big|}}
\end{array} \tag{113}
$$

The boundary symmetry operators are

$$
R_\Gamma = \mathrm{Tr}\left( \prod_{j=1}^{L_x} Z^{(\Gamma)}_{(j,L_y),x} \right), \tag{114}
$$

$$
U_z = \prod_{j=1}^{L_x} \overrightarrow{X}^{(z)}_{(j,L_y),x} \, \mathcal{X}^{(\bar{z})}_{(j,L_y-1),y} \,, \tag{115}
$$

which are $\mathrm{Rep}(G) \times Z(G)$ operators that satisfy the same algebra as (42)

$$
R_\Gamma U_z = e^{i\phi_\Gamma(z)} U_z R_\Gamma \,. \tag{116}
$$

This is the symmetry in the gauging web that is present in the $G$-based XY model and was discussed in Section 3.1.

Let us now gauge the $Z(G)$ sub-symmetry of this $\mathrm{Rep}(G) \times Z(G)$ symmetry. This is done by switching the rough boundary of the $Z(G)$ qudits to a smooth boundary, so now both qudit flavors have the smooth boundary conditions. After doing so, the boundary stabilizers become

$$
\begin{array}{cc}
\underset{\mathcal{Z}^{(\rho)}}{\overset{\mathcal{Z}^{(\bar{\rho})}}{\underset{\mathcal{Z}^{(\bar{\rho})} \quad \square \quad \mathcal{Z}^{(\rho)}}{}}} , &
\underset{Z^{(\Gamma)}}{\overset{Z^{(\bar{\Gamma})}}{\underset{Z^{(\bar{\Gamma})} \, \square \, Z^{(\Gamma)}}{\mathcal{Z}^{(\rho_\Gamma)\dagger}}}} , \\[2em]
\overset{\overleftarrow{X}^{(g)} \quad \overrightarrow{X}^{(g)}}{\underset{\overleftarrow{X}^{(g)}}{\big|}} , &
\overset{\mathcal{X}^{(\bar{z})} \quad \mathcal{X}^{(z)} \overrightarrow{X}^{(z)}}{\underset{\mathcal{X}^{(\bar{z})}}{\big|}}
\end{array} \tag{117}
$$

The boundary symmetry operators are now

$$
R_\Gamma = \mathrm{Tr}\left( \prod_{j=1}^{L_x} Z^{(\Gamma)}_{(j,L_y),x} \big[ \mathcal{Z}^{(\rho_\Gamma)}_{(j,L_y),x} \big]^{-j} \right), \tag{118}
$$

$$
U_\rho^\vee = \prod_{j=1}^{L_x} \mathcal{Z}^{(\rho_\Gamma)}_{(j,L_y),x} \,, \tag{119}
$$

which satisfy the same algebra as (74)

$$T R_\Gamma T^\dagger = U^\vee_{\rho_\Gamma} R_\Gamma. \tag{120}$$

These are $\mathrm{Rep}(G) \times Z(G)$ dipole symmetry operators, which form the symmetry from Section 5.1 of the gauging web.

Lastly, let us show how to get the $G \times Z(G)$ internal symmetry with non-invertible translations discussed in Section 5.3. To do so, we start with both qudits obeying the rough boundary and then gauge the $Z(G)$ symmetry by choosing a particular smooth boundary for the $Z(G)$ qudits. In particular, we consider the boundary stabilizers

$$\tag{121}$$

The boundary symmetry operators are

$$R^\vee_g = \prod_{j=1}^{L_x} \overleftarrow{X}^{(g)}_{(j,L_y-1),y}, \tag{122}$$

$$U^\vee_\rho = \prod_{j=1}^{L_x} \mathcal{Z}^{(\rho)}_{(j,L_y),x}, \tag{123}$$

which are non-modulated $G \times Z(G)$ operators. However, translations by one site in the $x$-direction do not preserve these boundary stabilizers. Therefore, this symmetry boundary explicitly breaks lattice translations to a subgroup. However, there is a non-invertible translation operator $\mathsf{D}_T := \mathsf{D}\,T$ that does commute with the boundary stabilizers, where $\mathsf{D}$ acts on the boundary operators as

$$\tag{124}$$

$$\tag{125}$$

Therefore, there is a $G \times Z(G)$ non-modulated symmetry together with a non-invertible translations symmetry reproducing the symmetries from Section 5.3 of the gauging web.

## 6.3 Example $G = D_8$

Having formulated the SymTFT for the gauging web explored in Section 5, in this section, we specialize to the case when $G = D_8$. In particular, using a SymTFT perspective, we show that for $G = D_8$, the projective algebra allows SPTs phases to exist, confirming the discussions in Section 4.4.

When $G = D_8$, the SymTFT (105) is a $D_8 \times \mathbb{Z}_2$ gauge theory enriched non-trivially by lattice translations. This SymTFT has 88 different anyon types, corresponding to the different composites of the 22 anyons of $D_8 \subset D_8 \times \mathbb{Z}_2$ gauge theory and 4 anyons of $\mathbb{Z}_2 \subset D_8 \times \mathbb{Z}_2$ gauge theory. We label the 22 anyons related to $D_8$ by $a_{([g],\alpha)}$, where $[g]$ is the conjugacy class of $g \in D_8$ and $\alpha$ is an irrep of the centralizer $C(g)$, and the four anyons related to $\mathbb{Z}_2$ as $1$, $m$, $e$, and $f = e \times m$. In terms of the irreps and conjugacy classes of $\mathbb{Z}_2$, $m$ corresponds to the non-trivial conjugacy class $[-1]$ and trivial irrep $\mathbf{1}$, while $e$ corresponds to the trivial conjugacy class $[1]$ and non-trivial irrep $\mathbf{1}'$. Furthermore, when either the irrep $\alpha$ or the conjugacy class $[g]$ is trivial, we omit their label from $a_{([g],\alpha)}$ for brevity.

Lattice translations in the $x$ direction enrich this $D_8 \times \mathbb{Z}_2$ gauge theory non-trivially and act as an automorphism on these anyons. The action on the pure electric and magnetic anyons follows from (107) and is given by

$$m \to m \, a_{[s^2]}, \tag{126}$$

$$a_\mathsf{E} \to e \, a_\mathsf{E}. \tag{127}$$

Indeed, it is consistent with the non-trivial instances of Eqs. (112) and (120) for $G = D_8$ which are

$$T \, U_{s^2} \, T^\dagger = U_{s^2} \, R_{s^2}^\vee, \tag{128}$$

$$T \, R_\mathsf{E} \, T^\dagger = U_{\mathbf{1}'}^\vee \, R_\mathsf{E}, \tag{129}$$

respectively. The action of $T$ on the dyonic anyons follows from the fact that it is an automorphism and, therefore, preserves properties such as the anyons braiding statistics. Using this, we find that the entire action of $T$ is given by

$$
\begin{aligned}
T : \ & \left(a_{([s],\omega^3)}m, a_{([s],\omega)}f\right)\left(a_{([s],\omega^3)}e, a_{([s],\omega^3)}\right)\left(a_{([s],\omega)}m, a_{([s],\omega^3)}f\right)\left(a_{([s],\omega)}e, a_{([s],\omega)}\right) \\
& \left(a_{([s^2],\mathsf{P}_3)}f, a_{\mathsf{P}_3}f\right)\left(a_{([sr],01)}m, a_{([sr],10)}f\right)\left(a_{\mathsf{P}_2}f, a_{([s^2],\mathsf{P}_2)}f\right)\left(a_{([s^2],\mathsf{P}_1)}f, a_{\mathsf{P}_1}f\right)\left(a_\mathsf{E}e, a_\mathsf{E}\right) \\
& \left(a_{[s^2]}m, m\right)\left(a_{[s^2]}f, f\right)\left(a_{([s^2],\mathsf{P}_3)}m, a_{\mathsf{P}_3}m\right)\left(a_\mathsf{E}f, a_{([s^2],\mathsf{E})}m\right)\left(a_\mathsf{E}m, a_{([s^2],\mathsf{E})}f\right) \\
& \left(a_{\mathsf{P}_1}m, a_{([s^2],\mathsf{P}_1)}m\right)\left(a_{([r],01)}, a_{([r],01)}e\right)\left(a_{([r],01)}m, a_{([r],10)}f\right)\left(a_{([sr],01)}, a_{([sr],01)}e\right) \\
& \left(a_{([sr],10)}, a_{([sr],10)}e\right)\left(a_{([s^2],\mathsf{P}_2)}m, a_{\mathsf{P}_2}m\right)\left(a_{([sr],01)}f, a_{([sr],10)}m\right)\left(a_{([r],10)}, a_{([r],10)}e\right) \\
& \left(a_{([r],10)}m, a_{([r],01)}f\right)\left(a_{([s^2],\mathsf{E})}, a_{([s^2],\mathsf{E})}e\right).
\end{aligned}
\tag{130}
$$

We use notation that follows the conventional notation for permutations: $(x, y)$ means that translation acts as $x \to y$ and $y \to x$, and if an anyon does not appear in any parenthesis of (130) then $T$ acts on it trivially.

Having discussed the SymTFT's anyons and its enrichment by lattice translations, we can now discuss quantum phases characterized by the projective $\mathrm{Rep}(D_8) \times \mathbb{Z}_2$ symmetry captured by this SymTFT. As mentioned above, we are interested in possible SPTs phases, which correspond to Lagrangian condensable algebras that have trivial overlap with the Lagrangian algebra defining the symmetry boundary [81, 138–140]. We numerically find that there are 478 condensable algebras of $D_8 \times \mathbb{Z}_2$ gauge theory,[17] 58 of which are Lagrangian algebras. The symmetry boundary for the projective $\mathrm{Rep}(D_8) \times \mathbb{Z}_2$ symmetry with translations corresponds to the Lagrangian algebra

$$\mathcal{L} = 1 \oplus a_{[s]} \oplus a_{[s^2]} \oplus a_{[r]} \oplus a_{[sr]} \oplus e \oplus a_{[s]}e \oplus a_{[s^2]}e \oplus a_{[r]}e \oplus a_{[sr]}e. \tag{131}$$

---

[17]We are incredibly thankful to Xiao-Gang Wen for his generous help with numerics and for providing us with computational resources.

Indeed, this Lagrangian algebra agrees with the boundary stabilizers (113), which correspond to the rough and smooth boundaries for the $Z(D_8) = \mathbb{Z}_2$ and $D_8$ qudits, respectively, and, therefore, condense $Z(D_8) = \mathbb{Z}_2$ electric charges and $D_8$ magnetic fluxes. There are 12 Lagrangian algebras that have trivial overlap with $\mathcal{L}$, which are given by

$$\mathcal{L}_1 = 1 \oplus a_\mathsf{E} e \oplus a_{\mathsf{P}_1} e \oplus a_{\mathsf{P}_3} e \oplus a_\mathsf{E} \oplus a_{[r]} m \oplus a_{([r],01)} m \oplus a_{([r],11)} f \oplus a_{([r],10)} f \oplus a_{\mathsf{P}_2} \, ,$$

$$\mathcal{L}_2 = 1 \oplus a_\mathsf{E} e \oplus a_{\mathsf{P}_1} e \oplus a_\mathsf{E} \oplus a_{\mathsf{P}_3} \oplus a_{([sr],01)} m \oplus a_{[sr]} m \oplus a_{\mathsf{P}_2} e \oplus a_{([sr],10)} f \oplus a_{([sr],11)} f \, ,$$

$$\mathcal{L}_3 = 1 \oplus a_{([s^2],\mathsf{P}_2)} e \oplus a_{([s^2],\mathsf{P}_1)} e \oplus a_\mathsf{E} m \oplus a_{\mathsf{P}_3} \oplus a_{([sr],01)} \oplus a_{([sr],01)} e \oplus a_{[sr]} m \oplus a_{([s^2],\mathsf{E})} f \oplus a_{([sr],11)} f \, ,$$

$$\mathcal{L}_4 = 1 \oplus a_{([s^2],\mathsf{P}_2)} e \oplus a_{\mathsf{P}_3} e \oplus a_{([r],01)} m \oplus a_{([s^2],\mathsf{P}_1)} \oplus a_{([sr],01)} \oplus a_{([sr],01)} e \oplus a_{[s]} m \oplus a_{([r],10)} f \oplus a_{([s],\omega^2)} f \, ,$$

$$\mathcal{L}_5 = 1 \oplus a_{([s^2],\mathsf{P}_3)} e \oplus a_{([s^2],\mathsf{P}_1)} e \oplus a_\mathsf{E} m \oplus a_{[r]} m \oplus a_{([r],01)} \oplus a_{([r],01)} e \oplus a_{([r],11)} f \oplus a_{([s^2],\mathsf{E})} f \oplus a_{\mathsf{P}_2} \, ,$$

$$\mathcal{L}_6 = 1 \oplus a_{([s^2],\mathsf{P}_3)} e \oplus a_{([r],01)} \oplus a_{([r],01)} e \oplus a_{([s^2],\mathsf{P}_1)} \oplus a_{([sr],01)} m \oplus a_{[s]} m \oplus a_{\mathsf{P}_2} e \oplus a_{([sr],10)} f \oplus a_{([s],\omega^2)} f \, ,$$

$$\mathcal{L}_7 = 1 \oplus 2 a_\mathsf{E} e \oplus a_{[s^2]} m \oplus a_{([s^2],\mathsf{P}_2)} m \oplus a_{([s^2],\mathsf{P}_3)} m \oplus a_{\mathsf{P}_1} \oplus a_{\mathsf{P}_3} \oplus 2 a_{([s^2],\mathsf{E})} f \oplus a_{([s^2],\mathsf{P}_1)} m \oplus a_{\mathsf{P}_2} \, ,$$

$$\mathcal{L}_8 = 1 \oplus a_{\mathsf{P}_1} \oplus a_{\mathsf{P}_3} m \oplus 2 a_\mathsf{E} \oplus 2 a_\mathsf{E} m \oplus a_{\mathsf{P}_1} m \oplus a_{\mathsf{P}_3} \oplus a_{\mathsf{P}_2} m \oplus m \oplus a_{\mathsf{P}_2} \, ,$$

$$\mathcal{L}_9 = 1 \oplus a_{[s^2]} m \oplus a_{([s^2],\mathsf{P}_2)} \oplus a_{([s^2],\mathsf{P}_3)} m \oplus a_{\mathsf{P}_1} m \oplus a_{\mathsf{P}_3} \oplus a_{([s^2],\mathsf{P}_1)} \oplus 2 a_{([sr],01)} e \oplus 2 a_{([sr],10)} f \oplus a_{\mathsf{P}_2} m \, ,$$

$$\mathcal{L}_{10} = 1 \oplus a_{([s^2],\mathsf{P}_2)} \oplus a_{([s^2],\mathsf{P}_2)} m \oplus a_{\mathsf{P}_3} m \oplus a_{\mathsf{P}_3} \oplus a_{([s^2],\mathsf{P}_1)} \oplus 2 a_{([sr],01)} \oplus 2 a_{([sr],01)} m \oplus a_{([s^2],\mathsf{P}_1)} m \oplus m \, ,$$

$$\mathcal{L}_{11} = 1 \oplus a_{[s^2]} m \oplus a_{([s^2],\mathsf{P}_2)} m \oplus a_{([s^2],\mathsf{P}_3)} \oplus a_{\mathsf{P}_3} m \oplus a_{\mathsf{P}_1} m \oplus 2 a_{([r],01)} e \oplus a_{([s^2],\mathsf{P}_1)} \oplus 2 a_{([r],10)} f \oplus a_{\mathsf{P}_2} \, ,$$

$$\mathcal{L}_{12} = 1 \oplus a_{([s^2],\mathsf{P}_3)} \oplus a_{([s^2],\mathsf{P}_3)} m \oplus 2 a_{([r],01)} \oplus 2 a_{([r],01)} m \oplus a_{([s^2],\mathsf{P}_1)} \oplus a_{([s^2],\mathsf{P}_1)} m \oplus a_{\mathsf{P}_2} m \oplus m \oplus a_{\mathsf{P}_2} \, .$$

However, not all 12 correspond to $\mathrm{Rep}(D_8) \times \mathbb{Z}_2$ weak SPTs. Indeed, the translation anyon automorphism (130) acts on these Lagrangian algebras as

$$T : \; (\mathcal{L}_1)(\mathcal{L}_2)(\mathcal{L}_3)(\mathcal{L}_4)(\mathcal{L}_5)(\mathcal{L}_6)(\mathcal{L}_7, \mathcal{L}_8)(\mathcal{L}_9, \mathcal{L}_{10})(\mathcal{L}_{11}, \mathcal{L}_{12}) . \tag{132}$$

Therefore, the SPTs associated with $\mathcal{L}_i$ with $i = 6, 7, \cdots 12$ are not translation symmetric and, therefore, are not weak SPTs. If there was an LSM anomaly, all Lagrangian algebras corresponding to SPTs would transform non-trivially under translations, reflecting how LSM anomalies prevent SPTs only when translations are preserved [121]. Here, however, there are 6 Lagrangian algebras corresponding to SPTs that are translation invariant, i.e., $\mathcal{L}_i$ with $i = 1, 2, \cdots 6$, confirming how the projective algebra for $G = D_8$ does not give rise to LSM anomalies. According to the crystalline equivalence principle [59], these $\mathrm{Rep}(D_8) \times \mathbb{Z}_2$ weak SPTs are in one-to-one correspondence to SPTs protected by an internal $\mathrm{Rep}(D_8) \times \mathbb{Z}_2 \times \mathbb{Z}_2$ symmetry. In Appendix C, we construct the TQFTs for these SPTs.

# 7 Discussion

In this paper, we studied the consequences of projective algebras involving non-invertible symmetries and lattice translations of a $G$-based XY model on the entanglement structure of the ground states in $1 + 1$D. In particular, we have shown that this projective algebra constrains the entanglement structure by ruling out trivial product states, i.e., the ground states of local symmetric Hamiltonians must be either degenerate through spontaneous symmetry breaking, gapless, or a weak SPT with non-trivial entanglement. From this point of view, constraints due to projective non-invertible symmetries unify the ordinary LSM anomalies of invertible symmetries that rule out short-range entangled states and the SPT-LSM theorems that demand any short-range entangled state to be a non-trivial SPT. We further explored the duality web obtained by gauging internal sub-symmetries and showed the interplay of dual internal and translation symmetries in the form of non-Abelian dipole, non-invertible dipole, or non-invertible translation symmetries. We complemented this duality web by studying the SymTFTs with an appropriate enrichment by translations that capture the constraints due to the projective algebra.

There are several promising future directions that follow from this work. First, in this paper, we have shown that the projective algebra studied is sometimes compatible with non-trivial SPT states. This implies that in such cases, there is a way to gauge the $\text{Rep}(G) \times Z(G)$ symmetry without breaking lattice translations, and it would be interesting to explore these alternative gauging procedures. Furthermore, Hamiltonians in 1+1D with LSM anomalies have been a fruitful arena to explore deconfined quantum critical phase transitions [51, 141–145]. Because of the constraints we derived, the phase diagram of $G$-based XY model (12) does not contain a disordered phase with trivial entanglement. Hence, it is an interesting future direction to analyze such a phase diagram where we expect to find continuous transitions that feature deconfined criticality or topological phase transitions. Another natural generalization of our discussion is to consider non-invertible $\text{Rep}(H)$ symmetry with $H$ being a Hopf algebra that participates in a similar projective algebra. The techniques we developed for inserting $\text{Rep}(G)$ defects and gauging the $\text{Rep}(G)$ symmetry can be generalized to other non-invertible symmetries implemented by MPOs. Gauging or analyzing twisted boundary conditions of such symmetries can lead to new dualities or constraints on low-energy properties. Finally, an exciting direction is the generalizations of translation enriched SymTFT description presented in Section 6. In a related upcoming work [121], we showcase a detailed study of such spacetime symmetry-enriched SymTFTs for LSM anomalies in 1+1D involving finite Abelian groups.

## Acknowledgments

We thank Guilherme Delfino for collaboration on related project [95] and Arkya Chatterjee, Laurens Lootens, Max Metlitski, Nathan Seiberg, Shu-Heng Shao, and Xiao-Gang Wen for helpful discussions. We are especially grateful to Xiao-Gang Wen for his help with numerically calculating condensable algebras of quantum double models and for providing access to his computational resources.

**Funding information** S.D.P. is supported by the National Science Foundation Graduate Research Fellowship under Grant No. 2141064. H.T.L. is supported by the U.S. Department of Energy, Office of Science, Office of High Energy Physics of U.S. Department of Energy under grant Contract Number DE-SC0012567 (High Energy Theory research) and by the Packard Foundation award for Quantum Black Holes from Quantum Computation and Holography. Ö.M.A. is supported by Swiss National Science Foundation (SNSF) under Grant No. P500PT-214429 and National Science Foundation (NSF) DMR-2022428.

## A The Dihedral group of order eight $D_8$

In this appendix, we briefly summarize aspects of the order 8 dihedral group

$$D_8 \simeq \mathbb{Z}_2 \ltimes \mathbb{Z}_4 = \left\langle r\, s \mid r^2 = s^4 = 1,\; r\, s\, r = s^3 \right\rangle, \tag{A.1}$$

used in the main text. The group $D_8$ is generated by the elements $r$ and $s$. Geometrically, it is the symmetry group of a square with $s$ performing a 90 degree rotation and $r$ a reflection. Its multiplication table is shown in Table 2.

The group $D_8$ has five irreps, four of which are one-dimensional and one is two-dimensional. They are:

1. the one-dimensional trivial representation $\mathbf{1}\colon D_8 \to U(1)$, where

$$\mathbf{1}(r) = 1, \qquad \mathbf{1}(s) = 1. \tag{A.2}$$

Table 2: Multiplication table of $D_8$ where row elements multiply column elements from the left.

| $\bullet$ | $1$ | $s$ | $s^2$ | $s^3$ | $r$ | $sr$ | $s^2r$ | $s^3r$ |
|---|---|---|---|---|---|---|---|---|
| $1$ | $1$ | $s$ | $s^2$ | $s^3$ | $r$ | $sr$ | $s^2r$ | $s^3r$ |
| $s$ | $s$ | $s^2$ | $s^3$ | $1$ | $sr$ | $s^2r$ | $s^3r$ | $r$ |
| $s^2$ | $s^2$ | $s^3$ | $1$ | $s$ | $s^2r$ | $s^3r$ | $r$ | $sr$ |
| $s^3$ | $s^3$ | $1$ | $s$ | $s^2$ | $s^3r$ | $r$ | $sr$ | $s^2r$ |
| $r$ | $r$ | $s^3r$ | $s^2r$ | $sr$ | $1$ | $s^3$ | $s^2$ | $s$ |
| $sr$ | $sr$ | $r$ | $s^3r$ | $s^2r$ | $s$ | $1$ | $s^3$ | $s^2$ |
| $s^2r$ | $s^2r$ | $sr$ | $r$ | $s^3r$ | $s^2$ | $s$ | $1$ | $s^3$ |
| $s^3r$ | $s^3r$ | $s^2r$ | $sr$ | $r$ | $s^3$ | $s^2$ | $s$ | $1$ |

2. The first one-dimensional sign representation $\mathsf{P}_1 \colon D_8 \to U(1)$, where

$$\mathsf{P}_1(r) = -1\,, \qquad \mathsf{P}_1(s) = 1\,. \tag{A.3}$$

3. The second one-dimensional sign representation $\mathsf{P}_2 \colon D_8 \to U(1)$, where

$$\mathsf{P}_2(r) = 1\,, \qquad \mathsf{P}_2(s) = -1\,. \tag{A.4}$$

4. The third one-dimensional sign representation $\mathsf{P}_3 \colon D_8 \to U(1)$, where

$$\mathsf{P}_3(r) = -1\,, \qquad \mathsf{P}_3(s) = -1\,. \tag{A.5}$$

5. The two-dimensional standard representation $\mathsf{E} \colon D_8 \to \mathrm{GL}(2, \mathbb{C})$, where

$$\mathsf{E}(r) = \begin{pmatrix} 1 & 0 \\ 0 & -1 \end{pmatrix}, \qquad \mathsf{E}(s) = \begin{pmatrix} 0 & -1 \\ 1 & 0 \end{pmatrix}. \tag{A.6}$$

These irreps satisfy

$$\begin{aligned} \mathsf{P}_i \otimes \mathsf{P}_i &= \mathbf{1}\,, & \mathsf{P}_3 &= \mathsf{P}_1 \otimes \mathsf{P}_2\,, \\ \mathsf{P}_i \otimes \mathsf{E} &= \mathsf{E} \otimes \mathsf{P}_i = \mathsf{E}\,, & \mathsf{E} \otimes \mathsf{E} &= \mathbf{1} \oplus \mathsf{P}_1 \oplus \mathsf{P}_2 \oplus \mathsf{P}_3\,. \end{aligned} \tag{A.7}$$

The group $D_8$ is one of the smallest non-Abelian groups with a non-trivial center. Indeed, the conjugacy classes of $D_8$ are

$$\begin{aligned} [1] &= \{1\}\,, & [s^2] &= \{s^2\}\,, \\ [s] &= \{s, s^3\}\,, & [r] &= \{r, s^2r\}\,, & [sr] &= \{sr, s^3r\}\,. \end{aligned} \tag{A.8}$$

Since $[1]$ and $[s^2]$ are the only one-dimensional conjugacy classes, the center $Z(D_8)$ of $D_8$ is

$$Z(D_8) = \{1, s^2\} \simeq \mathbb{Z}_2\,. \tag{A.9}$$

# B $D_{2n}$-qudit operators as $\mathbb{Z}_n \times \mathbb{Z}_2$ clock and shift operators

In this appendix, we present the $D_{2n}$-qudit Pauli operators in terms of $\mathbb{Z}_n \times \mathbb{Z}_2$-qudit operators (i.e., $\mathbb{Z}_n$ clock and shift operators and the ordinary Pauli operators). $D_{2n}$ is the dihedral group of order $2n$ and can be presented as

$$D_{2n} \simeq \mathbb{Z}_2 \ltimes \mathbb{Z}_n = \left\langle r, s \mid r^2 = s^n = 1, \ r\, s\, r = s^{n-1} \right\rangle. \tag{B.1}$$

Because each $D_{2n}$-qudit $|g\rangle$, with $g \in D_{2n}$, is simply a $|D_{2n}|$-level quantum mechanical system, we can view a $D_{2n}$-qudit as a $\mathbb{Z}_n \times \mathbb{Z}_2$-qudit because $|D_{2n}| = |\mathbb{Z}_n \times \mathbb{Z}_2|$. We denote the $\mathbb{Z}_n \times \mathbb{Z}_2$-qudit state by $|a, b\rangle$, where $a = 0, 1, \cdots, n$ and $b = 0, 1$, and make the identification

$$|a, b\rangle \equiv \left| s^a\, r^b \right\rangle, \tag{B.2}$$

with $s^a\, r^b \in D_{2n}$. On a length $L$ chain, we label a many-body $D_{2n}$-qudit configuration by the pair of strings $|\boldsymbol{a}, \boldsymbol{b}\rangle$ such that the local state at site $j$ is labeled by the pair $\left| a_j, b_j \right\rangle$.

We now relate the $D_{2n}$ qudit operators to the clock and shift operators acting on $\mathbb{Z}_n \times \mathbb{Z}_2$-qudits. Consider the $\mathbb{Z}_n$ and $\mathbb{Z}_2$ clock and shift operators $\left\{ Z_j, X_j \right\}$ and $\left\{ \sigma_j^z, \sigma_j^x \right\}$ that satisfy the algebra

$$\begin{aligned}
Z_j X_k &= \omega_n^{\delta_{j,k}} X_k Z_j, & \sigma_j^z \sigma_k^x &= (-1)^{\delta_{j,k}} \sigma_k^x \sigma_j^z, \\
Z_j^n &= X_j^n = 1, & (\sigma_j^z)^2 &= (\sigma_j^x)^2 = 1,
\end{aligned} \tag{B.3a}$$

with $\omega_n = e^{2\pi i/n}$. These operators act on the basis $|\boldsymbol{a}, \boldsymbol{b}\rangle$ as

$$\begin{aligned}
Z_j |\boldsymbol{a}, \boldsymbol{b}\rangle &= \omega_n^{a_j} |\boldsymbol{a}, \boldsymbol{b}\rangle, & \sigma_j^z |\boldsymbol{a}, \boldsymbol{b}\rangle &= (-1)^{b_j} |\boldsymbol{a}, \boldsymbol{b}\rangle, \\
X_j |\boldsymbol{a}, \boldsymbol{b}\rangle &= \left| \boldsymbol{a} + \boldsymbol{\delta}^{(j)}, \boldsymbol{b} \right\rangle, & \sigma_j^x |\boldsymbol{a}, \boldsymbol{b}\rangle &= \left| \boldsymbol{a}, \boldsymbol{b} + \boldsymbol{\delta}^{(j)} \right\rangle,
\end{aligned} \tag{B.3b}$$

where $\boldsymbol{\delta}^{(j)}$ is a vector which is 1 at site $j$ and 0 at all other sites.

The $X$-type $D_{2n}$ qudit operators satisfy

$$[\overrightarrow{X}_j^{(r)}]^2 = [\overrightarrow{X}_j^{(s)}]^n = 1, \qquad \overrightarrow{X}_j^{(r)} \overrightarrow{X}_j^{(s)} \overrightarrow{X}_j^{(r)} = [\overrightarrow{X}_j^{(s)}]^{n-1}, \tag{B.4}$$

$$[\overleftarrow{X}_j^{(r)}]^2 = [\overleftarrow{X}_j^{(s)}]^n = 1, \qquad \overleftarrow{X}_j^{(r)} \overleftarrow{X}_j^{(s)} \overleftarrow{X}_j^{(r)} = [\overleftarrow{X}_j^{(s)}]^{n-1}, \tag{B.5}$$

$$\overleftarrow{X}_j^{(g)} \overrightarrow{X}_j^{(h)} = \overrightarrow{X}_j^{(h)} \overleftarrow{X}_j^{(g)}. \tag{B.6}$$

It is then straightforward to verify that a consistent decomposition of the $D_{2n}$-based $X$ operators in terms of the $\mathbb{Z}_n \times \mathbb{Z}_2$ shift operators that satisfy the above is

$$\overrightarrow{X}_j^{(r)} = \sigma_j^x C_j, \qquad \overrightarrow{X}_j^{(s)} = X_j, \qquad \overleftarrow{X}_j^{(r)} = \sigma_j^x, \qquad \overleftarrow{X}_j^{(s)} = X_j^{-\sigma_j^z}, \tag{B.7}$$

where $C_j$ is the local charge conjugation operator satisfying

$$C_j Z_j C_j^\dagger = Z_j^\dagger, \qquad C_j X_j C_j^\dagger = X_j^\dagger, \tag{B.8}$$

and $X_j^{-\sigma_j^z}$ is a shorthand notation for $\frac{1}{2}[(1 + \sigma_j^z) X_j^\dagger + (1 - \sigma_j^z) X_j]$. Indeed, acting these operators on a single $\mathbb{Z}_n \times \mathbb{Z}_2$ qudit state yields

$$\overrightarrow{X}^{(r)} |a, b\rangle = \sigma^x C |a, b\rangle = |-a, b+1\rangle, \tag{B.9}$$

$$\overrightarrow{X}^{(s)} |a, b\rangle = X |a, b\rangle = |a+1, b\rangle, \tag{B.10}$$

$$\overleftarrow{X}^{(r)} |a, b\rangle = \sigma^x |a, b\rangle = |a, b+1\rangle, \tag{B.11}$$

$$\overleftarrow{X}^{(s)} |a, b\rangle = X^{-\sigma^z} |a, b\rangle = \left| a - (-1)^b, b \right\rangle, \tag{B.12}$$

which correctly reproduce how these operators act on the $D_{2n}$-qudit state $\left|s^a r^b\right\rangle$.

The number of different group-based $Z$ operators depends on the degree $n$ of the dihedral group $D_{2n}$. When $n$ is odd, there are 2 one-dimensional irreps and $(n-1)/2$ two-dimensional irreps. The corresponding group-based $Z$ operators can be obtained from a particular set of representation matrices. For instance, in the only-non-trivial one-dimensional irrep P, the two generators $s$ and $r$ are represented by $+1$ and $-1$, respectively. Therefore, the corresponding group-based $Z$ operator is given by

$$Z_j^{(\text{P})} = \sigma_j^z. \tag{B.13a}$$

In the $(n-1)/2$ two-dimensional irreps $\text{E}_k$, with $k = 1, 2, \cdots, (n-1)/2$, the two generators are represented as

$$\text{E}_k(s) = \begin{pmatrix} \cos(2\pi k/n) & -\sin(2\pi k/n) \\ \sin(2\pi k/n) & \cos(2\pi k/n) \end{pmatrix}, \qquad \text{E}_k(r) = \begin{pmatrix} 1 & 0 \\ 0 & -1 \end{pmatrix}. \tag{B.13b}$$

Hence, the corresponding group-based $Z$ MPOs are

$$Z_j^{(\text{E}_k)} = \frac{1}{2}\begin{pmatrix} Z_j^k + Z_j^{-k} & i(Z_j^k - Z_j^{-k})\sigma_j^z \\ -i(Z_j^k - Z_j^{-k}) & (Z_j^k + Z_j^{-k})\sigma_j^z \end{pmatrix}. \tag{B.13c}$$

When $n$ is even, there are 4 one-dimensional irreps and $n/2-1$ two-dimensional irreps. The two-dimensional representation matrices are still given by (B.13b), but now $k = 1, 2, \cdots n/2-1$. We can find the group-based $Z$ operators in a similar manner as for odd $n$. The group-based $Z$ operator corresponding to the 3 non-trivial one-dimensional irreps are then

$$Z_j^{(\text{P}_1)} = \sigma_j^z, \qquad Z_j^{(\text{P}_2)} = Z_j^{n/2}, \qquad Z_j^{(\text{P}_3)} = Z_j^{n/2}\sigma_j^z, \tag{B.14}$$

while the $n/2-1$ group-based $Z$ operators corresponding to the two-dimensional irreps are still given by (B.13c) but where $k = 1, \cdots, n/2-1$.

## B.1 The case of $G = D_8$

Following the discussion from this Appendix, when $G = D_8$, we can express the $\text{Rep}(D_8) \times Z(D_8)$ symmetry operators in terms of $\mathbb{Z}_4 \times \mathbb{Z}_2$ clock and shift operators. In particular, using the general expressions (B.13c) and (B.14), the $\text{Rep}(D_8)$ symmetry operators can be written as

$$R_{\text{P}_1} = \prod_j \sigma_j^z, \qquad R_{\text{P}_2} = \prod_j Z_j^2, \qquad R_{\text{P}_3} \equiv R_{\text{P}_1} R_{\text{P}_2} = \prod_j Z_j^2 \sigma_j^z,$$
$$R_{\text{E}} = \frac{1}{2}\left(1 + R_{\text{P}_1}\right)\left(1 + R_{\text{P}_2}\right)\prod_j Z_j^{\prod_{k=1}^{j-1}\sigma_k^z}. \tag{B.15}$$

One verifies that these satisfy the operator algebra

$$R_{\text{P}_i} R_{\text{E}} = R_{\text{E}} R_{\text{P}_i} = R_{\text{E}}, \qquad R_{\text{E}} R_{\text{E}} = 1 + \sum_{i=1}^{3} R_{\text{P}_i}, \tag{B.16}$$

and are, in fact, $\text{Rep}(D_8)$ operators. Furthermore, the $Z(D_8) \cong \mathbb{Z}_2$ symmetry is generated by the unitary operator

$$U_{s^2} = \prod_j X_j^2, \tag{B.17}$$

which satisfies the expected algebra

$$U_{s^2} R_{\mathsf{P}_i} = R_{\mathsf{P}_i} U_{s^2}, \qquad U_{s^2} R_{\mathsf{E}} = (-1)^L R_{\mathsf{E}} U_{s^2}. \tag{B.18}$$

Using the $\mathbb{Z}_4 \times \mathbb{Z}_2$ clock and shift operators, the bond algebra of $\mathsf{Rep}(D_8) \times \mathbb{Z}_2$ symmetric operators is

$$\mathfrak{B}\left[\mathsf{Rep}(D_8) \times \mathbb{Z}_2\right] = \left\langle \sigma_j^z, \, Z_j^2, \, Z_j Z_{j+1}, \, \sigma_j^x C_{j+1} \sigma_{j+1}^x, \, X_j^{\sigma_j^z} X_{j+1}^\dagger \right\rangle. \tag{B.19}$$

Using the elements in this bond algebra, we can construct $\mathsf{Rep}(D_8) \times \mathbb{Z}_2$ symmetric SPT Hamiltonians. For simplicity, let us consider one such example, namely

$$H_{\mathsf{SPT}} = -\sum_j \sigma_j^x C_{j+1} \sigma_{j+1}^x + \sum_j \frac{1}{4}\left(Z_j - Z_j^\dagger\right)\sigma_j^z\left(Z_{j+1} - Z_{j+1}^\dagger\right), \tag{B.20}$$

which is $\mathsf{Rep}(D_8) \times \mathbb{Z}_2$ symmetric, consists of pairwise commuting terms and has a non-degenerate gapped ground state. To see this, we first note that the second term squares to

$$\left[\frac{1}{4}\left(Z_j - Z_j^\dagger\right)\sigma_j^z\left(Z_{j+1} - Z_{j+1}^\dagger\right)\right]^2 = \frac{1}{4}\left(Z_j^2 - 1\right)\left(Z_{j+1}^2 - 1\right), \tag{B.21}$$

which vanishes in the $Z_j^2 = 1$ subspace. Therefore, the second term in the Hamiltonian (B.20) can take negative values only in the $Z_j^2 = -1$ subspace, which restricts the low-energy subspace of $H_{\mathsf{SPT}}$ to configurations with $a_j = 1, 3$. In this subspace, the charge conjugation operator $C_j$ exchanges the $a_j = 1$ state with the $a_j = 3$ state and, hence, acts like a Pauli-$X$ operator $\tau^x$ of the $a_j = 1, 3$ two-level system. Furthermore, $\frac{1}{2}(Z_j - Z_j^\dagger)$ acts like i times Pauli-$Z$ operator $\tau^z$. Therefore, in the $Z_j^2 = -1$ subspace, the Hamiltonian (B.20) is nothing but a $\mathbb{Z}_2$ cluster Hamiltonian

$$H_{\mathsf{cluster}} = -\sum_j \sigma_j^x \tau_{j+1}^x \sigma_{j+1}^x - \sum_j \tau_j^z \sigma_j^z \tau_{j+1}^z. \tag{B.22}$$

Its ground state is then

$$|\psi_{\mathsf{GS}}\rangle = \sum_{\{\alpha_j, \varphi_j = 0,1\}} (-1)^{\sum_j \alpha_j(\varphi_j + \varphi_{j-1})} \bigotimes_j \left|2\alpha_j + 1, \varphi_j\right\rangle, \tag{B.23}$$

where $\left|2\alpha_j + 1, \varphi_j\right\rangle$ are the basis states on which the $\tau^z$ and $\sigma_j^x$ operators are diagonalized, i.e.,

$$\begin{aligned} \tau^z \left|2\alpha_j + 1, \varphi_j\right\rangle &= (-1)^{\alpha_j}\left|2\alpha_j + 1, \varphi_j\right\rangle, \\ \sigma_j^x \left|2\alpha_j + 1, \varphi_j\right\rangle &= (-1)^{\varphi_j}\left|2\alpha_j + 1, \varphi_j\right\rangle. \end{aligned} \tag{B.24}$$

These basis states are related to the ones in (B.2) by the following basis transformation

$$\left|2\alpha_j + 1, \varphi_j\right\rangle = \frac{1}{\sqrt{2}}\left(\left|a_j = 2\alpha_j + 1, b_j = 0\right\rangle + (-1)^{\varphi_j}\left|a_j = 2\alpha_j + 1, b_j = 1\right\rangle\right). \tag{B.25}$$

Being non-degenerate and gapped, the ground state (B.23) is invariant under $\mathsf{Rep}(D_8) \times \mathbb{Z}_2$ symmetry as well as translations. In particular, it satisfies

$$\begin{aligned} U_{s^2} |\psi_{\mathsf{GS}}\rangle &= |\psi_{\mathsf{GS}}\rangle, & T |\psi_{\mathsf{GS}}\rangle &= |\psi_{\mathsf{GS}}\rangle, \\ R_{\mathsf{P}_1} |\psi_{\mathsf{GS}}\rangle &= |\psi_{\mathsf{GS}}\rangle, & R_{\mathsf{P}_2} |\psi_{\mathsf{GS}}\rangle &= (-1)^L |\psi_{\mathsf{GS}}\rangle, \\ R_{\mathsf{E}} |\psi_{\mathsf{GS}}\rangle &= (1 + (-1)^L)|\psi_{\mathsf{GS}}\rangle. \end{aligned} \tag{B.26}$$

Therefore, the ground state (B.23) can carry $\mathrm{Rep}(D_8)$ symmetry charge depending on the system size. In particular, for even $L$, the ground state is symmetric, but for odd $L$, it carries $\mathrm{Rep}(D_8)$ symmetry charge and, consequentially, is annihilated by the non-invertible $R_\mathsf{E}$ operator.

We close our discussion by considering the ground states of the Hamiltonian (B.20) in the presence of a $\mathsf{E}$ $\mathrm{Rep}(D_8)$ symmetry defect. Following Section 3.2.2, after inserting the $\mathsf{E}$ symmetry defect, the defect-free Hilbert space $\mathcal{H} \cong \mathbb{C}^{8^L}$ is enlarged to $\mathbb{C}^{d_\mathsf{E}} \otimes \mathcal{H} \cong \mathbb{C}^2 \otimes \mathcal{H}$, and the Hamiltonian (B.20) is modified to

$$H_{\mathrm{SPT}}^\mathsf{E} = -\sum_{j=1}^{L-1} \sigma_j^x C_{j+1} \sigma_{j+1}^x - \widehat{\mathsf{E}}(r)\, \sigma_L^x C_1 \sigma_1^x + \sum_j \frac{1}{4}\left(Z_j - Z_j^\dagger\right) \sigma_j^z \left(Z_{j+1} - Z_{j+1}^\dagger\right), \quad \text{(B.27)}$$

where we used the fact that $C_1 \sigma_1^x$ corresponds to $\overrightarrow{X}_1^{(r)}$ operator. Here, the operator $\widehat{\mathsf{E}}(r)$ acts on the two-dimensional defect Hilbert space $\mathbb{C}^{d_\mathsf{E}} = \mathbb{C}^2$ just like a Pauli-$Z$ matrix (recall Eq. (A.6)). Hence, there are two degenerate ground states of twisted Hamiltonian (B.27) depending on the defect state. Let us denote by $|\pm 1\rangle$ the eigenstate of $\widehat{\mathsf{E}}(r)$ with eigenvalue $\pm 1$. Then, the two degenerate ground states are

$$
\begin{aligned}
\left|\psi_{\mathrm{GS}}^+\right\rangle &= |+1\rangle \otimes \left( \sum_{\{\alpha_j, \varphi_j = 0,1\}} (-1)^{\sum_j \alpha_j(\varphi_j + \varphi_{j-1})} \bigotimes_j \left|2\alpha_j + 1, \varphi_j\right\rangle \right), \\
\left|\psi_{\mathrm{GS}}^-\right\rangle &= |-1\rangle \otimes \left( \sum_{\{\alpha_j, \varphi_j = 0,1\}} (-1)^{\alpha_1}(-1)^{\sum_j \alpha_j(\varphi_j + \varphi_{j-1})} \bigotimes_j \left|2\alpha_j + 1, \varphi_j\right\rangle \right).
\end{aligned}
\quad \text{(B.28)}
$$

These two states transform under the $\mathrm{Rep}(G)$ symmetry in the same manner as the untwisted SPT ground state (B.23)

$$R_{\mathsf{P}_1}\left|\psi_{\mathrm{GS}}^\pm\right\rangle = \left|\psi_{\mathrm{GS}}^\pm\right\rangle, \quad R_{\mathsf{P}_2}\left|\psi_{\mathrm{GS}}^\pm\right\rangle = (-1)^L \left|\psi_{\mathrm{GS}}^\pm\right\rangle, \quad R_\mathsf{E}\left|\psi_{\mathrm{GS}}^\pm\right\rangle = (1 + (-1)^L)\left|\psi_{\mathrm{GS}}^\pm\right\rangle. \quad \text{(B.29a)}$$

However, they differ under the action of $\mathbb{Z}_2$ center and twisted translation symmetries. In particular, we have

$$U_{s^2}\left|\psi_{\mathrm{GS}}^+\right\rangle = +\left|\psi_{\mathrm{GS}}^+\right\rangle, \qquad U_{s^2}\left|\psi_{\mathrm{GS}}^-\right\rangle = -\left|\psi_{\mathrm{GS}}^-\right\rangle, \quad \text{(B.29b)}$$

$$T_\mathsf{E}\left|\psi_{\mathrm{GS}}^+\right\rangle = \left|\psi_{\mathrm{GS}}^-\right\rangle, \qquad T_\mathsf{E}\left|\psi_{\mathrm{GS}}^-\right\rangle = \left|\psi_{\mathrm{GS}}^+\right\rangle. \quad \text{(B.29c)}$$

Therefore, we find that when a $\mathsf{E}$ defect is inserted, the twisted translation $T_\mathsf{E}$ acts like a Pauli-$X$ operator on the low-energy states while $U_{s^2}$ acts like a Pauli-$Z$ operator. Therefore, they anti-commute on the ground state manifold. This is nothing but the result of the projective algebra (46), and shows how the projectivity gives rise to degeneracies involving states in the defect Hilbert space.

## C  TQFTs for $(\mathrm{Rep}(D_8) \times \mathbb{Z}_2 \times \mathbb{Z}_2)$-SPTs

In this appendix, we discuss the TQFTs for the SPTs protected by a $\mathcal{C} = \mathrm{Rep}(D_8) \times \mathbb{Z}_2 \times \mathbb{Z}_2$ symmetry that obeys a projective algebra similar to the ones in Eqs. (42), (45), and (46), but with the $\mathbb{Z}$ translation symmetry being replaced by an internal $\mathbb{Z}_2$ symmetry. The $\mathcal{C}$ symmetry defects are labeled by $(a, b) \in \mathbb{Z}_2 \times \mathbb{Z}_2$ and irreps $\Gamma$ of $D_8$ and denoted as[18]

$$\hat{U}^a \hat{T}^b \hat{R}_\Gamma, \quad \text{(C.1)}$$

---

[18]We put a hat on symmetry defects to distinguish them from the symmetry operators.

where $\hat{U}$, $\hat{T}$ are the symmetry defects associated to the generators of the $\mathbb{Z}_2 \times \mathbb{Z}$ sub-symmetry while $\hat{R}_\Gamma$ are the symmetry defects of the $\text{Rep}(D_8)$ sub-symmetry. The projectivity of the $\mathcal{C}$ symmetry manifests as the following projective algebras in various defect Hilbert spaces:

$$
\begin{aligned}
\hat{T}: \quad & R_\Gamma\, U = e^{i\phi_\Gamma(s^2)} U R_\Gamma\,, \\
\hat{U}: \quad & R_\Gamma\, T = e^{i\phi_\Gamma(s^2)} T R_\Gamma\,, \\
\hat{R}_\Gamma: \quad & T\, U = -U\, T\,,
\end{aligned}
\tag{C.2}
$$

where we follow the convention in Appendix A for the $D_8$ group. According to the crystalline equivalence principle [59], the weak SPTs protected by the projective $\text{Rep}(D_8) \times \mathbb{Z}_2$ and $\mathbb{Z}$ translation symmetry, discussed in Section 4.4, are in a one-to-one correspondence to the $\mathcal{C}$-SPTs if we interpret the Hamiltonian on a periodic spin chain of even (odd) length $L$ as the defect-free ($\hat{T}$ defect) Hamiltonian.

In what follows, we provide a classification of $\mathcal{C}$-SPTs. We do not claim our classification to be complete, and there may be more SPTs than we find. We start by recalling that in a $1+1$D SPT, all symmetry defect lines can end on some local topological operators. The symmetry actions on these local operators characterize an SPT. By state-operator correspondence, this is equivalent to the symmetry actions on the defect Hilbert spaces. In what follows, we use this to classify $\mathcal{C}$-SPTs, and the table below summarizes the data for the 12 $\mathcal{C}$-SPTs we will find:

|  | $R_{\text{P}_2}$ | $R_{\text{P}_3}$ | $R_{\text{E}}$ | $U$ | $T$ |
|---|---|---|---|---|---|
| $\hat{R}_{\text{P}_2}$ | $1$ | $1$ | $-$ | $x$ | $-x$ |
| $\hat{R}_{\text{P}_3}$ | $1$ | $1$ | $-$ | $-x$ | $x$ |
| $\hat{R}_{\text{E}}$ | $z_1 \mathbb{1}$ | $z_2 \mathbb{1}$ | $-$ | $\sigma^x$ | $\sigma^z$ |
| $\hat{U}$ | $x$ | $-x$ | $0$ | $1$ | $y$ |
| $\hat{T}$ | $-x$ | $x$ | $0$ | $y$ | $1$ |

In this table, the variables $x = \pm 1$, $y = \pm 1$, and $(z_1, z_2) = (1, 1), (1, -1), (-1, 1)$ characterize the 12 distinct $\mathcal{C}$-SPTs we find. Its entries represent the action of the symmetry operator from the corresponding column on the defect Hilbert space associated with the symmetry defect from the corresponding row. For example, $U$ acts on the one-dimensional $\hat{R}_{\text{P}_2}$ defect Hilbert space as a phase $x = \pm 1$ and the two-dimensional $\hat{R}_{\text{E}}$ defect Hilbert space as the $\sigma^x$ operator permuting the two states. Here, we did not list the action of $R_{\text{E}}$ on the $\hat{R}_{\text{P}_1}$, $\hat{R}_{\text{P}_2}$, or $\hat{R}_{\text{E}}$ defect Hilbert spaces because, in this case, the symmetry operator can map across different defect Hilbert spaces, which complicates the symmetry action. We also did not list the action of $R_{\text{P}_1}$ since it can be inferred from the action of $R_{\text{P}_2}$ and $R_{\text{P}_3}$ using the fusion rule $R_{\text{P}_2} \times R_{\text{P}_3} = R_{\text{P}_1}$.[19] One can check that the symmetry actions in this table are compatible with the projective algebras (C.2).

The variables $x$, $y$, and $(z_1, z_2)$ in the table can be understood as follows. The variable $y = \pm 1$ affects only the action of $U$ on the $\hat{T}$ defect Hilbert space and $T$ on the $\hat{U}$ defect Hilbert space, and corresponds to the two distinct $\mathbb{Z}_2 \times \mathbb{Z}_2$ SPTs. Similarly, the variables $(z_1, z_2) = (1, 1), (1, -1), (-1, 1)$ affect only the action of the $\text{Rep}(D_8)$ symmetry operators on the $\text{Rep}(D_8)$ defect Hilbert space. They correspond to the three distinct $\text{Rep}(D_8)$ SPTs [74, 109, 110, 112, 146]. Lastly, the variable $x = \pm 1$ affects the action of the $\mathbb{Z}_2 \times \mathbb{Z}_2$ symmetry, generated by $U$ and $T$, on the $\text{Rep}(D_8)$ defect Hilbert space and vice versa. It corresponds to different spontaneous symmetry breaking (SSB) patterns of the dual $G = (D_8 \times \mathbb{Z}_2) \rtimes \mathbb{Z}_2$[20] symmetry after gauging the $\text{Rep}(D_8)$ sub-symmetry.

---

[19]Note that when acting on $\hat{R}_{\text{E}}$ defect Hilbert space $R_{\text{P}_1} = -R_{\text{P}_2} \times R_{\text{P}_3}$.

[20]In GAP, this is SmallGroup(32, 49).

In the remainder of this appendix, we show how $x = \pm 1$ is related to the SSB patterns of the dual $G$ symmetry. We denote the $G$ symmetry defects by

$$L(u^a t^b s^c r^d), \tag{C.3}$$

where $a, b, c, d \in \mathbb{Z}_2 \times \mathbb{Z}_2 \times \mathbb{Z}_4 \times \mathbb{Z}_2$ and $u, t, s$, and $r$ are the generators of the $G$ symmetry group, satisfying

$$s^4 = r^2 = u^2 = t^2 = 1, \quad rs = s^3 r, \quad tu = us^2 t, \tag{C.4}$$

where trivial commutation relations are omitted.

The $\mathcal{C}$-SPT with $(x, y, z_1, z_2) = (1, 1, 1, 1)$ after gauging the $\mathrm{Rep}(D_8)$ sub-symmetry is dual to the SSB phase with SSB pattern

$$G = (D_8 \times \mathbb{Z}_2) \times \mathbb{Z}_2 \xrightarrow{\text{ssb}} H = \mathbb{Z}_2 \times \mathbb{Z}_2 = \langle ur, tsr \rangle. \tag{C.5}$$

There are 8 vacua in this SSB phase, labeled by the $G/H$ coset. By the state-operator correspondence, they are in one-to-one correspondence with topological local operators

$$v(s^c r^d) \longleftrightarrow |s^c r^d H\rangle. \tag{C.6}$$

The $G$ symmetry actions on these vacua are given by

$$L(g)|s^c r^d H\rangle = |g s^c r^d H\rangle. \tag{C.7}$$

For the symmetry's generators, these actions are

$$
\begin{aligned}
L(s) &: \ |s^c r^d H\rangle \to |s^{c+1} r^d H\rangle, \\
L(r) &: \ |s^c r^d H\rangle \to |s^{-c} r^{d+1} H\rangle, \\
L(u) &: \ |s^c r^d H\rangle \to |s^c r^{d+1} H\rangle, \\
L(t) &: \ |s^c r^d H\rangle \to |s^{c+(-1)^d} r^{d+1} H\rangle.
\end{aligned}
\tag{C.8}
$$

The symmetry defects can be decomposed into the direct sum of domain walls between these vacua. For example,

$$L(s) = \bigoplus_{c \in \mathbb{Z}_4, d = \mathbb{Z}_2} D(s^c r^d, s^{c+1} r^d), \tag{C.9}$$

where $D(g, h)$ denotes the domain wall between the $|gH\rangle$ and $|hH\rangle$ vaccua. The symmetry lines that contain domain walls between two identical vacua can end at the local operators corresponding to the vacuum. For instance, $L(s)$ is not preserved by any vacua, so it is not endable. On the other hand, the following symmetry lines are endable, and we list their end-point operators

$$
\begin{aligned}
\hat{L}(ur) &: \ v(e) \oplus v(s^2) \oplus v(r) \oplus v(s^2 r), \\
\hat{L}(tsr) &: \ v(e) \oplus v(s^2) \oplus v(sr) \oplus v(s^3 r), \\
\hat{L}(uts^3) &: \ v(e) \oplus v(s) \oplus v(s^2) \oplus v(s^3), \\
\hat{L}(us^2 r) &: \ v(s) \oplus v(s^3) \oplus v(sr) \oplus v(s^3 r), \\
\hat{L}(ts^3 r) &: \ v(s) \oplus v(s^3) \oplus v(r) \oplus v(s^2 r), \\
\hat{L}(uts) &: \ v(r) \oplus v(s^2 r) \oplus v(sr) \oplus v(s^3 r).
\end{aligned}
\tag{C.10}
$$

These end-point operators all transform trivially under the centralizer of the defect.

We now check that by gauging the $D_8 = \langle s, r \rangle$ sub-symmetry of $G$, we recover the $\mathcal{C}$-SPT with $(x, y, z_1, z_2) = (1, 1, 1, 1)$ as claimed above. To gauge the symmetry, we first reorganize the topological local operators into representations of the $D_8$

$$
\begin{aligned}
\mathcal{O}_1 &= v(e) + v(s) + v(s^2) + v(s^3) + v(r) + v(sr) + v(s^2r) + v(s^3r), \\
\mathcal{O}_{\mathsf{P}_1} &= v(e) + v(s) + v(s^2) + v(s^3) - v(r) - v(sr) - v(s^2r) - v(s^3r), \\
\mathcal{O}_{\mathsf{P}_2} &= v(e) - v(s) + v(s^2) - v(s^3) + v(r) - v(sr) + v(s^2r) - v(s^3r), \\
\mathcal{O}_{\mathsf{P}_3} &= v(e) - v(s) + v(s^2) - v(s^3) - v(r) + v(sr) - v(s^2r) + v(s^3r), \\
\mathcal{O}_{\mathsf{E}}^{(1)} &= \left\{ \begin{array}{c} \mathcal{O}_{\mathsf{E}}^{(1),1} \\ \mathcal{O}_{\mathsf{E}}^{(1),2} \end{array} \right\} = \left\{ \begin{array}{c} v(e) - v(s) - v(s^2) + v(s^3) + v(r) + v(sr) - v(s^2r) - v(s^3r) \\ v(e) + v(s) - v(s^2) - v(s^3) - v(r) + v(sr) + v(s^2r) - v(s^3r) \end{array} \right\}, \\
\mathcal{O}_{\mathsf{E}}^{(2)} &= \left\{ \begin{array}{c} \mathcal{O}_{\mathsf{E}}^{(2),1}, \\ \mathcal{O}_{\mathsf{E}}^{(2),2} \end{array} \right\} = \left\{ \begin{array}{c} v(e) + v(s) - v(s^2) - v(s^3) + v(r) - v(sr) - v(s^2r) + v(s^3r) \\ -v(e) + v(s) + v(s^2) - v(s^3) + v(r) + v(sr) - v(s^2r) - v(s^3r) \end{array} \right\}.
\end{aligned}
$$
(C.11)

After gauging the $D_8$ symmetry, the operators in non-trivial $D_8$ representations become topological defect operators that are attached to the Wilson lines of the same representations. These Wilson lines are the dual $\mathrm{Rep}(D_8)$ symmetry lines. Their end-point operators are

$$
\begin{aligned}
\hat{R}_{\mathsf{P}_{1,2,3}} &: \mathcal{O}_{\mathsf{P}_{1,2,3}}, \\
\hat{R}_{\mathsf{E}} &: \quad \mathcal{O}_{\mathsf{E}}^{(1)} \oplus \mathcal{O}_{\mathsf{E}}^{(2)}.
\end{aligned}
$$
(C.12)

Note that there are two end-point operators living at the end of the $\hat{R}_{\mathsf{E}}$ defect. The $\mathbb{Z}_2 \times \mathbb{Z}_2 = \langle u, t \rangle$ sub-symmetry acts trivially on the end-point operators $\mathcal{O}_{\mathsf{P}_{1,2,3}}$ but non-trivially on $\mathcal{O}_{\mathsf{E}}^{(1)}, \mathcal{O}_{\mathsf{E}}^{(2)}$ as

$$
\begin{aligned}
U &: \quad \mathcal{O}_{\mathsf{E}}^{(1)} \to \mathcal{O}_{\mathsf{E}}^{(2)}, \quad \mathcal{O}_{\mathsf{E}}^{(2)} \to \mathcal{O}_{\mathsf{E}}^{(1)}, \\
T &: \quad \mathcal{O}_{\mathsf{E}}^{(1)} \to \mathcal{O}_{\mathsf{E}}^{(1)}, \quad \mathcal{O}_{\mathsf{E}}^{(2)} \to -\mathcal{O}_{\mathsf{E}}^{(2)}.
\end{aligned}
$$
(C.13)

This implies that the $\mathbb{Z}_2 \times \mathbb{Z}_2 = \langle u, v \rangle$ sub-symmetry acts projectively in the $\hat{R}_E$ defect Hilbert space consistent with the projective algebra (C.2). Next, we discuss what the defect operators are mapped to after gauging. The symmetry defects related by fusion with the $D_8$ sub-symmetry defects are identified as the same line after gauging. If the end-point operators transform non-trivially under the $D_8$ symmetry, they must be attached to a Wilson line. Since none of the $D_8$ symmetry lines can end, we don't get any local operators from the twisted sector, and all defect operators remain defect operators after gauging. This means that the theory after gauging has a unique ground state and is an $\mathcal{C}$-SPT. The defect operators are

$$
\hat{U}^a \hat{T}^b \hat{R}_\Gamma : \{\mathcal{O}_\Gamma\},
$$
(C.14)

where $\{\mathcal{O}_\Gamma\}$ are operators transforming in the $\Gamma$ irrep of $D_8$. Notably, $\hat{U} = L(ur) \oplus L(us^2r)$, so its defect operators carry non-trivial $\mathrm{Rep}(G)$ charges associated with the conjugacy class $\{r, s^2r\}$. It implies that $R_\Gamma$ acts on the defect Hilbert space as a multiplication by the character $\chi_\Gamma(r)$. In particular, $R_{\mathsf{E}}$ annihilates the $\hat{U}$ defect Hilbert space and $R_{\mathsf{P}_1}, R_{\mathsf{P}_2}$ acts as $+1$ and $-1$, respectively, which corresponds to $x = 1$ in the table. Similarly, $R_{\mathsf{E}}$ also annihilates the $\hat{T}$ and $\hat{U}\hat{T}$ defect Hilbert spaces. This is consistent with the projective algebra (C.2).

Following a similar analysis, one can show that the $\mathcal{C}$-SPT with $(x, y, z_1, z_2) = (-1, 1, 1, 1)$ after gauging the $\mathrm{Rep}(D_8)$ sub-symmetry is dual to the SSB phase of the following SSB pattern

$$
G = (D_8 \times \mathbb{Z}_2) \times \mathbb{Z}_2 \to H = \mathbb{Z}_2 \times \mathbb{Z}_2 = \langle usr, tr \rangle.
$$
(C.15)

Indeed, after gauging, the $D_8 = \langle s, r \rangle$ sub-symmetry of this SSB phase, we have $\hat{U} = L(usr) \oplus L(us^3r)$, whose defect operators carry non-trivial $\mathrm{Rep}(G)$ charges associated with the conjugacy class $\{sr, s^3r\}$. It implies that $R_\Gamma$ acts on the defect Hilbert space as a multiplication by the character $\chi_\Gamma(sr)$, which matches with the $x = -1$ symmetry action.

# D Gauging $G$ and $\mathrm{Rep}(G)$ symmetries: Bond algebras and gauge fixing

In Section 5 of the main text, we explore parts of the $G$-based XY model's gauging web by gauging its various invertible and non-invertible sub-symmetries and dual sub-symmetries. In this appendix, we discuss in detail various gauging procedures for $G$ and $\mathrm{Rep}(G)$ symmetries in $1+1$D quantum Hamiltonian lattice models of $G$-qudits. Furthermore, to derive results intrinsic to the symmetry and independent from any particular Hamiltonian model, we implement these gauging procedures at the level of the algebra of symmetric operators called the bond algebra [147, 148].

## D.1 Gauging $G$ symmetry

We first review how to gauge a finite $G$ symmetry in a $G$-qudit quantum spin chain. Without a loss of generality, consider a $G$ symmetry represented by the unitary operators

$$U_g = \prod_j \overrightarrow{X}_j^{(g)}. \tag{D.1}$$

The bond algebra for the $G$ symmetry represented by (D.1) is[21]

$$\mathfrak{B}[G] = \left\langle \overleftarrow{X}_j^{(g)}, \ [Z_j^{(\overline{\Gamma})} Z_{j+1}^{(\Gamma)}]_{\alpha\beta}, \ \cdots \right\rangle, \tag{D.2}$$

where $\overline{\Gamma}(g) \equiv \Gamma(g)^\dagger = \Gamma(\bar{g})$. The power of the bond algebra is that it contains all $G$-symmetric operators and, hence, any operator that can appear in a $G$-symmetric Hamiltonian. Therefore, any statement deduced from $\mathfrak{B}[G]$ applies to all $G$-symmetric Hamiltonians.

To gauge the $G$ symmetry, we introduce $G$-qudits on each link $\langle j, j+1 \rangle$, acted on by the group-based Pauli operators $\overrightarrow{\mathcal{X}}_{j,j+1}^{(g)}$, $\overleftarrow{\mathcal{X}}_{j,j+1}^{(g)}$, and $\mathcal{Z}_{j,j+1}^{(\Gamma)}$, and enforce the Gauss laws

$$G_j^{(g)} = \overleftarrow{\mathcal{X}}_{j-1,j}^{(g)} \overrightarrow{X}_j^{(g)} \overrightarrow{\mathcal{X}}_{j,j+1}^{(g)} \overset{!}{=} 1, \tag{D.3}$$

where the notation $\overset{!}{=}$ means the equality holds on the constrained gauge-invariant Hilbert space. When gauging a symmetry, the Gauss laws should trivialize the symmetry operators representing the commutative sub-algebra of the symmetry operator algebra (e.g., the $Z(G)$ sub-symmetry of $G$).[22] The Gauss laws (D.3) satisfies this requirement because for any $z \in Z(G)$,

$$\prod_j G_j^{(z)} = U_z, \tag{D.4}$$

so enforcing $G_j^{(g)} \overset{!}{=} 1$ causes $U_z = 1$.

The Gauss laws introduce gauge redundancies to the enlarged Hilbert space of site and link $G$ qudits, and physical states and operators are those on which $G_j^{(g)}$ acts trivially. Therefore, the physical Hilbert space after gauging is spanned by all states $|\psi\rangle \in \bigotimes_{j=1}^L \mathbb{C}[G \times G]$ satisfying

---

[21]The $G$-symmetric operator $\sum_{h \in [g]} \overrightarrow{X}_j^{(h)}$ is included in the bond algebra once we include $\overleftarrow{X}_j^{(g)}$ as a generator of the bond algebra since $\sum_{h \in [g]} \overrightarrow{X}_j^{(h)} = \sum_{h \in [g]} \overleftarrow{X}_j^{(\bar{h})}$.

[22]Gauging involves both projecting to the symmetric subspace (enforcing Gauss law) *and* inserting symmetry defects into the Hamiltonian (adding states and minimal coupling). The insertion of symmetry defects explicitly breaks the symmetry being gauged down to the sub-symmetry that acts trivially on them, and, therefore, only this sub-symmetry is trivialized when enforcing the Gauss laws. Thus, when gauging the entire symmetry, Gauss's law trivializes only its commutative sub-algebra.

$G_j^{(g)}|\psi\rangle = |\psi\rangle$. Not all operators in the bond algebra (D.2) commute with $G_j^{(g)}$. to make (D.2) gauge-invariant, we minimally couple in the new $G$-qudit Pauli operators to construct the dual bond algebra

$$\mathfrak{B}^{\mathrm{mc}}[G] = \left\langle \overleftarrow{X}_j^{(g)}, \; [Z_j^{(\overline{\Gamma})}\, \mathcal{Z}_{j,j+1}^{(\Gamma)}\, Z_{j+1}^{(\Gamma)}]_{\alpha\beta}, \; \cdots \right\rangle_{G_j^{(g)} \overset{!}{=} 1}. \tag{D.5}$$

It is convenient to solve the Gauss constraint by gauge fixing to the unitary gauge. This is implemented by a change of basis using the unitary operator

$$W = \prod_j W_j, \qquad W_j = \sum_{\boldsymbol{h},\mathsf{h}} \big| \boldsymbol{h}; \cdots, \mathsf{h}_{j-1,j}\, h_j, \; \bar{h}_j\, \mathsf{h}_{j,j+1}, \cdots \big\rangle \langle \boldsymbol{h};\mathsf{h}|, \tag{D.6}$$

where $\boldsymbol{h}$ and $\mathsf{h}$ denote the configurations of $G$-qudits on sites and links, respectively. The unitary operator $W_j$ is a group-based control gate that shifts the value of $G$-qudits on links $\langle j-1, j\rangle$ and $\langle j, j+1\rangle$ according to the $G$-qudit state on the site $j$. Using this unitary operator, the Gauss operator transforms to

$$W\, G_j^{(g)}\, W^\dagger = \overrightarrow{X}_j^{(g)}, \tag{D.7a}$$

while the minimally coupled bond algebra (D.5) becomes

$$\mathfrak{B}^\vee[G] \equiv W\, \mathfrak{B}^{\mathrm{mc}}[G]\, W^\dagger = \left\langle \overleftarrow{\mathcal{X}}_{j-1,j}^{(g)}\, \overleftarrow{X}_j^{(g)}\, \overrightarrow{\mathcal{X}}_{j,j+1}^{(g)}, \; [\mathcal{Z}_{j,j+1}^{(\Gamma)}]_{\alpha\beta}, \; \cdots \right\rangle_{G_j^{(g)} \overset{!}{=} 1}. \tag{D.7b}$$

Since the Gauss operator in this new basis is (D.7a), the Gauss laws become $\overrightarrow{X}_j^{(g)} \overset{!}{=} 1$. The only $G$-qudit state for which $\overrightarrow{X}^{(g)} = 1$ for all $g \in G$ is $|\boldsymbol{1}\rangle \equiv |G|^{-1/2} \sum_g |g\rangle$. Therefore, after transforming by $W$, states in the physical Hilbert space have the general form

$$|\psi\rangle = \overbrace{|\boldsymbol{1}, \boldsymbol{1}, \cdots, \boldsymbol{1}\rangle}^{\text{site } G\text{-qudits}} \otimes \overbrace{\left( \sum_{\{\mathsf{g}\}} C_{\mathsf{g}}\, |\mathsf{g}\rangle \right)}^{\text{link } G\text{-qudits}}, \tag{D.8}$$

with the link $G$-qudits unconstrained. Because each site $G$-qudit is polarized in the $|\boldsymbol{1}\rangle$ state, they decouple from the model. More precisely, in the physical Hilbert space after gauging, $\overleftarrow{X}_j^{(g)} = 1$ which causes the dual bond algebra (D.7b) to become

$$\mathfrak{B}^\vee[G] = \left\langle \overleftarrow{\mathcal{X}}_{j-1,j}^{(g)}\, \overrightarrow{\mathcal{X}}_{j,j+1}^{(g)}, \; [\mathcal{Z}_{j,j+1}^{(\Gamma)}]_{\alpha\beta}, \; \cdots \right\rangle, \tag{D.9}$$

which is free of any site $G$-qudit operators. The dual bond algebra (D.9) is symmetric under a dual $\mathrm{Rep}(G)$ symmetry generated by

$$R_\Gamma = \mathrm{Tr}\left( \prod_j \mathcal{Z}_{j,j+1}^{(\Gamma)} \right). \tag{D.10}$$

Therefore, the dual bond algebra of a $G$ symmetry is the bond algebra of a $\mathrm{Rep}(G)$ symmetry:

$$\mathfrak{B}^\vee[G] \simeq \mathfrak{B}[\mathrm{Rep}(G)]. \tag{D.11}$$

This is consistent with the field theory expectation from Ref. 52.

### D.2 Gauging Rep(*G*) symmetry

Having shown a way of gauging a finite $G$ symmetry that gives rise to a dual Rep($G$) symmetry, let us now gauge a Rep($G$) symmetry on the lattice to verify that it leads to a dual $G$ symmetry [52]. In particular, we consider the Rep($G$) symmetry whose symmetry operators are

$$R_\Gamma = \text{Tr}\left( \prod_{j=1}^{L} Z_j^{(\Gamma)} \right), \tag{D.12}$$

which are the same operators discussed in Section 3.1. The bond algebra for the Rep($G$) symmetry (D.12) is

$$\mathfrak{B}[\text{Rep}(G)] = \left\langle \overleftarrow{X}_j^{(g)} \overrightarrow{X}_{j+1}^{(g)}, \, [Z_j^{(\Gamma)}]_{\alpha\beta}, \, \cdots \right\rangle, \tag{D.13}$$

which is manifestly isomorphic to the dual bond algebra $\mathfrak{B}^\vee[G]$ in Eq. (D.9).

To gauge Rep($G$), we again introduce $G$-qudits on each link $\langle j, j+1 \rangle$ of the lattice, acted on by the group-based Pauli operators $\overrightarrow{\mathcal{X}}_{j,j+1}^{(g)}$, $\overleftarrow{\mathcal{X}}_{j,j+1}^{(g)}$, and $\mathcal{Z}_{j,j+1}^{(\Gamma)}$. However, we next differ from gauging $G$ by imposing different Gauss laws, namely the matrix product identities

$$[G_j^{(\Gamma)}]_{\alpha\beta} = [\mathcal{Z}_{j-1,j}^{(\Gamma)} Z_j^{(\Gamma)} \mathcal{Z}_{j,j+1}^{(\overline{\Gamma})}]_{\alpha\beta} \overset{!}{=} \delta_{\alpha\beta}. \tag{D.14}$$

Because we impose (D.14) for all irreps $\Gamma$, the Gauss laws implement the constraints $h_j = \bar{\mathsf{h}}_{j-1,j}\mathsf{h}_{j,j+1}$ for all the site and link qudit states $\boldsymbol{h}$ and $\mathsf{h}$, respectively. Furthermore, because the monoidal product of Rep($G$) is commutative, we expect the Gauss laws (D.14) to trivialize the Rep($G$) symmetry operators (D.12) on the nose. Indeed, because

$$\text{Tr}\left( \prod_{j=1}^{L} G_j^{(\Gamma)} \right) = R_\Gamma, \tag{D.15}$$

imposing $[G_j^{(\Gamma)}]_{\alpha\beta} \overset{!}{=} \delta_{\alpha\beta}$ enforces $R_\Gamma = 1$ and trivializes the entire Rep($G$) symmetry.

The Gauss laws introduce gauge redundancies in the enlarged Hilbert space of $G$-qudit states on sites and links of the lattice. The physical states and operators are those for which each $G_j^{(\Gamma)}$ acts trivially, so the physical Hilbert space after gauging Rep($G$) is spanned by all states $|\psi\rangle \in \bigotimes_{j=1}^{L} \mathbb{C}[G \times G]$ satisfying $[G_j^{(\Gamma)}]_{\alpha\beta} |\psi\rangle = \delta_{\alpha\beta} |\psi\rangle$. The Rep($G$) symmetric bond algebra (D.13) is not gauge-invariant since it includes operators that do not commute with $[G_j^{(\Gamma)}]_{\alpha\beta}$. However, just like when gauging an invertible symmetry, it can be made gauge-invariant by minimally coupling in the new $G$-qudit operators. By doing so, we find the dual bond algebra

$$\mathfrak{B}^{\text{mc}}[\text{Rep}(G)] = \left\langle \overleftarrow{X}_j^{(g)} \overleftarrow{\mathcal{X}}_{j,j+1}^{(g)} \overrightarrow{X}_{j+1}^{(g)}, \, [Z_j^{(\Gamma)}]_{\alpha\beta}, \, \cdots \right\rangle_{[G_j^{(\Gamma)}]_{\alpha\beta} \overset{!}{=} \delta_{\alpha\beta}}. \tag{D.16}$$

The dual bond algebra $\mathfrak{B}^{\text{mc}}[\text{Rep}(G)]$ can be simplified by solving for the Gauss constraints (i.e., gauge fixing to the unitary gauge). We do so by rotating the enlarged Hilbert space into a new basis using the unitary operator

$$W = \prod_j W_j, \qquad W_j = \sum_{\boldsymbol{h},\mathsf{h}} \left| \cdots, \mathsf{h}_{j-1,j}\, h_j\, \bar{\mathsf{h}}_{j,j+1}, \cdots; \mathsf{h} \right\rangle \langle \boldsymbol{h}; \mathsf{h} |. \tag{D.17a}$$

Using $W$, the Gauss operators becomes

$$W\,[G_j^{(\Gamma)}]_{\alpha\beta}\,W^\dagger = [Z_j^{(\Gamma)}]_{\alpha\beta}, \tag{D.17b}$$

and the bond algebra (D.16) is transformed to

$$\mathfrak{B}^{\vee}[\mathsf{Rep}(G)] \equiv W \, \mathfrak{B}^{\mathrm{mc}}[\mathsf{Rep}(G)] \, W^{\dagger} = \left\langle \overleftarrow{\mathcal{X}}_{j,j+1}^{(g)}, \ [\mathcal{Z}_{j-1,j}^{(\bar{\Gamma})} Z_j^{(\Gamma)} \mathcal{Z}_{j,j+1}^{(\Gamma)}]_{\alpha\beta}, \ \cdots \right\rangle_{[G_j^{(\Gamma)}]_{\alpha\beta} \overset{!}{=} \delta_{\alpha\beta}}. \qquad \text{(D.18)}$$

This can be further simplified using that the Gauss laws in this basis are $[Z_j^{(\Gamma)}]_{\alpha\beta} \overset{!}{=} \delta_{\alpha\beta}$, which cause the original $G$-qudits on the lattice sites to decouple. The dual bond algebra then becomes

$$\mathfrak{B}^{\vee}[\mathsf{Rep}(G)] = \left\langle \overleftarrow{\mathcal{X}}_{j,j+1}^{(g)}, \ [\mathcal{Z}_{j-1,j}^{(\bar{\Gamma})} \mathcal{Z}_{j,j+1}^{(\Gamma)}]_{\alpha\beta}, \cdots \right\rangle, \qquad \text{(D.19)}$$

which is clearly isomorphic to the $G$-symmetric bond algebra (D.2).[23] Therefore,

$$\mathfrak{B}^{\vee}[\mathsf{Rep}(G)] \simeq \mathfrak{B}[G], \qquad \text{(D.20)}$$

and the presented gauging procedure on the lattice correctly yields the expected dual $G$ symmetry arising from gauging a $\mathsf{Rep}(G)$ symmetry.

## D.3   Gauging a $Z(G)$ symmetry in a system of $G$-qudits

At the beginning of this appendix—Section D.1—we reviewed how to gauge a finite $G$ symmetry in a lattice system of $G$-qudits. It is possible, however, for a $1+1$D system of $G$-qudits to have a finite invertible symmetry whose symmetry group $H$ is a subgroup of $G$. In particular, if $H$ is a normal subgroup, then after gauging the $H$ sub-symmetry, the model of $G$ qudits on sites should become (in an appropriate basis) a tensor product model of $G/H$-qudits on sites and $H$-qudits on links. In this appendix, we describe such a gauging and gauge fixing procedure for $H = Z(G)$, but generalizing our discussion to other Abelian $H$ is straightforward. We will assume that $G$ is a non-Abelian group with a non-trivial center $Z(G)$.

When gauging $Z(G)$ in a system of $G$-qudits, particularly when solving the Gauss laws, it will be convenient to describe $G$ as an extension of the inner automorphism group $\mathsf{Inn}(G) \cong G/Z(G)$ by $Z(G)$. These three groups fit into the short exact sequence of groups

$$0 \longrightarrow Z(G) \overset{\iota}{\longrightarrow} G \overset{\pi}{\longrightarrow} \mathsf{Inn}(G) \longrightarrow 1, \qquad \text{(D.21)}$$

from which each group element $g$ of $G$ can be denoted by the pair $(x, z) \in \mathsf{Inn}(G) \times Z(G)$. The isomorphism classes of such group extensions are classified by the group cohomology $H^2(\mathsf{Inn}(G), Z(G))$. In particular, the extension class $[\gamma] \in H^2(\mathsf{Inn}(G), Z(G))$ defines the group multiplication rule

$$(x, z) \cdot (x', z') = (x\,x', \ z + z' + \gamma(x, x')), \qquad \text{(D.22)}$$

where $\gamma$ is a representative group 2-cocycle of $[\gamma]$, normalized as $\gamma(x, 1) = \gamma(1, x) = 0$. Using the group extension (D.21), a $G$-qudit can be decomposed into a $Z(G)$-qudit and an $\mathsf{Inn}(G)$-qudit. In particular, we label each many-body $G$-qudit state $|\boldsymbol{g}\rangle$ by $|\boldsymbol{x}, \boldsymbol{z}\rangle$, with each $x_j \in \mathsf{Inn}(G)$ and $z_j \in Z(G)$. Using this decomposition, the $X$-type $G$-based Pauli operators become[24]

$$\overrightarrow{X}_j^{(x,z)} = \sum_{\{\boldsymbol{w}, \boldsymbol{a}\}} \left| x_j \boldsymbol{w}, \ \boldsymbol{a} + z_j + \gamma(x_j, w_j) \right\rangle \langle \boldsymbol{w}, \boldsymbol{a} |, \qquad \text{(D.23)}$$

$$\overleftarrow{X}_j^{(x,z)} = \sum_{\{\boldsymbol{w}, \boldsymbol{a}\}} \left| \boldsymbol{w}\,\bar{x}_j, \ \boldsymbol{a} - z_j - \gamma(w_j \bar{x}_j, x_j) \right\rangle \langle \boldsymbol{w}, \boldsymbol{a} |, \qquad \text{(D.24)}$$

---

[23]The Gauss laws (D.14) produce a dual bond algebra invariant under the $G$ symmetry operator $\prod_j \overrightarrow{\mathcal{X}}_{j,j+1}^{(g)}$. The Gauss laws $[\mathcal{Z}_{j-1,j}^{(\bar{\Gamma})} Z_j^{(\Gamma)} \mathcal{Z}_{j,j+1}^{(\Gamma)}]_{\alpha\beta} \overset{!}{=} \delta_{\alpha\beta}$ also gauge $\mathsf{Rep}(G)$ and lead to a dual bond algebra of a $G$ symmetry, but one that is represented by $\prod_j \overleftarrow{\mathcal{X}}_{j,j+1}^{(g)}$.

[24]In defining the operator $\overleftarrow{X}_j^{(x,z)}$, we use that the inverse of $(x, z)$ is $(\bar{x}, -z - \gamma(x, \bar{x}))$ and that as a 2-cocycle, $\gamma(x, x')$ satisfies $\gamma(x, \bar{x}) = \gamma(\bar{x}, x)$ and $\gamma(w, \bar{x}) + \gamma(w\bar{x}, x) = \gamma(\bar{x}, x)$.

which are defined to satisfy (D.22). To define the $Z$-type group based Pauli operators, we first note that because $(x, z) = (1, z) \cdot (x, 0)$, the matrix elements of $\Gamma(g)$ can be written as

$$[\Gamma(x, z)]_{\alpha\beta} = e^{i\phi_\Gamma(z)} [\Gamma(x, 0)]_{\alpha\beta} . \tag{D.25}$$

Using this, the $Z$-type group-based Pauli operators act on the $Z(G)$ and $\text{Inn}(G)$ qudits as

$$[Z_j^{(\Gamma)}]_{\alpha\beta} = \sum_{\{w, a\}} e^{i\phi_\Gamma(a_j)} [\Gamma(w_j, 0)]_{\alpha\beta} |w, a\rangle \langle w, a| . \tag{D.26}$$

Having expressed the $G$-qudit operators in terms of $Z(G)$ and $\text{Inn}(G)$ variables, let us now gauge the $Z(G)$ symmetry in a system of $G$-qudits represented by

$$U_z = \prod_j \overrightarrow{X}_j^{(1, z)} . \tag{D.27}$$

The bond algebra of $Z(G)$ symmetric operators is

$$\mathfrak{B}[Z(G)] = \left\langle \overrightarrow{X}_j^{(1, z)}, \overleftarrow{X}_j^{(x, 0)}, \overrightarrow{X}_j^{(x, 0)}, [Z_j^{(\overline{\Gamma})} Z_{j+1}^{(\Gamma)}]_{\alpha\beta}, [Z_j^{(\Gamma)} Z_{j+1}^{(\overline{\Gamma})}]_{\alpha\beta}, \cdots \right\rangle . \tag{D.28}$$

To gauge the $Z(G)$ symmetry, we introduce $Z(G)$-qudits on the links, which are acted on by the generalized Pauli operators $\overleftarrow{\mathcal{X}}_{j,j+1}^{(z)} = \overrightarrow{\mathcal{X}}_{j,j+1}^{(\tilde{z})}$ and $\mathcal{Z}^{(\rho)}$, and enforce the Gauss laws

$$G_j^{(z)} = \overleftarrow{\mathcal{X}}_{j-1,j}^{(z)} \overrightarrow{X}_j^{(1, z)} \overrightarrow{\mathcal{X}}_{j,j+1}^{(z)} \overset{!}{=} 1 . \tag{D.29}$$

The bond algebra (D.28) can be made gauge invariant by minimal coupling, which yields the dual bond algebra

$$\mathfrak{B}^{\text{mc}}[Z(G)] = \left\langle \overrightarrow{X}_j^{(1, z)}, \overleftarrow{X}_j^{(x, 0)}, \overrightarrow{X}_j^{(x, 0)}, [Z_j^{(\overline{\Gamma})} \mathcal{Z}_{j,j+1}^{(\rho_\Gamma)} Z_{j+1}^{(\Gamma)}]_{\alpha\beta}, [Z_j^{(\Gamma)} \mathcal{Z}_{j,j+1}^{(\rho_\Gamma)\dagger} Z_{j+1}^{(\overline{\Gamma})}]_{\alpha\beta}, \cdots \right\rangle_{G_j^{(z)}=1} . \tag{D.30}$$

To simplify the bond algebra (D.30), we use a unitary transformation to solve the Gauss law constraints. Namely, we rotate the Hilbert space to a new basis using the unitary operator

$$W = \prod_j W_j, \qquad W_j = \sum_{w, z, \mathsf{z}} |w, z; \cdots, z_{j-1,j} + z_j, z_{j,j+1} - z_j, \cdots\rangle \langle w, z; \mathsf{z}| , \tag{D.31}$$

where $(w, z)$ and $\mathsf{z}$ denote the configurations of $G$-qudits on sites and $Z(G)$-qudits on links, respectively. Conjugation by $W$ transforms the Hilbert space to a basis in which the Gauss operator is

$$W G_j^{(z)} W^\dagger = \overrightarrow{X}_j^{(1, z)} . \tag{D.32}$$

In this new basis, the physical Hilbert space is formed by states in the $\overrightarrow{X}_j^{(1, z)} = 1$ subspace, which is spanned by $|x; z\rangle \bigotimes_j \sum_{\mathsf{z}_j \in Z(G)} |\mathsf{z}_j\rangle$. Therefore, after gauging, the only remaining degrees of freedom are $\text{Inn}(G) \simeq G/Z(G)$-qudits on sites and $Z(G)$-qudits on links. We notice that on these basis states, the operators $\overrightarrow{X}_j^{(x, 0)} \equiv \overrightarrow{X}_j^{(x)}$, $\overleftarrow{X}^{(x, 0)} \equiv \overleftarrow{X}^{(x)}$ and $Z_j^{(\Gamma)}$ act as $\text{Inn}(G)$-qudit Pauli operators. After implementing these Gauss laws, the dual bond algebra becomes

$$\mathfrak{B}^\vee[Z(G)] = \left\langle \overrightarrow{\mathcal{X}}_{j-1,j}^{(z)} \overleftarrow{\mathcal{X}}_{j,j+1}^{(z)}, \overleftarrow{X}_j^{(x)}, \overrightarrow{X}_j^{(x)}, [Z_j^{(\overline{\Gamma})} \mathcal{Z}_{j,j+1}^{(\rho_\Gamma)} Z_{j+1}^{(\Gamma)}]_{\alpha\beta}, [Z_j^{(\Gamma)} \mathcal{Z}_{j,j+1}^{(\rho_\Gamma)\dagger} Z_{j+1}^{(\overline{\Gamma})}]_{\alpha\beta}, \cdots \right\rangle . \tag{D.33}$$

# E   Finite non-Abelian modulated symmetries

In this appendix, we highlight some aspects of non-Abelian finite modulated symmetries. In particular, the class of finite modulated symmetries generated by the unitary operators

$$U_q^{(g)} = \prod_j (\overrightarrow{X}_j^{(g)})^{f_j^{(q)}}, \qquad g \in G,$$

(E.1)

where $q = 1, 2, \cdots, n$ label the set of lattice functions $S = \{f^{(1)}, f^{(2)}, \cdots, f^{(n)}\}$ with $f_j^{(q)} \in \mathbb{Z}$ that defines the modulated symmetry. These symmetry operators generate an internal symmetry group $G_{\text{int}}$ that can be Abelian or non-Abelian. Since we take $U_q^{(g)}$ to be generators of the symmetry operators, we require the set of lattice functions $S$ to be independent. Because the symmetry operators obey

$$U_{q_1}^{(g)} \times U_{q_2}^{(g)} = \prod_j \overrightarrow{X}_j^{(g^{f_j^{(q_1)} + f_j^{(q_2)}})},$$

(E.2)

it is most natural to take $S$ to be independent over the ring $\mathbb{Z}_r$,[25] where $r = \text{lcm}(r_1, r_2, \cdots, r_{|G|})$ with $r_j$ the order of $g_j \in G$.

When $G$ is an Abelian group, all symmetry operators $U_q^{(g)}$ commute and obey

$$U_q^{(g)} \times U_q^{(h)} = U_q^{(gh)} \qquad \text{(Abelian } G\text{).}$$

(E.3)

Therefore, when $G$ is Abelian, the internal symmetry group $G_{\text{int}}$ is a subgroup of $G^{\times n}$. However, for non-Abelian $G$, the product of two symmetry operators,

$$U_q^{(g)} \times U_q^{(h)} = \prod_j \overrightarrow{X}_j^{(g^{f_j^{(q)}} h^{f_j^{(q)}})},$$

(E.4)

generally does not take the form $U_{q'}^{(g')}$ since $g^{f_j^{(q)}} h^{f_j^{(q)}} \neq (g')^{f_j^{(q')}}$, but instead is an entirely new symmetry operator. Furthermore, if the group elements $g$ and $h$ do not commute, then the two symmetry operators $U_q^{(g)}$ and $U_{q'}^{(h)}$ generally fail to commute as well whether $q$ and $q'$ are the same or different. Therefore, for non-Abelian $G$, the internal symmetry group $G_{\text{int}}$ is non-Abelian but much more complicated than just a subgroup of $G^{\times n}$ and depends sensitively on $f^{(q)}$. However, as long as there exists a finite integer $x_q$ such that $g^{f_j^{(q)}} = g^{f_{j+x_q}^{(q)}}$ for each $q$, then $G_{\text{int}}$ is a finite group. This is because each element of $G_{\text{int}}$ corresponds to a modulated action of $g \in G$, and there are finitely many ways to modulate a $g \in G$ symmetry action throughout $\text{lcm}(x_1, x_2, \cdots, x_n)$ lattice sites, which is the common periodicity of the generators (E.1). In particular, $G_{\text{int}}$ is a finite group of order $\leq |G|^{\text{lcm}(x_1, x_2, \cdots, x_n)}$.

Any Hamiltonian with the internal symmetry $G_{\text{int}}$ generated by (E.1) will be constructed from $G_{\text{int}}$-symmetric operators. $\overleftarrow{X}_j^{(g)}$ is a symmetry operator, while the operators $\overrightarrow{X}_j^{(g)}$ and $Z_j^{(\Gamma)}$ transform under $U_q^{(g)}$ as

$$U_g^{(q)} \overrightarrow{X}_j^{(h)} (U_g^{(q)})^\dagger = \overrightarrow{X}_j^{(g^{f_j^{(q)}} h g^{-f_j^{(q)}})},$$

(E.5)

$$U_g^{(q)} [Z_j^{(\Gamma)}]_{\alpha\beta} (U_g^{(q)})^\dagger = [\Gamma(g^{f_j^{(q)}})^\dagger]_{\alpha\gamma} [Z_j^{(\Gamma)}]_{\gamma\beta}.$$

(E.6)

---

[25]More precisely, the functions in $S$ are independent in the sense of Ref. [95, Section 3.1].

The operators $\overrightarrow{X}_j^{(g)}$ can be made symmetric by summing over elements in the conjugacy class $[g]$ of $g$:

$$\sum_{h \in [g]} \overrightarrow{X}_j^{(h)}, \tag{E.7}$$

which does not depend on the choice of modulated functions $f^{(q)}$. The symmetric operators constructed from $[Z_j^{(\Gamma)}]_{\alpha\beta}$, however, are more complicated. They have the generic form

$$\text{Tr}\left( \prod_\ell (Z_\ell^{(\Gamma)})^{D_{j,\ell}^{(\Gamma)}} \right), \tag{E.8}$$

and depend sensitively on $\Gamma$ and $S$. When $\Gamma$ is a $1d$ irrep, the trace over virtual states in (E.8) does nothing, and the requirement for (E.8) to be symmetric is

$$\sum_\ell D_{j,\ell}^{(\Gamma)} f_\ell^{(q)} = 0 \bmod N_\Gamma \qquad (d_\Gamma = 1), \tag{E.9}$$

where $N_\Gamma$ is the smallest positive integer satisfying $\bigotimes_{i=1}^{N_\Gamma} \Gamma = \mathbf{1}$. However, when $d_\Gamma > 1$, because the matrices $\Gamma(g)$ do not commute, the simplest expressions for (E.8) depends on $S$. However, a general symmetric operator of type (E.8) can be constructed from two $Z_\ell^{(\Gamma)}$ with

$$D_{j,\ell}^{(\Gamma)} = \delta_{j+\text{lcm}(x_1,x_2,\cdots,x_n),\ell} - \delta_{j,\ell} \qquad (d_\Gamma > 1). \tag{E.10}$$

## E.1  Finite $G$ dipole symmetry

In the remainder of this Appendix, we specialize the above to a finite $G$ dipole symmetry. This is symmetry generated by unitary operators

$$U_0^{(g)} = \prod_j \overrightarrow{X}_j^{(g)}, \qquad U_1^{(g)} = \prod_j (\overrightarrow{X}_j^{(g)})^j, \tag{E.11}$$

where $g \in G$. For Abelian $G$, the internal symmetry group $G_{\text{int}} = G \times G$ and each group element $(g_0, g_1) \in G \times G$ is represented by $U_0^{(g_0)} U_1^{(g_1)}$. For non-Abelian $G$, the symmetry group $G_{\text{int}}$ describing these operators' multiplication is more complicated. While $U_0^{(g)}$ represents a $G$ subgroup of $G_{\text{int}}$, $U_1^{(g)}$ and its products generate a symmetry group that is much larger than $G$, which has a nontrivial interplay with the $G$ sub-symmetry generated by $U_0^{(g)}$. Indeed, a general symmetry operator of a $G$ dipole symmetry corresponds to the tuple

$$( (g_1, n_1), (g_2, n_2), \cdots, (g_n, n_m) ) \equiv U_{n_1}^{(g_1)} \times U_{n_2}^{(g_2)} \times \cdots \times U_{n_m}^{(g_m)}, \tag{E.12}$$

where $g_\bullet \in G$ and $n_\bullet \in \{0, 1\}$.

For example, when $G = S_3$, we numerically find 324 such inequivalent tuples, which requires considering up to $m = 5$ tuples. Therefore, $G_{\text{int}}$ of an $S_3$ dipole symmetry is an order 324 non-Abelian finite group. By Burnside's theorem, this is a solvable group. Another example is $G = D_8$, for which we numerically find that $G_{\text{int}}$ is an order 128 non-Abelian finite group. Interestingly, while $|D_8| > |S_3|$, a $D_8$ dipole symmetry group is smaller than an $S_3$ dipole symmetry group. This is related to the fact that the generators of $S_3$ dipole symmetry have a larger periodicity, $\text{lcm}(x_1, x_2) = 6$, compared to the ones for $D_8$ dipole symmetry, $\text{lcm}(x_1, x_2) = 4$.

The simplest translation-invariant Hamiltonian that commutes with the $G$ dipole symmetry (E.11) is

$$H = \sum_j \left( \sum_{\Gamma:\, d_\Gamma=1} J_\Gamma\, Z_{j-1}^{(\Gamma)} (Z_j^{(\Gamma)})^{-2} Z_{j+1}^{(\Gamma)} + \sum_{\Gamma:\, d_\Gamma>1} J_\Gamma\, \text{Tr}\left( Z_j^{(\Gamma)\dagger} Z_{j+r}^{(\Gamma)} \right) + \sum_g K_g\, \overleftarrow{X}_j^{(g)} \right) + \text{H.c.}, \tag{E.13}$$

where in the second term, $r = \text{lcm}(r_1, r_2, \cdots, r_{|G|})$ with $r_j$ the order of $g_j \in G$. Notice that the last term in $H$ can be written in terms of (E.7), and the first two terms can be expressed as (E.8). Indeed, the symmetry operators act on $Z^{(\Gamma)}$ when $d_\Gamma = 1$ as a $\mathbb{Z}_{N_\Gamma}$ dipole symmetry, hence the first term is symmetric. The second term is symmetric because $g^{j+r} = g^j$ for all $g \in G$.

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
