# Peer review of "(SPT-)LSM theorems from projective non-invertible symmetries"

_SciPost Physics, doi:SciPost Phys. 18, 028 (2025)_

## Round 1 · Referee Report · Anonymous (Referee 1) · 2024-10-24

Strengths

  1. The paper studies a very interesting generalization of traditional LSM theorems to systems with non-invertible symmetries.
  2. The generalized XY model studied in the paper encompasses numerous intriguing details and a wide range of interesting phenomena is demonstrated by this model.

Weaknesses

  1. The paper focuses solely on non-invertible symmetry $Rep(G)\times Z(G)$, and potential generalizations to other non-invertible symmetries are not straightforward.

Report

The paper investigates the consequences of projective algebras formed by non-invertible symmetries and lattice translations in a generalized 1+1D quantum XY model based on group-valued qudits. This model is written in terms of G-qudits and enjoys a projective $Rep(G)\times Z(G)$ symmetry and translation symmetry. Depending on the detailed properties of $G$, this paper uncovers that the model either has some LSM anomaly or satisfies the condition of some SPT-LSM theorem.

As stated in the Strengths, this paper is very well-written and serves as a very timely addition to the subject. I recommend it to be published on Scipost.

Here are several questions that I have about the paper. The authors may consider integrating them in the paper if these questions are relevant.

  1. A natural question is how to generalize these results to any fusion-category symmetry in 1D lattice systems. Hence there are some natural questions regarding this. For example, for invertible symmetry $G$, projective representations of $G$ are classified by $H^2(G, U(1))$. Is there a similar classification for non-invertible symmetry? It is known that LSM anomaly is characterized by an element $H^3(G, U(1))$ (for example, see https://arxiv.org/abs/2401.02533 for the most recent account on this), where G contains both internal as well as translation symmetries, can I obtain a similar classification in the context of non-invertible symmetries? What about SPT-LSM?

  2. I am confused about the details of the SymTFT construction, and some clarification may be helpful. For example, why can we not also gauge the translation symmetry (by adding an extra $Z(\mathbb{Z}_L)$ quantum double in the symmetry-topological order, instead of SET)? This paper demonstrates that we can consider the nontrivial SPT state by calculating the Lagrangian algebra for $G=D_8$. Can we identify the corresponding Lagrangian algebra for general $G$ as well, whenever $Z(G)\subset [G, G]$? What about just LSM anomalies from the point of view of SymTFT? Do they correspond to condensable algebras, or something else? Also, according to https://journals.aps.org/prb/abstract/10.1103/PhysRevB.100.115147, I think that a consistent set of U-symbols has to be chosen (which are possibly trivial, but I think that it is still worth mentioning) when specifying the action of translation on the topological order. And do we need to choose a consistent set of eta-symbols as well?

  3. A smaller question, in Appendix C, can we just classify the Lagrangian algebra for $Z(Rep(D_8)\times \mathbb{Z}_2 \times \mathbb{Z}_2)$?

Requested changes

Just consider questions in the report and integrate them in the paper if relevant.

Recommendation

Publish (easily meets expectations and criteria for this Journal; among top 50%)

---

## Round 1 · Referee Report · Anonymous (Referee 2) · 2024-12-2

Report

The authors study a generalization of the 1+1d XY quantum spin chain model based on group-valued qudits for a finite group G. They study the global symmetries of such models and the corresponding LSM-type constraints/anomalies. These models have lattice translation symmetry and an internal Rep(G) x Z(G) symmetry, where Rep(G) is a non-invertible symmetry described by representations of G and Z(G) is an ordinary abelian symmetry described by the center of the group G.

They study the projective action of global symmetry in the presence of symmetry defects and examine its relation to LSM constraints. Unlike ordinary invertible symmetries, they show that such projective actions do not always lead to LSM constraints/anomalies. Furthermore, they relate the LSM constraint to a mixed 't Hooft anomaly by gauging the internal Rep(G) x Z(G) symmetry.

The paper contains important results and examples. I recommend the manuscript for publication in SciPost.

Recommendation

Publish (surpasses expectations and criteria for this Journal; among top 10%)

---

## Editorial Decision

published